

# IAP-AACM v1.0: Global to regional evaluation of the atmospheric chemistry model in CAS-ESM

Ying Wei[1,2], Xueshun Chen[1,4], Huansheng Chen[1], Jie Li[1], Zifa Wang[1,2,4], Wenyi Yang[1], Baozhu Ge[1], Huiyun Du[1,2], Jianqi Hao[1,*], Wei Wang[3], Jianjun Li[3], Yele Sun[1], Huili Huang[1]

[1] The State Key Laboratory of Atmospheric Boundary Layer Physics and Atmospheric Chemistry, Institute of Atmospheric Physics, Chinese Academy of Sciences, Beijing 100029, China

[2] University of Chinese Academy of Sciences, Beijing 100049, China

[3] China National Environmental Monitoring Center, Beijing 100012, China

[4] Center for Excellence in Regional Atmospheric Environment, Institute of Urban Environment, Chinese Academy of Sciences, Xiamen 361021, China

[*] Now at: School of Civil and Environmental Engineering, Georgia Institute of Technology, Atlanta, GA, 30332, USA

*Correspondence to*: Zifa Wang (zifawang@mail.iap.ac.cn)

## Abstract:

In this study, a full description and comprehensive evaluation of a global-regional nested model, the Aerosol and Atmospheric Chemistry Model of the Institute of Atmospheric Physics (IAP-AACM), is presented for the first time. Not only the global budgets and distribution, but also a comparison of nested simulation over China against multi-datasets are investigated, benefiting from the access of air quality monitoring data in China since 2013 and the Model Inter-Comparison Study for Asia



project. The model results and analysis can greatly help reduce uncertainties and understand model diversity in assessing global and regional aerosol effects, especially

over East Asia and areas affected by East Asia. The 1-year simulation for 2014 shows that the IAP-AACM is within the range of other models, and well reproduces both spatial distribution and seasonal variation of trace gases and aerosols over major continents and oceans (mostly within the factor of two). The model nicely captures spatial variation for carbon monoxide except an underestimation over the ocean that

also shown in other models, which suggests the need for more accurate emission rate of ocean source. For aerosols, the simulation of fine-mode particulate matter ($PM_{2.5}$) matches observation well and it has a better simulating ability on primary aerosols than secondary aerosols. This calls for more investigation on aerosol chemistry. Furthermore, IAP-AACM shows the superiority of global model, compared with

regional model, on performing regional transportation for the nested simulation over East Asia. For the city evaluation over China, the model reproduces variation of sulfur dioxide ($SO_2$), nitrogen dioxide ($NO_2$) and $PM_{2.5}$ accurately in most cities, with correlation coefficients above 0.5. Compared to the global simulation, the nested simulation exhibits an improved ability to capture the high temporal and spatial

variability over China. In particular, the correlation coefficients for $PM_{2.5}$, $SO_2$ and $NO_2$ are raised by ~0.25, ~0.15 and ~0.2 respectively in the nested grid. The summary provides constructive information for the application of chemical transport models. In future, we recommend the model's ability to capture high spatial variation of $PM_{2.5}$ is yet to be improved.





**Key words:** IAP-AACM, model evaluation, multi-model inter-comparison, PM$_{2.5}$,

China

## 1. Introduction

Atmospheric composition can affect climate and environment through direct and

indirect effects (Intergovernmental Panel on Climate Change (IPCC), 2001). The

composition of the troposphere has changed a lot due to anthropogenic activities over

the past decades (Akimoto, 2003; Tsigaridis et al., 2006). Changes in the

concentration of trace gases such as SO$_2$ and nitrogen oxides (NO$_x$ = NO + NO$_2$) have

a substantial impact on acid deposition (Mathur et al., 2003), atmospheric oxidation

(Calvert, 1984), and gas-particle transformation processes (Saxena et al., 1987).

Formation of aerosol from these precursor gases, together with aerosol from other

sources, have a direct radiative forcing which is estimated to be -0.3 ~ -1.0 W m$^{-2}$,

with a factor of two uncertainty (Luo et al., 1998). By modifying cloud properties, the

aerosols also have important indirect effects. As reported in the Fourth Assessment

Report (AR4) of IPCC (2007), the first indirect radiative forcing of aerosol ranges

from -1.8 ~ -0.3 W m$^{-2}$. This is more accurate than the result presented in IPCC AR3

(Penner et al., 2001) which ranges from -2.0 ~ 0 W m$^{-2}$, but there is still much

uncertainty. In addition, aerosols have adverse impacts on human health including

respiratory diseases and lung cancer, which has drawn increasing public attention (Li

et al., 2006; Kan and Chen ,2002; Pope et al., 2002). It is necessary to represent the

key physical and chemical parameters controlling trace gases and aerosols in order to




quantify these adverse effects and project the influence of aerosols in the future (IPCC, 2007).

Chemical Transport Models (CTMs) are mathematical tools for studying the evolution of chemical constituents in the atmosphere. CTMs have irreplaceable

advantages in terms of source and sink assessment of trace gases, historical process reproduction, and future scenario prediction. CTMs, together with observations and laboratory simulations, have become the main methods for atmospheric environmental research (Wang et al., 2008). But there are numerous uncertain factors affecting model results (e.g. meteorology, emissions and model framework and

physiochemical schemes). Therefore, model evaluation is essential for model development and scientific analysis. To date, many assessments with a single model using various observation datasets and multi-model inter-comparisons (with or without observations) have provided us with a comprehensive understanding of model performance and uncertainty. e.g., Badia et al. (2017) evaluated the gas-phase

chemistry of the Multi-scale Online Nonhydrostatic Atmosphere Chemistry model (NMMB-MONARCH), Mann et al. (2010) evaluated both mass concentration and number concentration of the Global Model of Aerosol Processes (GLOMAP), and Tsigaridis et al. (2014) gave a detail evaluation of organic aerosol in the Aerosol Comparisons between Observations and Models Project (AeroCom). However,

evaluation against site observation are mainly for America and Europe while inadequate for EA due to a limited set of data (Søvde et al., 2012; Lee et al., 2015; Kaiser et al. 2018). Besides, observation of China is scarcely included in model

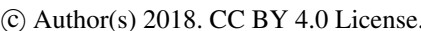
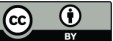

evaluation. Spatial distribution of aerosols affects estimation of radiative forcing (Shindell et al., 2013; Giorgi et al., 2003). Thereby, more observation used to test the

model results enables us to reduce uncertainties of climate effect prediction over EA.

Along with economic development and urbanization, most megacities in China have been plagued by haze in recent years. There are many reports on observation and simulation studies addressing particulate matter. The model studies mainly focus on the relationship between haze and weather conditions (Zhang et al., 2015; Tie et al.,

2015, 2017), pollutant source apportionment (Wang et al., 2013; Wang et al., 2014), and the chemical mechanism of particulate formation (Cheng et al., 2016). Regional models are more often used in local air pollution research due to its precise ability to capture the variation of inputs (e.g. meteorology, underlying surface and emissions) and therefrom the temporal and spatial variation of pollutants. However, the setting of

boundary condition limits continuous transportation from upwind and may lead to unrealistic description in the model.

Based on the Global Nested Grid Air Quality Prediction Model System (GNAQPMS) (Chen et al., 2015), we developed IAP-AACM and coupled it into CAS-ESM (the Earth System Model of the Chinese Academy of Science) as the

atmosphere chemistry component of the model, using the framework of coupler 7 (CPL7) (Tang et al., 2015;Zhu et al., 2018). As a multi-scale nested aerosol model based on air quality prediction in China, IAP-AACM not only has the capability for global and regional simulations, but also has strengths in localization of the process parameterization (e.g. dust mode and heterogeneous chemistry) (Wang et al., 2000; Li



et al., 2018). The development of IAP-AACM allows us to quantify climate effects on

a global scale and elucidate air pollution problems on a regional scale over China.

Here a large number of datasets are used to evaluate the model, including a dataset of

city sites covering China. Continuous year-round observations at city sites can help

study of air pollution and model evaluation in China. As we are currently building a

global forecasting platform, the model evaluation across a wide range of cities will

also provide knowledge for global model forecasting and assessment.

In this study, the off-line IAP-AACM is applied to a 1-year simulation for 2014

and the model results of trace gases and aerosol mass concentration are evaluated

against other model datasets and a range of observational datasets, including site

observations and satellite data. Firstly we present the global evaluation in section

3.1~3.2. The global budgets of sulfur (dimethylsulfide (DMS), $SO_2$ and sulfate) and

carbonaceous (organic matter (OM) and black carbon (BC)) aerosol are compared

with other aerosol models in section 3.1. The global distribution and evaluation of

trace gases and aerosol are shown in section 3.2. In section 3.3~3.4, we focus on the

model simulation of $PM_{2.5}$ and its components in Chinese cities. The nested

simulation is compared with an abundant dataset of city sites which cover most areas

in China, and the impact of different resolutions on model performance is also

explored. An inter-comparison with the Model Inter-Comparison Study for Asia

(MICS-Asia) models is presented in section 3.3, to give a general comparison across

East Asia (EA).

## 2. Model description and setup





## 2.1 Model description

### 2.1.1 CAS-ESM

CAS-ESM is the Earth System Model developed by the Chinese Academy of

Sciences. It is coupled with the Atmospheric General Circulation Model of IAP

(IAP-AGCM) (Su et al., 2014), the Climate System Ocean Model (LICOM) (Liu et al.,

2012), the Common Land Model (CoLM), the sea ice model (CICE), the Dynamic

Global Vegetation Model of IAP (IAP-DGVM) (Zhu et al., 2018), the IAP-AACM,

and the land and ocean biogeochemical models of IAP (IAP-OBGCM). The

IAP-AACM provides mass concentration of trace gases and aerosols for CAS-ESM

and thus provides the corresponding climate effect through the two-way feedback of

aerosols. Currently, global climate and ecological environment change is not only one

of the core issues of international climate and environment diplomacy, but also an

important factor governing the sustainable development of China. Earth system model

is a basic tool to understand and solve these problems. The resolution of the

CAS-ESM is $1°\times1°$ currently and later will be updated to $0.25°\times0.25°$. The CAS-ESM

will calculate a climate numerical experiment with high resolution for 100 years

(1950 ~ 2050) and provide simulation results for the sixth IPCC assessment report and

CMIP6.

### 2.1.2 IAP-AACM

The IAP-AACM is developed on the basis of the Nested Grid Air Quality

Prediction Model System (NAQPMS) (Wang et al. 2006b) and the Global Nested

Grid Air Quality Prediction Model System (GNAQPMS) (Chen et al., 2015).





NAQPMS/GNAQPMS is widely used in the simulation of dust (Li et al., 2012),

ozone ($O_3$) (Wang et al., 2006a; Li et al., 2007), deposition (Ge et al., 2014), air

pollution policy control (Wu et al., 2011; Li et al., 2016; Wei et al., 2017) and the

global transportation of mercury (Chen et al., 2015).

Like GNAQPMS, the IAP-AACM is a multi-scale nested model that describes

atmospheric chemistry and aerosol process on both global and regional scales. In the

IAP-AACM, sea salt and dust are emitted as dynamic sources. The dust scheme

originates from the wind erosion model developed by Wang et al. (2000) and

improved by Luo et al. (2006). The simulation of sea salt is based on the scheme of

Athanasopoulou et al. (2008). Dry deposition processes are based on the resistance

model approach of Zhang et al. (2003). The gas-phase chemistry scheme is Carbon

Bond Mechanism Z (CBM-Z) (Zaveri et al., 1999) which contains 176 chemical

reactions, 67 reactive species, and 20 species undergo photolysis. The cloud

convection, aqueous chemistry, in-cloud and below-cloud scavenging use the second

generation of Regional Acid Deposition Model (RADM2) (Stockwell et al., 1997).

For aerosols, the thermodynamic equilibrium module ISORROPIA (Nenes et al., 1998,

1999) is used to calculate gas-particle partitioning of inorganic aerosols and aerosol

water content. Furthermore, an aerosol microphysics dynamic module (APM) (Yu et

al., 2009) was added to expand the simulation from mass concentration to size

distribution (Chen et al., 2014, 2017). The secondary organic aerosol (SOA) module is

based on the mechanism developed by Strader (1999), considering two anthropogenic





emission precursors (toluene and other aromatic hydrocarbons) and four bio-emission

precursors (isoprene, monoterpene, etc.) (Li et al., 2011).

In addition, the IAP-AACM includes an updated DMS emission module from

Lana et al. (2011). The DMS concentration in seawater is calculated using 47,313

observations of the Global Surface Waters DMS database

(http://saga.pmel.noaa.gov/dms/) and an additional 63 observations in the South

Pacific (Lee et al., 2010). The IAP-AACM also provides a second, simplified

gas-phase chemistry mechanism based on a sulfur-only, chemistry box model. The

simplified sulfur-only scheme is specially designed for CAS-ESM to provide the

major aerosol components (sulfate, OM, BC, dust and sea salt). Retaining aerosols

with significant climatic radiative effects while cutting computational load, nitrate and

its chemical reactions are excluded. This approach is common in global aerosol

models such as the Integrated Massively Parallel Atmospheric Chemical Transport

(IMPACT) model (Liu et al, 2005) and GLOMAP (Mann et al., 2010). The simplified

scheme contains sulfur species ($SO_2$, DMS, and sulfur acid gas ($H_2SO_4$)), ammonia

($NH_3$) and hydrogen peroxide. Offline monthly fields of the oxidants hydroxyl radical

(OH), nitrate ion radical ($NO_3$), $O_3$ and super oxidation of hydrogen ($HO_2$), generated

from a simulation of the standard version of IAP-AACM, are read in and interpolated.

Chemical processes in the simplified version are the same as those in the standard

version except for the gas-phase scheme mentioned above. In this paper we focus on

evaluating simulations of chemical composition in the standard version model driven

by a global version of Weather Research and Forecasting version 3.3 (WRFv3.3). The



global WRF is an extension of mesoscale WRF that was developed for global weather

research and forecasting applications (Richardson et al., 2007).

### 2.2 IAP-AACM setup

In this study, the simulation region covers the globe at $1\,°\times1\,°$ resolution and has a

nested domain over EA at $0.33\,°\times0.33\,°$. Vertically, the model uses 20 layers, from the

bottom layer of about 50 m to the model top of 20 km, and about 10 layers are located

below 3 km. The simulation area of the nested domain is shown in Fig. 1. The

synchronous time step is 1800 s. The meteorology input frequency is 6 h in the global

domain but 3 h in the nested domain. The simulation period is from December 1st,

2013 to December 31th, 2014, and the first month is spin up time. Boundary

conditions for the nested region are provided by the parent grid. As stratospheric

chemistry is not considered in the IAP-AACM, the top boundary concentrations of $O_3$,

$NO_x$ and CO are prescribed from the Model for Ozone and Related Chemical Tracers

210     version4 (MOZART-4) (Emmons et al., 2010).

### 2.3 Emissions

By integrating data from publicly-released emission inventories, we compiled a

global high-resolution ($0.1\,° \times 0.1\,°$) emission dataset with source categories (29

species and 14 sectors) and interpolate it to the model resolution. The benchmark year

215     is 2010. Detailed information on the emissions is shown in Table 1. We note that

volcanic emissions are not yet considered here.

As a consequence of government control policy included in the Five-Year Plan

(FYP), China has achieved an obvious decrease in air pollution in the past ten years,



especially for $SO_2$. According to an announcement by the Ministry of Environmental

220    Protection                                    of                                    China

(http://www.zhb.gov.cn/gkml/hbb/qt/201507/t20150722_307020.htm),    the    country

completed the emission reduction task of $12^{th}$ FYP (2010~2015) ahead of schedule in

2014 with a reduction ratio reaching by 12.9%. As the FYP controls suppressed $SO_2$

emissions mainly in the energy and industry sectors, we adjusted the total $SO_2$

emission for 2014 by a factor of 0.9 in China. The annual mean $SO_2$ emission is

show in Fig. 1.

### 2.4 Meteorology and evaluation

As noted in 2.1.2, meteorological fields were provided offline by the WRFv3.3.

The temporal and horizontal spatial resolution of WRFv3.3 was consistent with

IAP-AACM. The atmosphere was divided into 27 vertical layers up to 10 hundred

Pascal (hPa). WRF was driven by the National Centers for Environmental Prediction

(NCEP) Final Analysis (FNL) data.

To understand the model performance overall, a comparison of annual mean

meteorological fields (temperature, wind and relative humidity) between WRF and

reanalysis data (National Centers for Environmental Prediction Reanalysis 1

(NCEP-R1)) are presented in Fig. 2. 443 surface sites in the nested domain are also

analyzed with the National Climate Data Center (NCDC) data and the statistical

parameters are shown in Table 2. A comparison of annual mean precipitation between

the model and reanalysis data from the Global Precipitation Climatology Project

(GPCP)) is also shown in Fig. S1. Globally, as shown in Fig. 2, the difference in



temperature at 2 m ($T_2$) and wind at 10 m ($W_{10}$) between the model and observation is

within 2 °C and 2 m s$^{-1}$ respectively, except in high-latitude areas. The relative

humidity at 2 m ($RH_2$) is generally underestimated on land and overestimated over the

ocean, but the difference in most areas is within ±10%. The difference in precipitation

is within 2 mm day$^{-1}$ except in equatorial regions. The frequently strong convection in

tropical areas is difficult to reproduce in the model. The agreement in $T_2$ and $RH_2$ with

observations is better than that of $W_{10}$, with annual correlation coefficients (R) of 0.98,

0.84 and 0.53, respectively. Generally, the meteorology calculated by WRF is similar

to observations.

**2.5 Observation data**

Trace gas observation data for CO, $O_3$, $SO_2$, and $NO_2$ in this paper are collected

from    the    World    Data    Center    for    Greenhouse    Gas    (WDCGG)

(http://ds.data.jma.go.jp/gmd/wdcgg/cgi-bin/wdcgg/catalogue.cgi),    the    Acid

Deposition    Monitoring    Network    in    East    Asia    (EANET)

(http://www.eanet.asia/product/index.html#datarep),    and    the    Chinese    National

Environmental    Monitoring    Center    (CNEMC)    (http://www.cnemc.cn).    Annual

observation data of particle and aerosol species are from the European Monitoring and

Evaluation Program (EMEP) (http://www.emep.int/), EANET, the United States

Environmental    Protection    Agency    (EPA)

(http://aqsdr1.epa.gov/aqsweb/aqstmp/airdata/download_files.html#Daily)    and    the

Interagency Monitoring of Protected Visual Environments (IMPROVE) network

(http://vista.cira.colostate.edu/improve/). As there is a lack of observations of BC and




organic carbon (OC) in Asia in 2014, we collected earlier site results from the China

Atmosphere Watch Network (CAWNET) from Zhang et al. (2008). Hourly air quality

data in China are downloaded from CNEMC. The other aerosol observations in China

are collected from monitoring sites of Nanjing and Wuhan, and scientific observation

at Xinzhou and Beijing (Chen et al., 2015). Aerosol Optical Depth (AOD) data from

the Moderate Resolution Imaging Spectroradiometer (MODIS) is used to evaluate the

simulated AOD. All these datasets are for 2014, except that the WDCGG is used as an

average of 2006~2015 and the CAWNET is for 2006. The observation datasets are

summarized in Table 3 and detailed information of the observation sites is given in

Table S1. Note that the observed species in Table 3 are not always available at the

corresponding sites.

       To focus on the severe haze problem in China, and to investigate the model

performance over China in more depth, we selected 89 stations in 12 cities

representing typical areas in China. The 12 cities are Beijing, Tianjin, and Langfang

(representing North China, NC), Shanghai, Nantong, and Yancheng (representing

Yangtze River Delta, YRD), Guangzhou and Zhongshan (representing Pearl River

Delta, PRD), Urumqi (representing Northwest China, NWC), Zhengzhou and Wuhan

(representing Central China, CC) and Chengdu (representing Southwest China, SWC)

(shown in Fig. 1). The daily mean city-averaged concentration of pollutants are

displayed in figures and used to calculate statistics. In addition, we collected the mass

concentrations of BC, OM, sulfate, nitrate and ammonium in Beijing, Xinzhou,

Nanjing and Wuhan (also shown in Fig. 1) to evaluate the model performance in



simulating aerosol components.

## 3 Model results and evaluation

### 3.1 Budgets

On account of the significant radiative effect of sulfate and carbonaceous aerosols,

their budgets play an important role in the climate change (Penner et al., 1998). So

here we elucidate the budgets of sulfate with its precursor gases (DMS and $SO_2$) and

carbonaceous aerosols.

The global sulfur budgets for DMS, $SO_2$ and sulfate in IAP-AACM are

summarized in Table 4, along with results from other global aerosol models including

IMPACT (Liu et al., 2005), Goddard Institute for Space Studies General Circulation

Model with TwO-Moment Aerosol Sectional (GISS-TOMAS) (Lee et al., 2010),

Atmospheric Chemistry and Climate Model Intercomparison Project (ACCMIP)

models (Lee et al., 2013) and the AeroCom models (Textor et al., 2006). The DMS

emission (23.3 TgS $yr^{-1}$) is within the range of other models (10.7~23.7 TgS $yr^{-1}$).

Note that the dry deposition of DMS is zero in IAP-AACM so the sink is just

oxidation. This treatment is common in some other models such as ModelE2-TOMAS

and ModelE2-OMA (Lee et al., 2015). As a result, we have a higher burden of DMS of

0.19 TgS, just outside the range (0.05~0.15 TgS), and a longer lifetime of 3 days. For

$SO_2$, the emissions are a bit lower than the reference range (54.3 TgS $yr^{-1}$ vs.

63.4~94.9 TgS $yr^{-1}$), and this is ascribed to the lack of volcanic emissions. The

volcanic emissions used in most models is based on the work of Andres and Kasgnoc

(1998) and Dentener et al. (2006), in which the average flux for $SO_2$ is about 12.5 TgS



yr$^{-1}$ including continuous degassing and explosive volcanoes. We also note that the benchmark year of the anthropogenic emissions used in the reference models is as early as 2000 or 1990, but that anthropogenic emissions in IAP-AACM are for 2010,

which may explain some of the discrepancy. The oxidation of DMS to SO$_2$ is 22.8 TgS yr$^{-1}$, within the range of other models' results. The aqueous-phase process is responsible for 61% of the oxidation to sulfate and gas-phase processes are responsible for the remaining 39%. Although it's a bit lower conversion efficiency for aqueous-phase chemistry compared with other models (about 70% ~ 80%), both

aqueous phase and gas phase oxidation are well within the range of other models. The SO$_2$ burden is at the high end of the reference range, because the aqueous-phase oxidation rate for SO$_2$ is higher than that in the gas-phase (Liu et al., 2005). Due to an inefficient removal in aqueous-phase oxidation (29.8 Tg S) and wet deposition (shown in Table 4 as zero), the lifetime of SO$_2$ in the model is a little longer than other models

(3 days vs. 0.6~2.6 days). In IAP-AACM, the emission of H$_2$SO$_4$ is assumed as 2.5% of the total sulfur emission. With a strong wet scavenging effect, 94% of sulfate is removed by wet deposition and the rest by dry deposition.

Table 5 presents the budgets for OM and BC with a range of results from other models including Liu et al. (2005), Lee et al. (2013), Lee et al. (2015), Textor et al.

(2006), and those listed in Liu et al. (2005). For the same reasons described above, the emissions of BC/OM are at the low end compared with other models (BC: 7.42 TgS yr$^{-1}$ vs. 7.4~19.0 TgS yr$^{-1}$;OM: 56.7 TgS yr$^{-1}$ vs 34~144 TgS yr$^{-1}$). The ratio of dry deposition to wet deposition for BC and OM is 15.8% and 13.6%, respectively. Both

the burden and lifetime of carbonaceous aerosol are within the other models' results.

The burden of BC and OM is 0.13 Tg and 1.16 Tg respectively and the lifetime is 6.4

days and 7.4 days respectively.

### 3.2 Global distribution and evaluation

### 3.2.1 Trace gases

Global annual-averaged surface-layer trace gas distributions from IAP-AACM

are evaluated against site observations in Fig. 3. Scatter plots of observation and

simulation data divided into 11 different geographical regions are exhibited in Fig. 4

and the corresponding Normalized Mean Bias (NMB) are shown in Table 6. The 11

geographical regions include Africa, Antarctica, Arctic Ocean, Asia, Atlantic Ocean,

Europe, Indian Ocean, North America (NAmerica), South America (SAmerica),

Oceania and Pacific Ocean. Fig. 5 shows the comparison of annual surface

concentrations of CO, $O_3$ and $NO_2$ between IAP-AACM and some HTAP atmospheric

chemical models including CAM-Chem (Lamarque et al., 2012), OsloCTM3 (Søvde

et al., 2012), and CHASER(Sudo et al., 2002).

Overall, the global surface CO simulation of IAP-AACM is lower than

observations, especially in natural source regions. Antarctic continents and oceans

have the largest difference between site and model as shown in Fig. 3, and the

difference can reach ~100 ppb. Fig. 4 also reveals a significant underestimation in

ocean areas, with NMB ranging from -0.59 to -0.45 shown in Table 6. Over

anthropogenic source regions, by and large, the model is consistent with site data in

eastern NAmerica, EA, and the coastal areas of SAmerica and Africa, but is about 50



ppb lower in western NAmerica and Europe. The scatter plot clearly shows a negative

bias between the model and observations, especially over the ocean. The lower model

results may be caused by underestimated emissions or overestimated OH. For AACM,

the tropospheric (200 hPa to the surface) mean OH derived by the model is $10.6 \times 10^5$

molec cm$^{-3}$. This value agrees well with a study of 14 models for 2000 by Voulgarakis

et al. (2013), where the mean OH concentration was estimated to be $11.1 \pm 1.8 \times 10^5$

molec cm$^{-3}$. On the other hand, the anthropogenic emission is 546.4 Tg yr$^{-1}$, lower

than some other emission inventory (e.g. ACCMIP with 610.5 Tg yr$^{-1}$) (Badia et al.,

2017). Janssens-Maenhout et al. (2015) pointed out that CO emission from HTAPv2

has an uncertainty of 15~100% and 35~150% in data from well maintained countries

and poorly maintained countries respectively. Furthermore, the underestimation of CO

is common in other models. Shindell et al. (2006) evaluated 26 global models and

showed that all the model results are lower than observations in the North Hemisphere

(NH) except in the tropics, and it is concluded that this is related to a lower CO

emission source. The spatial distribution of CO concentrations in IAP-AACM is

similar to that in other models from HTAP in 2010. High values are found in

industrial areas such as NAmerica, Europe and EA, and biomass burning areas such as

South Africa (SAfrica) and SAmerica. The other models also display lower CO

concentrations over ocean as in IAP-AACM.

As displayed in Fig. 3 and Fig. 4, the $O_3$ spatial distribution simulated by

IAP-AACM is in a good agreement with observations. The $O_3$ simulations at most

sites are within a factor of two of the observation and the majority of regions have a



NMB within $\pm 0.2$ (Table 6) except for Africa, Antarctica and Asia. It is worth noting that the annual concentrations of $O_3$ at the three sites in Southeast Asia are more than

twice the observed values. As the sites are coastal, it may be not representative of the wider region simulated by the model. On the other hand, South Asia is a high-emission area for biogenic VOCs. Uncertainty in the biogenic source inventory may also cause large errors in $O_3$ simulation due to photochemical processes. The model shows a good skill in capturing the seasonal variation of surface $O_3$ in different

regions, as shown in Fig. 6. In the NH, the maximum $O_3$ concentration occurs in spring or summer on land but over the sea the value is higher in spring and lower in summer. In contrast, higher values occur in autumn or summer in the Southern Hemisphere (SH). The model results match the seasonal cycle both in the trends and the concentration over the ocean in the NH and in the SH very well, with only a small

underestimation in Antarctica. However, there is a positive bias during July-September over the land in the NH (NAmerica, Europe and Asia). A similar overestimation occurs in the evaluation of NMMB-MONARCH by Badia et al. (2017). They suspect this may be caused by a reduction of the $NO_x$ titration effect in the summer, which leads to corresponding lower $O_3$ concentrations in highly

industrialized regions. The horizontal distribution of $O_3$ in IAP-AACM is generally similar to that in other models, as shown in Fig. 5. High concentrations mainly occur downwind of highly polluted areas due to the $NO_x$ titration mentioned above. The model exhibits a higher concentration in the source regions and a lower concentration downwind. However, the model result is in good agreement with observations over





the ocean, as shown in Fig. 6. The difference in $O_3$ between models may partly be

related to the dry deposition scheme used by the model. We show the dry deposition

velocity (mainly 0.04 ~ 0.05 cm s$^{-1}$ on water surface) of $O_3$ in IAP-AACM in Fig. S2.

In the common parameterization $v_d = \frac{1}{r_a + r_b + r_c}$, the canopy resistance $r_c$ is the

dominant term for $O_3$ dry deposition to water surfaces (Luhar et al., 2017). It is

commonly assumed that $r_c$ for water is constant ($\approx$ 2000 s m$^{-1}$) (Wesely, 1989) and

this is used by default in many global models including CAM-Chem. Based on this,

we do not believe there are large differences in dry deposition between IAP-AACM

and the other models.

NO$_2$ is limited to continental source regions due to human activities. IAP-AACM

captures the spatial characteristics well, see Fig. 3. Concentrations in NAmerica,

Europe, and most parts of EA are in good agreement with observations. As shown in

Fig. 4, simulation results are within a factor of two of the observations at most sites

except for a few sites in China where they are underestimated (NMB of -0.59). As we

use the same anthropogenic emission inventory, the spatial distribution of NO$_2$ in NH

from IAP-AACM and other models is similar, as shown in Fig. 5. The maxima in the

NH are located in industrial areas due to fossil fuel combustion, and the concentration

of NO$_2$ is much higher in eastern China (>20 ppb) than that in eastern NAmerica and

Europe (3-10 ppb). The maxima in the SH are located in SAmerica and South Africa

due to biomass burning, where NO$_2$ is 1-10 ppb, slightly lower (~3 ppb) than the other

models shown in Fig. 5, probably related to the different biomass burning inventory

used (GFED3 vs. GFED4) (Janssens-Maenhout et al., 2015). The concentration over



the ocean is lower than 0.1 ppb except at 30 N~60 N, since the source of $NO_2$ over

the ocean is small, mainly emitted by ships.

Similar to $NO_2$, $SO_2$ is high in the NH and low in the SH. The source of $SO_2$ over

the ocean is mainly DMS oxidation from marine organisms while the source over land

is mainly fossil fuel combustion. Maximum concentrations are mainly found in

NAmerica, Europe, India and EA. As a result of environmental protection policies

lagging behind industrial development, $SO_2$ concentrations in India and China can

reach 15-20 ppb, much higher than that in the US or Europe. As shown in Fig. 3, the

IAP-AACM calculates a distribution of 0.1-20 ppb in EA and 0.1-5 ppb in western

NAmerica, which is consistent with observations. The simulation of $SO_2$ in eastern

NAmerica and Europe is about 1-10 ppb, both of which are overestimated with

NMB=3.52 and NMB=0.52 respectively, as shown in Table 6.

### 3.2.2 Aerosol composition

Fig. 7 and Fig. 8 show the annual surface concentrations of aerosol and $PM_{2.5}$ in

IAP-AACM in comparison with site observations in Europe, NAmerica and Asia. The

corresponding NMB in different regions are also displayed in Table 6. Overall,

aerosol simulations are consistent with site datasets.

As shown in Fig. 7, Sulfate, Nitrate and Ammonium (SNA) are mainly distributed

in NH due to their close association with human activities. Sulfate is reproduced

accurately at most sites. The model result is consistent with site records in Asia and

Europe and the simulations at most sites here are within a factor of two of

observations as shown in Fig. 8, with NMB of 0.36 and 0.11 respectively. In America,




the simulation of sulfate is about 2 μg m$^{-3}$ higher than observations. This is consistent

with the high level of SO$_2$ in the eastern America described previously, as the

precursor of sulfate. The simulation of nitrate over the land is more uncertain due to

the complex photochemical reactions of NO$_x$, but our model reproduces the nitrate

distribution overall. The simulation is close to observation in America but

overestimates it in Europe. There is an underestimation of ~5 μg m$^{-3}$ in Southeast Asia

and Japan. Even so, the NMBs are within ±0.8 (the NMB of America, Europe and

Asia is 0.5, 0.74 and -0.61 respectively). Generally, the simulation of ammonium is

comparable to observations in America and most parts of Asia, with an NMB of -0.46

and 0.85 respectively. But there is a positive bias in Europe with an overestimation of

about ~4 μg m$^{-3}$.

For carbonaceous aerosols, there are high values in developing countries or

regions as biomass burning and fossil fuel combustion dominate the sources.

SAmerica and SAfrica contribute most of biomass combustion emissions, and

developing countries (e.g. China and India) where industrial emission controls are less

well implemented contribute most of fossil fuel emissions. BC and OC are low in

America and Europe, with values of ~1 μg m$^{-3}$ and ~3 μg m$^{-3}$ respectively. By and

large, the model results are consistent with observations in the three regions shown in

Fig. 8, with the NMB of BC within ±0.65 and OC within ±0.7. The simulations of

both BC and OC are highly consistent with the IMPROVE dataset in America as well

as with the EMEP dataset in Europe. The accuracy of the simulation mainly depends

on the emission source, since BC is quite inert to chemical reactions. The simulation

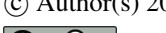



of BC in China is accurate with 70% of the stations within a factor of two of observation while OC is underestimated by about 5-10 μg m$^{-3}$, mainly due to the decrease in emissions from 2006 to 2014 due to the government's pollution control policy (Zheng et al., 2018).

Generally, the model shows good skill in simulating PM$_{2.5}$. Model results at most sites are close to observation as shown in Fig. 7, especially in Europe and Asia with NMB of -0.35 and -0.36 respectively. The underestimation in urban areas in western China may be related to uncertainty of emissions.

### 3.2.3 Comparison with satellite data

In order to evaluate the simulation of aerosol in IAP-AACM above the surface, we compared the aerosol optical depth (AOD) of IAP-AACM with MODIS satellite data. The calculation of light-extinction coefficient, $b_{ext550}$ (1/Mm, at 550nm), follows equation (1) given by Li et al. (2011):

$$b_{ext550} = 3.0 \times f_{SNA}(RH)\{[(NH_4)SO_4]+[(NH_4)NO_3]\} + 4.0 \times [OC] \\ +10.0 \times [LAC]+1.0 \times [FD]+0.6 \times [CD]+1.7 \times f_{ss}(RH) \times [SS] \tag{1}$$

where $f_{SNA}(RH)$ and $f_{SS}(RH)$ represent the hygroscopic growth factor for SNA and sea salt respectively, and the variables in brackets are the mass concentration of aerosol species (OC: organic carbon; LAC: light-absorbing carbon; FD: fine dust; CD: coarse dust; SS: sea salt).

Fig. 9 shows the comparison of AOD between IAP-AACM and MODIS for each

season in 2014. In general, the model AOD reproduces the spatial features of aerosol exhibited by satellites globally. For example, the high value around 60 °S, ranging from 0.1 to 0.3, is due to high concentrations of sea salt. The maximum in SAmerica

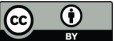

and SAfrica is due to the large amount of carbonaceous aerosol produced by biomass

burning. The desert maximum over 0.5 is caused by mineral dust in NAfrica, Arabian

Peninsula and western China. High AOD in NAmerica, Europe, India, and EA is

caused by anthropogenic aerosols. Furthermore, there is a good agreement of the

seasonal variations with satellite observations. For example, the AOD in the desert

areas of NH reaches a maximum in March-April-May (MAM) since this is the season

with frequent dust storms. SAmerica and SAfrica have lowest AOD in the

September-October-November (SON) season as there is less biomass burning in this

season. The east of China has highest AOD in December-January-February (DJF), as

the country suffers severe haze at this time. However, there are several biases between

model and satellite. The model shows a weaker AOD in Southeast Asia than

observation where the value is mainly controlled by biomass burning. The AOD from

IAP-AACM is also lower than observation to about ~0.4 in eastern China, mainly due

to the negative bias of anthropogenic aerosols.

### 3.3 Nested simulation evaluation

3.3.1 Distribution and evaluation in EA

Fig. 10 shows the annual distribution of the four pollutants $SO_2$, $NO_2$, $PM_{10}$ and

$PM_{2.5}$, against 45 city stations from the nested simulation. In general, the simulation

shows better agreement with sites in Eastern China than Western China. Model results

for $NO_2$ concentration are a little low, probably due to an underestimation of the

emissions, while $SO_2$, $PM_{10}$ and $PM_{2.5}$ are highly consistent with observations. The

model values of $PM_{10}$ and $PM_{2.5}$ include dust aerosol in Fig. 10, and dust contributes a



lot to particles in North China. In the following discussion of PM$_{2.5}$, we focus on

primary PM$_{2.5}$, BC, OC, SNA and SOA. The distribution of PM$_{2.5}$ and its precursors

show high levels in the East and low in the West, which is related to urbanization and

industrialization. Despite good agreement with observations in most places, the

concentration in Tibet is greatly underestimated.

510       Unlike global models, regional models need boundary conditions and total mass

in a region is not conserved. The physical and chemical processes are relatively

complex, with higher spatio-temporal resolution and more computationally-intensive.

Here a comparison between IAP-AACM and several regional models of MICS-Asia is

presented in Fig. 11. The MICS-Asia models included here are WRF-Chem (Tuccella

et al., 2012), CMAQ (Mebust et al., 2003) and NAQPMS (Wang et al., 2006). Their

simulations are for 2010 with the same meteorological fields and emissions, and the

horizontal resolution is 45 km. As shown in Fig. 11, the nested simulation of

IAP-AACM has consistent spatial distribution of pollutants in the far northwest of the

domain, which cannot be reproduced in the regional models. The fixed boundary

conditions are responsible for this phenomenon, for there is no transportation from

upwind of the boundary. Overall, the IAP-AACM shows similar annual distributions

to MICS-Asia models in EA, as the emission inventory used in IAP-AACM is largely

the same as MICS-Asia models. But concentrations in IAP-AACM are slightly lower

in China and Japan, especially for NO$_2$ and PM$_{2.5}$. Furthermore, NAQPMS shows

stronger transportation downwind of the continent. As the regional version of

IAP-AACM, NAQPMS has the same dynamic framework and physicochemical



processes as IAP-AACM, but the meteorology conditions are not the same.

3.3.2 Trace gas evaluation in cities

To get deeper insight into the performance of IAP-AACM in cities, a nested

simulation was compared with daily averaged observations in 12 cities across China.

We first focus on $NO_2$ and $SO_2$ since they are precursors of SNA aerosols. The

monthly variation of $SO_2$ and $NO_2$ against observations is shown in Fig. 12 and Fig.

13. Three-quarters of cities show an annual concentration of $NO_2$ of around 50 μg m$^{-3}$,

twice as high as $SO_2$ in summer and autumn, owing to the tougher $SO_2$ emission

reduction policy implemented since 2005. For $SO_2$, the model shows good agreement

with observations except in Wuhan as shown in Fig. 12. This probably implies an

overestimation of emissions in this city. Furthermore, IAP-AACM reproduces the

seasonal variation well, showing good comparison to observations with R over 0.5 in

most cities. In particular, the cities in NC have a high R in the range 0.76-0.89.

As illustrated in Fig. 13, the model shows a good performance for $NO_2$ in most

cities, especially in YRD, PRD and SWC. The statistics summarized in Table 7 clearly

show that the model captures the monthly variations well, with R of 0.49-0.7 in NC,

YRD and PRD, and matches the observations well with Root Mean Square Error

(RMSE) typically less than 20 μg m$^{-3}$ in SWC and PRD. Overall, the model results are

more likely to be overestimated in NC, YRD and CC in summer. As the "$NO_2$" values

reported by routine monitoring sites are $NO_2^*$, which partially includes $HNO_3$ and

$NO_3^-$, it is common to underestimate the observed "$NO_2$". Here we also give a

comparison of $NO_2$ column concentration between model and satellite observations



from GOME-2A in Fig. S3. Generally, there is a good agreement in $NO_2$ column

concentration in China in different seasons. The model significantly underestimates

concentrations in Urumqi, probably as a consequence of uncertainties in emissions. In

brief, the nested forecasting generally captures the seasonal variation of $NO_2$ with R

near or above 0.5 in NC, YRD and PRD, but shows a poorer performance in the other

parts of China.

3.3.3 Aerosol composition evaluation in cities

As shown in Fig. 14, the model performs very well in the simulation of $PM_{2.5}$.

Statistics are shown in Table   7. The model reproduces $PM_{2.5}$ trends over the 12

cities well, particularly in NC, YRD and SWC, with an R of 0.70-0.79, 0.71-0.80 and

0.77 respectively. The model results are close to or slightly lower than site

observations. The concentration in NC on some winter days is below the observations.

This underestimation of $PM_{2.5}$ in severe haze periods is common in atmospheric

chemistry models, mainly as a result of the deficiency in the SNA and Secondary

Organic Aerosol (SOA) simulation (Zheng et al., 2015; Donahue et al., 2006). Besides,

there is a clear underestimation in PRD and Urumqi where mean values are less than

half of the observations, with NMB around -0.5. This is partly caused by the

underestimation of emission sources, as $NO_2$ is lower in these cities. Furthermore,

dust plays an important role as a component of $PM_{2.5}$ in Urumqi, and this is not

included in the result. For these reasons, we investigate the aerosol simulation in

further detail below.

570        To assess the performance of IAP-AACM in representing aerosol components,



we compared the model results with 4 stations in NC, YRD and CC in Fig. 15. Generally, the model represents the variation of BC well with R ranging from 0.5 to 0.8 and the value get close to observations. As a primary specie, it is probably on account of its emission inventory. Unlike BC, there is an underestimation of OM at

the four stations, with a difference of 8-12 μg m$^{-3}$. For SNA aerosols, the model also underestimates observations. Sulfate is close to observations in the northern cities (Beijing and Xinzhou), but is underestimated in southern and central cities (Nanjing and Wuhan) by about 10 μg m$^{-3}$, with R from 0.4 to 0.6. As noted above, the concentration of $SO_2$ in Wuhan is overestimated. This suggests the sulfur oxidation

may be insufficient. This insufficient conversion has been discussed widely in recent years (Cheng et al., 2016; He et al., 2014). Moreover, $SO_2$ discharged by coal power plants plays a vital role in the formation of sulfate. The coarse grid resolution is insufficient to reproduce the rapid conversion of $H_2SO_4$ to particles in the plume. The gas-phase oxidation ($SO_2 + OH \rightarrow H_2SO_4(g)$) is very sensitive to meteorological

variables (particularly radiation and temperature) and gas (OH and NOx) concentration around the stacks (Stevens et al., 2012). This may lead to a local discrepancy between simulation and observation. The results for ammonium show similar characteristics. The simulation of nitrate is highly underestimated with R ranging from 0.3 to 0.5. The underestimation is due to a high frequency of 'zero'

value in daytime of summer and autumn. This is due to the sensitivity of nitrate to temperature in the thermodynamic equilibrium module which leads to decomposition to $NO_2$. This also leads to overestimation of $NO_2$ as shown in in Fig.13. Schaap et al.

(2011) found the same phenomenon in the LOTOSEUROS model using ISORROPIA

and recommended improvements in the equilibrium module, including coarse mode

nitrate. In other respects, the model can reproduce aerosol components reasonably

well in these cities

### 3.4 Global versus regional results

To evaluate the improvements due to higher model resolution, we compare the

global simulation ($1°\times1°$) with the nested simulation ($0.33°\times0.33°$) over China. Table

7 gives the statistics of $PM_{2.5}$, $SO_2$ and $NO_2$ simulated at the different resolutions. The

nested domain can effectively improve the simulation of city pollutants, especially

$PM_{2.5}$, because high-resolution grid can provide better resolved emissions and

meteorological fields in urban and rural areas. As shown in Table 7, the correlation

coefficients of the three species in the nested simulations are significantly higher than

in the global simulations. The RMSE of the nested results in most cities are reduced.

For $PM_{2.5}$, the correlation at high resolution rises in all the cities except Shanghai, and

the PRD has the most significant increase. The R for Guangzhou and Zhongshan

increase by 0.2 and 0.25 respectively, and the R for Urumqi increases by 0.19.

Moreover, the RMSE decreases over 9 cities. The impact of the nested grid on

simulation of $SO_2$ is clear, with R increasing over 8 cities getting R raised and RMSE

reducing over 9 cities. In particular, the simulation in NC, YRD and SCW improves

significantly, with better representation of monthly variation and closer comparison to

observations. For $NO_2$, the R significantly increases in 9 cities (all except Shanghai,

Nantong and Wuhan) and RMSE decreases in 7 cities. The best performance is in



Beijing where there is an improvement of R from 0.48 to 0.68.

## 4. Conclusions

A global-nested aerosol and atmospheric chemistry model coupled into CAS-ESM is introduced in this study. The aim is to provide more precise information on climate effects and air pollution on both global and regional scales. In IAP-AACM,

the emissions of sea salt, dust and DMS are calculated online. Gas-phase chemistry includes both the CBM-Z module and a simplified reaction scheme without nitrate chemistry specifically for CAS-ESM. Aqueous chemistry and wet deposition are calculated by RADM, and the dry deposition scheme is from Zhang et al. (2003). The aerosol module contains ISORROPIA and the scheme of Strader et al. (1999). Here

we concentrate on the evaluation of the standard version of IAP-AACM driven by WRF offline. The difference between the simplified and standard versions driven by WRF and the difference between the simplified version driven by IAP-AGCM (online) and WRF (offline) will be presented in our next work.

For the global simulation, the surface distribution of trace gas in the model agrees

reasonably well with site observations, mostly within a factor of two. Like other models, IAP-AACM underestimates CO over the oceans, mainly due to the underestimation of emissions over the sea. The model reproduces the spatial variation of $O_3$ well, with a range of 30-70 ppb at $30\,°$ S~$60\,°$ N. Furthermore, the model represents the seasonal variation of $O_3$ globally, with only a slightly overestimation in

summer over the land in NH. The simulation of $NO_2$ is consistent with both site records and other models. For $SO_2$, it shows a good agreement with observation



except for an overestimation in eastern America and Europe. With a weak scavenging

rate by deposition and oxidation, $SO_2$ in the model has a longer lifetime (3 days)

compared with other models (0.6-2.6 days) and the burden (0.63 Tg S $yr^{-1}$) is at the

high end of the range 0.2-0.69 Tg S $yr^{-1}$. The budgets of both carbonaceous aerosols

and sulfate are similar to those in other models. At the surface, IAP-AACM shows

very close comparison to observations for BC and OC but more variable performance

for SNA. In general, the simulation matches records on sulfate (NMB=0.36) in Asia

and on nitrate and ammonium (NMB within $\pm$0.5) in America. But it overestimates

sulfate and ammonium (NMB=1.1 and 1.49 respectively) in Europe and overestimates

only sulfate (NMB=1.94) in America. Above the surface, IAP-AACM captures the

broad seasonal and spatial features of AOD shown in the MODIS.

       For the nested simulation, IAP-AACM shows a very similar annual distribution

over EA and a more reasonable distribution on the boundary, compared with regional

models from the MICS-Asia project. IAP-AACM shows a good agreement with site

data from Chinese cities for both surface concentration and monthly variation. The

model compares well with observations of $NO_2$ and $SO_2$, especially in NC and YRD.

In most cities, IAP-AACM shows very good simulation skill for $PM_{2.5}$, not only for

the monthly variation, but also for daily variability, with R near or above 0.7. For

aerosol compositions, BC simulation shows better correlation coefficients (above 0.5)

in all four cities. The simulation of OM is lower than observations. The model results

of sulfate and ammonium in North China are close to observations, but it

underestimates in South China. As nitrate is easily decomposed at high temperatures

in the model, it is significantly underestimated in summer and autumn. The

comparison of global (1 °×1 °) and nested (0.33 °×0.33 °) results indicates that the

model reproduces the spatial variation of pollutants in the city significantly better at

fine resolution, as there are huge differences between urban and country in

meteorological field and emission.

In general, the model results for trace gases and carbonaceous aerosols show a

favorable performance. Nevertheless, the simulation of secondary aerosols shows

some weaknesses. To reduce uncertainties in the simulation of SNA, more work is

needed to improve not only aerosol chemistry but also emission inventory. Moreover,

the SOA module is relatively simple, and it could be upgraded to incorporate a

comprehensive scheme (e.g. Volatility Bias Set by Donahue et al. (2006)) and verified

with observations.

### Acknowledgments:

Thanks to Jiangsu Environmental Monitoring Center and Wuhan Environmental

Monitoring Center for their support with aerosol composition data of Nanjing and

Wuhan respectively. We sincerely thank Prof. Hajime Akimoto at National Institute

for Environmental Studies for his suggestion in improving this manuscript. We are

particularly grateful to Prof. Oliver Wild at Lancaster University for his support with

the HTAP model data and help in improving the language of this manuscript. This

research is supported by the National Major Research High Performance Computing

Program of China the National Major Research High Performance Computing

Program of China (Grant NO. 2016YFB0200800), the Chinese Ministry of Science



and Technology (Grant NO. 2017YFC0212402) and the Natural Science Foundation

of China (Grant NO. 41571130034; 91544227; 91744203; 41225019; 41705108).





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

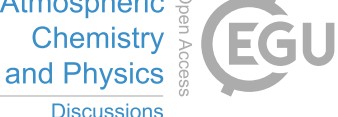


# Figures

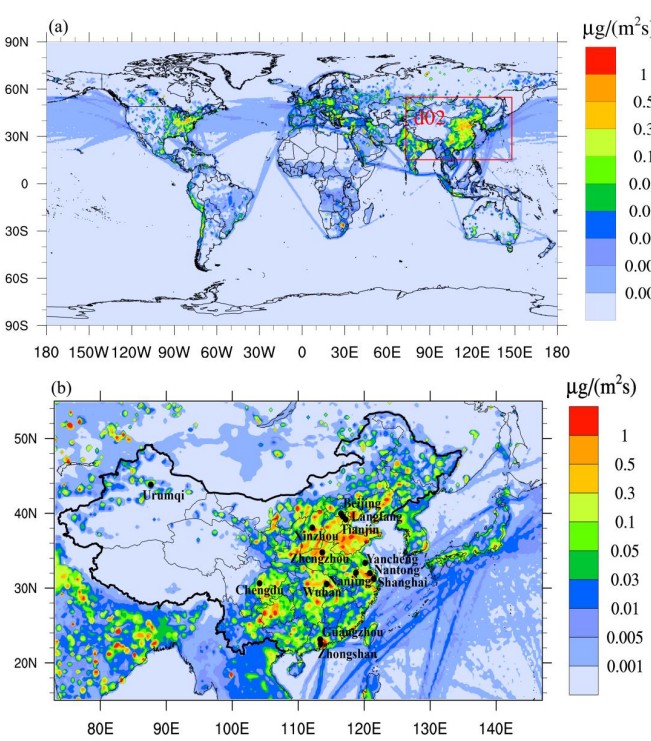


Fig. 1. The simulation domain with total SO$_2$ emission ($\mu$g m$^{-2}$ s$^{-1}$). (a) domain 1; (b) domain 2, black circles are locations of the city sites in China.



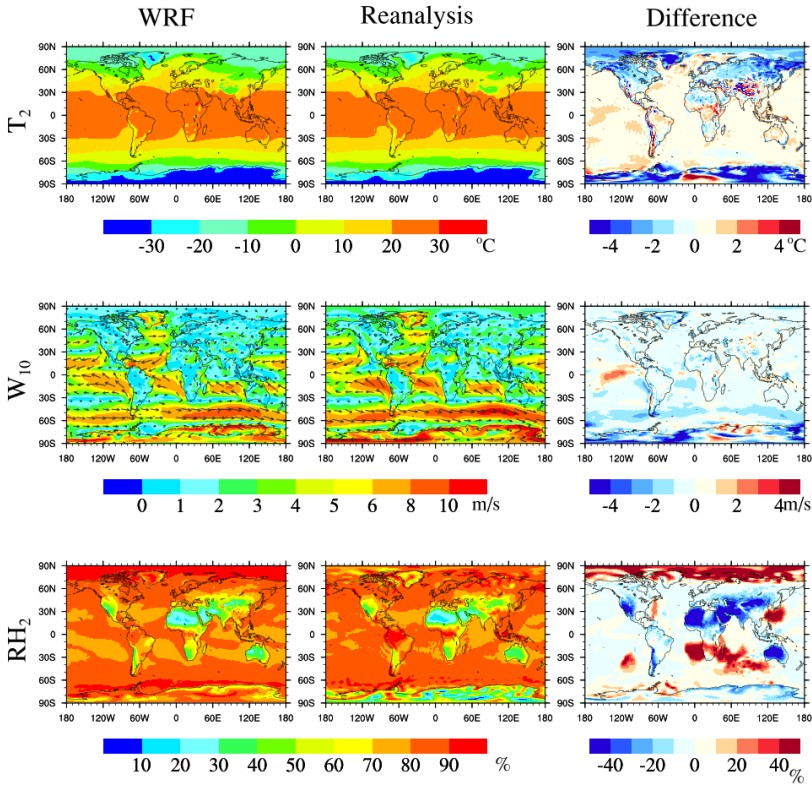

Fig. 2. Comparison of annual meteorological fields. The left column is WRF
simulation, the middle column is reanalysis data, and the right column is the
difference between simulation and reanalysis (WRF-Reanalysis). The reanalysis data
is NCEP Reanalysis1.





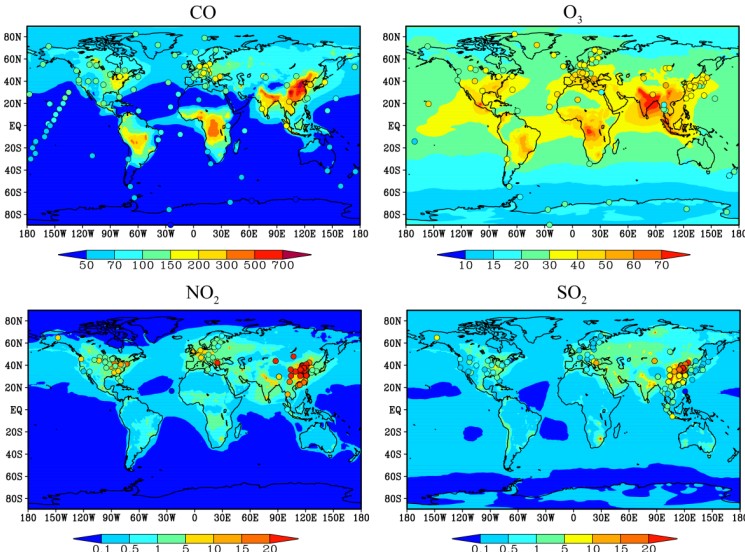

Fig. 3. Annual mean concentration (ppb) of the surface layer in IAP-AACM. The circles represent site observations. The first row is CO and $O_3$, the bottom row is $NO_2$ and $SO_2$.

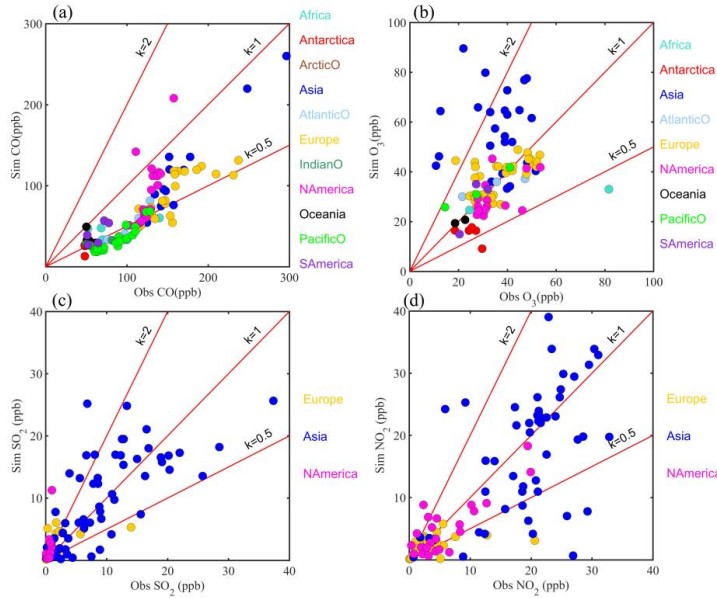

Fig. 4. Scatter plots of annual mean concentrations (ppb) in Africa, Antarctica, Arctic Ocean (ArcticO), Asia, Atlantic Ocean (AtlanticO), Europe, Indian Ocean (IndianO), North America (NAmerica), South America (SAmerica), Oceania and Pacific Ocean (PacificO). The abscissa shows the observation and the ordinate shows the simulation.





The color of the points represents different regions. (a) ~ (d) show CO, $O_3$, $SO_2$ and $NO_2$ respectively.



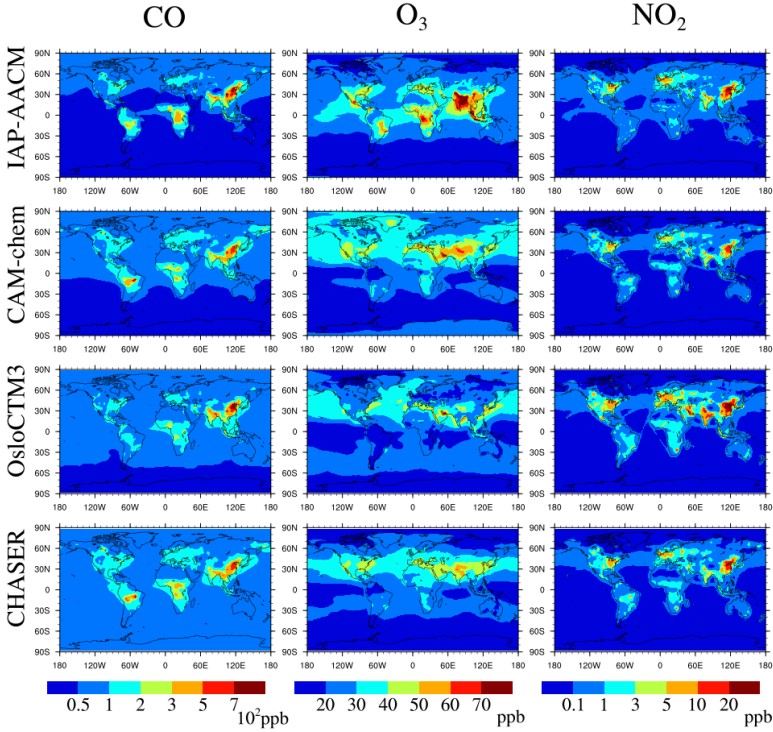


Fig. 5. Annual mean surface distributions (ppb) from IAP-AACM compared with HTAP models. Rows from top to bottom represent IAP-AACM, CAM-Chem, OsloCTM3 and CHASER respectively. The left column displays CO, the middle column displays O$_3$ and the right column is NO$_2$.

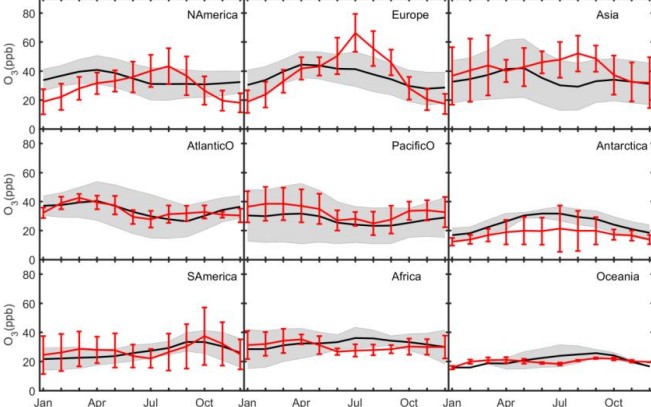


Fig. 6. Mean seasonal variation of O$_3$ (ppb) over NAmerica, Europe, Asia, AtlanticO, PacificO, Antarctica, SAmerica, Africa and Oceania sites. Black lines and red lines represent the average of observations and simulations respectively. Gray shaded areas and red vertical bars show 1 standard deviation over the sites for observations and for
model results respectively.





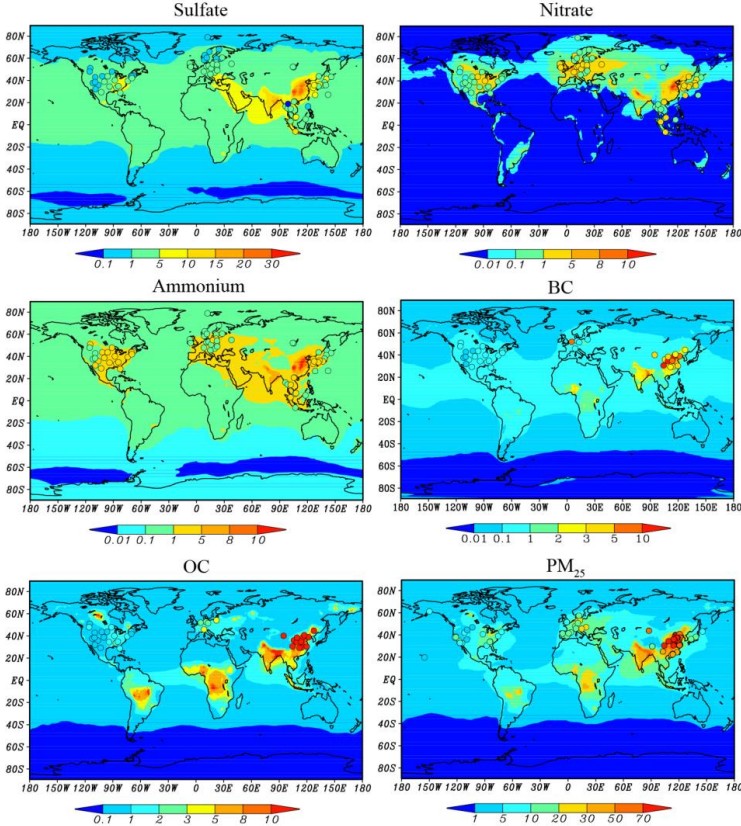

Fig. 7. The same as Fig. 3, except the species are sulfate, nitrate, ammonium, BC, OC, and $PM_{2.5}$ and the unit is µg m$^{-3}$.





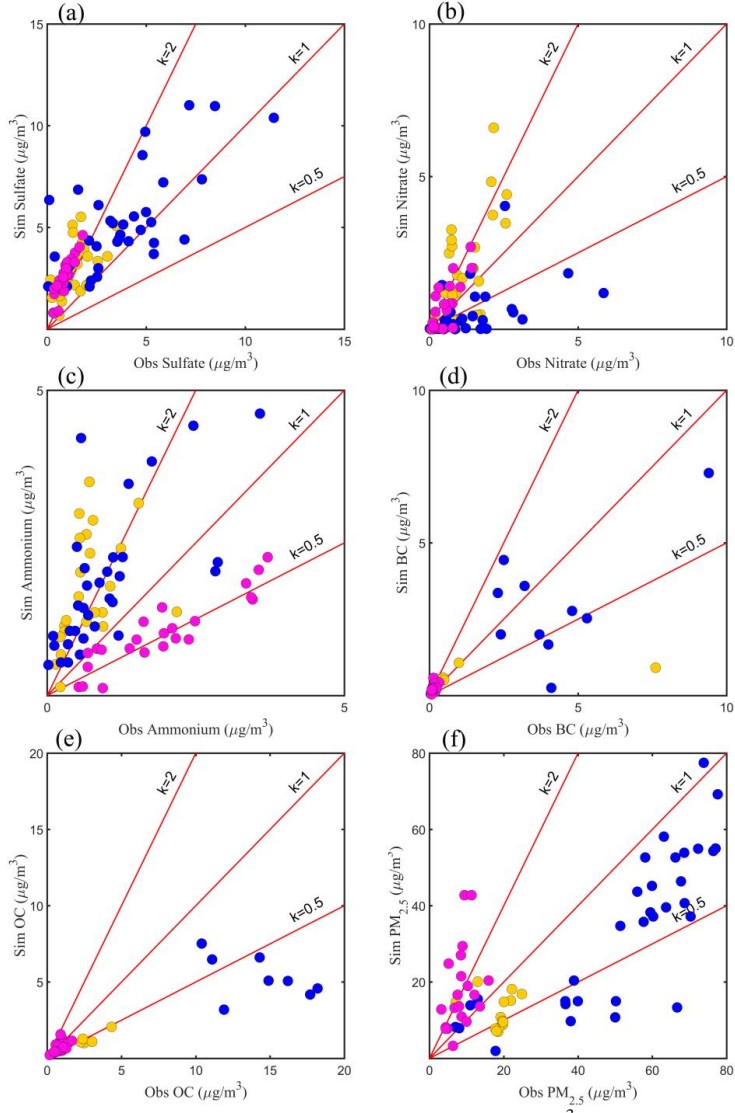


Fig. 8. Scatter plot of annual mean concentration (μg m$^{-3}$) in Europe, Asia and NAmerica. (a)~(f) is sulfate, nitrate, ammonium, BC, OC and PM$_{2.5}$ respectively. The abscissa shows the observation and the ordinate shows the simulation. The regions represented by different colors are consistent with Fig. 4c






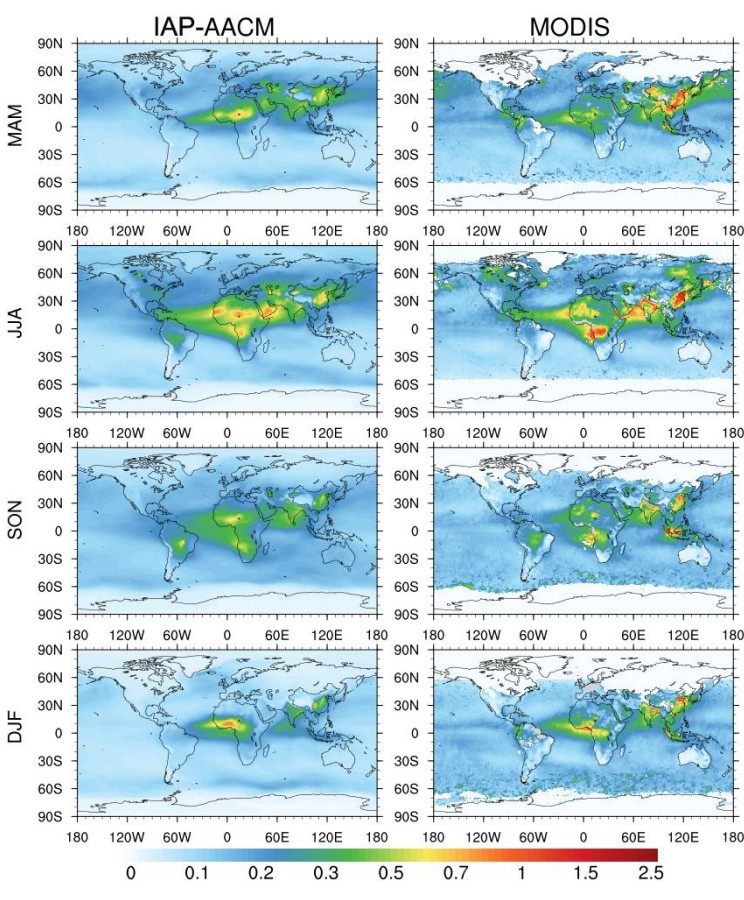

Fig. 9. Seasonal mean AOD from IAP-AACM and MODIS. Seasons are defined as December-January-February (DJF), March-April-May (MAM), June-July-August (JJA), and September-October-November (SON).





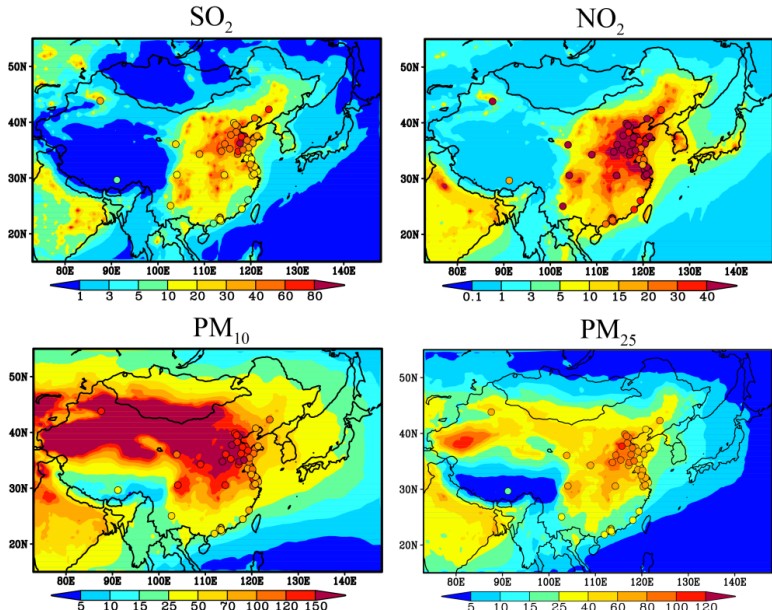

Fig. 10. Surface annual mean concentration (μg m$^{-3}$) of the nested domain. The circles represent sites observations. The top row is $SO_2$ and $NO_2$, the bottom row is $PM_{10}$ and $PM_{2.5}$.





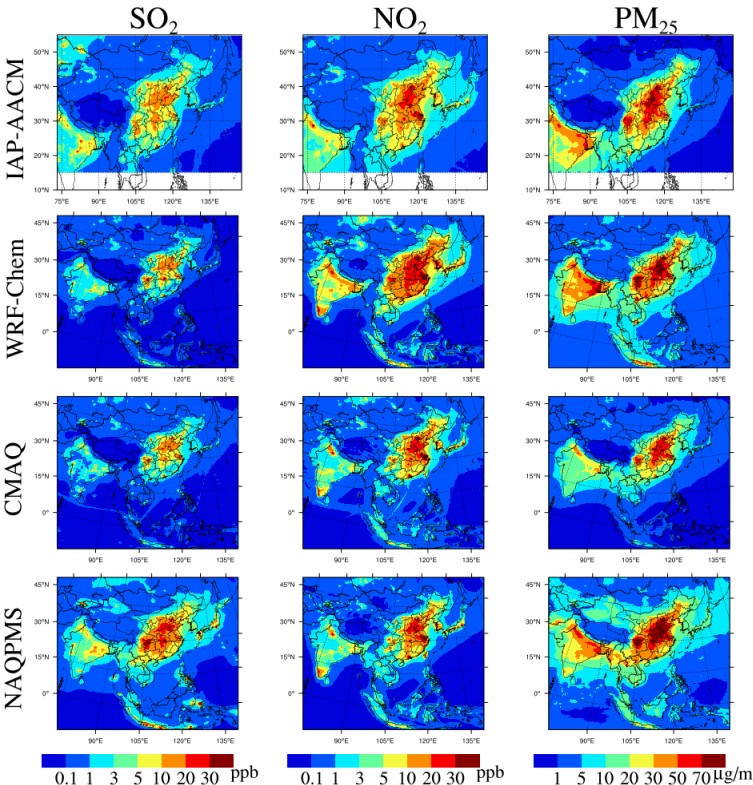

Fig. 11. Annual surface distributions from nested IAP-AACM compared with regional models from MICS-Asia. Each row from top to bottom represents IAP-AACM, WRF-Chem, CMAQ and NAQPMS respectively. The left column is $SO_2$, the middle column is $NO_2$ and the right column is $PM_{2.5}$. The unit for gases is ppb and for particles is $\mu g\ m^{-3}$.



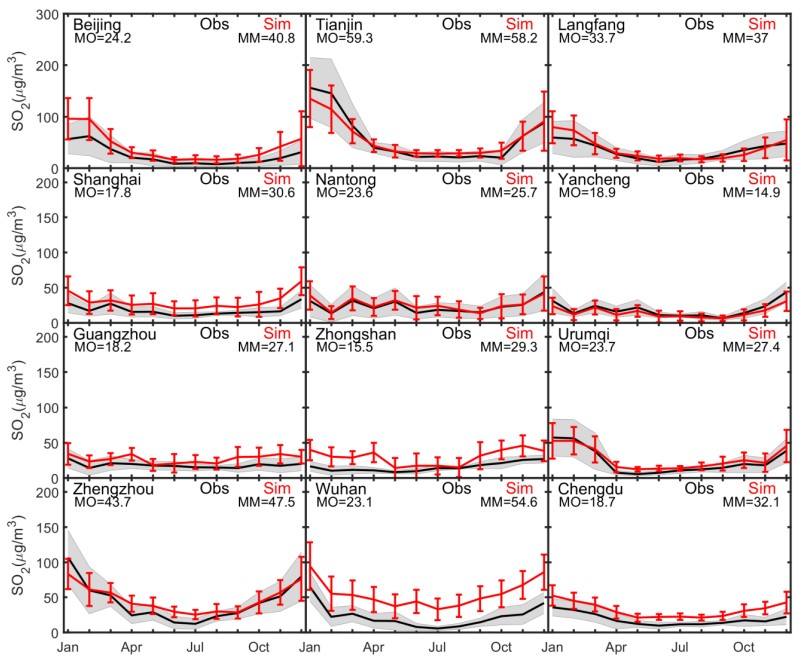


Fig. 12. Mean seasonal variation of SO$_2$ (µg m$^{-3}$) over China. The black line and red line represent monthly mean concentration of city-averaged observation and simulation respectively. Gray shaded areas and red vertical bars show 1 standard deviation over the sites for observations and for model results, respectively. MO and
MM stand for annual mean concentration of observation and simulation respectively.





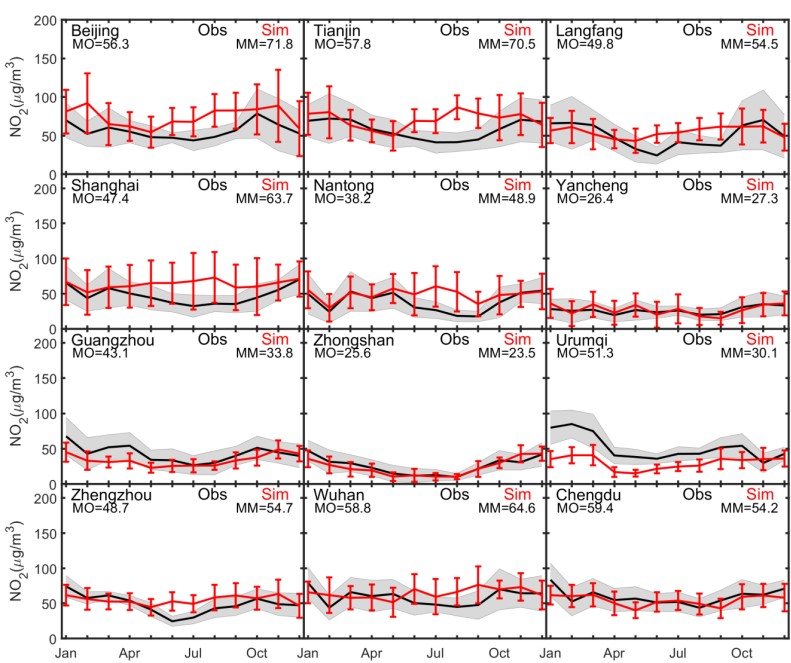

Fig. 13. The same as Fig. 12, except the pollutant is NO$_2$.

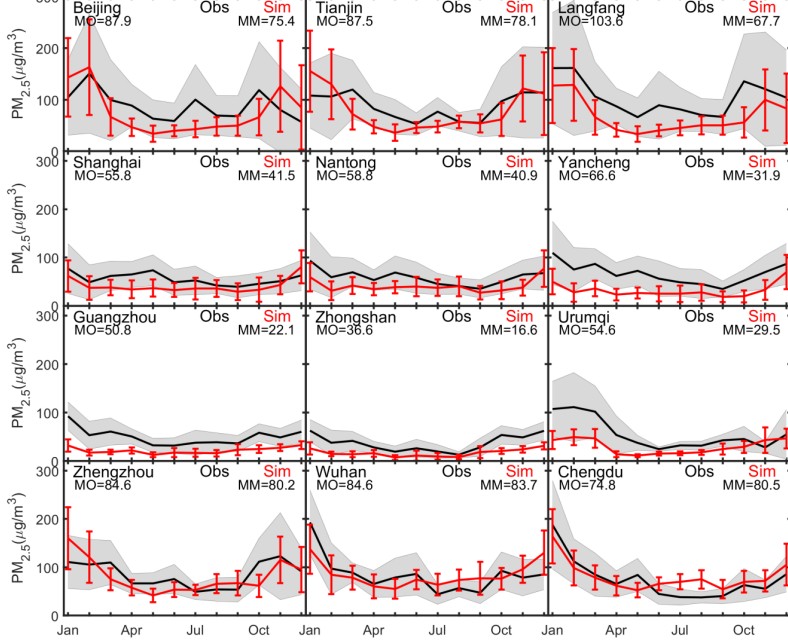

Fig. 14. The same as Fig. 12, except the pollutant is PM$_{2.5}$.

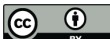



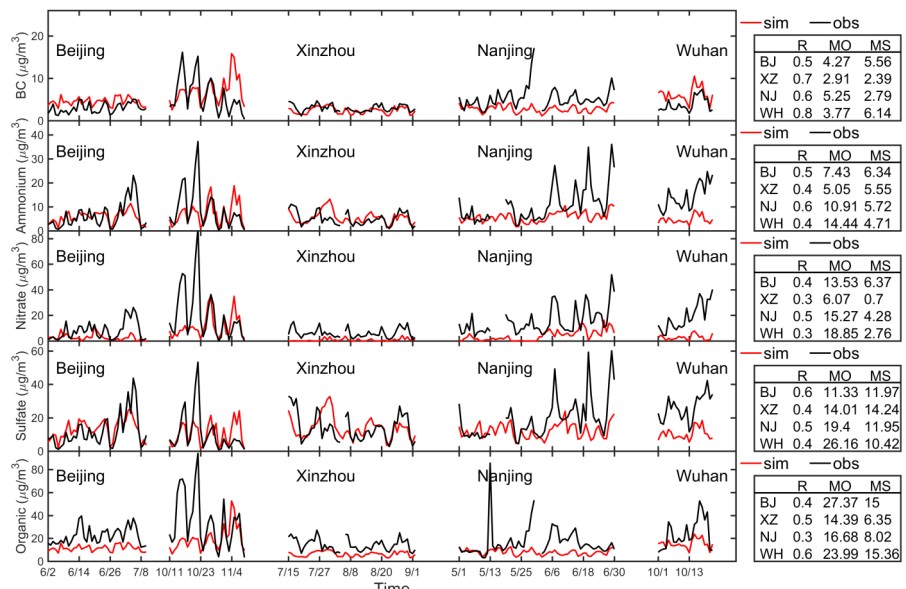

Fig. 15. Daily variation of aerosol components (µg m⁻³) over China. The black line and red line represent daily mean concentration of city-averaged observation and simulation respectively. BJ, XZ, NJ and WH mean Beijing, Xinzhou, Nanjing and Wuhan respectively. R, MO and MM stand for correlation coefficient, mean concentration of observation and model respectively.



# Tables

Table 1. Emissions used in IAP-AACM

| Database | Abbreviation | Base year | Source type | Reference |
|---|---|---|---|---|
| Hemispheric Transport of Air Pollution version2 | HTAP-v2 | 2010 | Anthropogenic | Janssens-maenhout et al., 2015 |
| Global Fire Emissions Database version4 | GFED-v4 | 2010 | Biomass burning | Randerson et al., 2015 |
| Model of Emissions of Gases and Aerosols from Nature–Monitoring Atmospheric Composition and Climate | MEGAN-MACC | 2010 | Biogenic | Sindelarova et al., 2014 |
| Regional Emission inventory in Asia | REAS | 2001 | Soil (NOx) | Yan et al., 2005 |
| Precursors of Ozone and their Effects in the Troposphere | POET | 2000 | Ocean (VOCs) | Granier et al., 2005 |
| Global Emission InitiAtive | GEIA | Average of 1983 ~ 1990 | Lightning (NOx) | Price et al., 1997 |


Table 2. Summary of statistical of annual and seasonal meteorology in the nested domain compared with NCDC sites. Seasons are defined as spring (March–May), summer (June–August), fall (September–November), and winter (December–February).

| | Period | MO | MM | RMSE | R |
|---|---|---|---|---|---|
| $T_2$ | 2014 | 17.6 | 17.5 | 1.8 | 0.98 |
| | Spring | 16.3 | 16.2 | 1.9 | 0.97 |
| | Summer | 24.3 | 24.0 | 2.0 | 0.93 |
| | Autumn | 17.2 | 17.0 | 1.7 | 0.97 |
| | Winter | 9.5 | 9.5 | 1.7 | 0.96 |
| $W_{10}$ | 2014 | 3.1 | 2.5 | 1.5 | 0.53 |
| | Spring | 3.2 | 2.7 | 1.8 | 0.61 |
| | Summer | 2.9 | 2.1 | 1.9 | 0.48 |
| | Autumn | 3.0 | 2.3 | 1.7 | 0.53 |
| | Winter | 3.1 | 2.4 | 1.8 | 0.56 |
| $RH_2$ | 2014 | 64.8 | 61.7 | 12.3 | 0.84 |
| | Spring | 58.5 | 56.2 | 12.6 | 0.86 |
| | Summer | 71.2 | 68.0 | 11.7 | 0.86 |
| | Autumn | 68.1 | 64.0 | 11.7 | 0.83 |
| | Winter | 61.4 | 58.6 | 13.2 | 0.76 |






Table 3. Summary of the site observation datasets

| Dataset | Site number | Year | Observed species |
|---------|-------------|------|------------------|
| WDCGG | 169 | Average of 2006~2015 | CO, $O_3$, $SO_2$, $NO_2$ |
| EANET | 41 | 2014 | $SO_2$, $NO_2$, $O_3$, $PM_{2.5}$, sulfate, nitrate, ammonium |
| EMEP | 46 | 2014 | $PM_{2.5}$, BC, OC, sulfate, nitrate, ammonium |
| IMPROVE | 23 | 2014 | $PM_{2.5}$, BC, OC, sulfate, nitrate, ammonium |
| EPA | 93 | 2014 | $SO_2$, $NO_2$, $PM_{2.5}$ |
| CAWNET | 13 | 2006 | BC, OC |
| CNEMC | 89 | 2014 | CO, $O_3$, $SO_2$, $NO_2$, $PM_{10}$, $PM_{2.5}$ |
| Others | 4 | 2014 | BC, OM, sulfate, nitrate, ammonium |

Table 4. Global budgets for DMS, $SO_2$ and sulfate

| Species | | IAP-AACM | Other models[a] |
|---------|---|----------|-----------------|
| DMS | Sources (Tg S $yr^{-1}$) | 22.8 | |
| | Emission | 22.8 | 10.7~23.7 |
| | Sinks (Tg S $yr^{-1}$) | 22.8 | |
| | Dry deposition | 0.0 | |
| | Oxidation | 22.8 | |
| | Burden (Tg S) | 0.19 ↑[b] | 0.02~0.15 |
| | Lifetime (days) | 3 | 0.5~3.0 |
| $SO_2$ | Sources (Tg S $yr^{-1}$) | 77.1 | |
| | Emission | 54.3 ↓[b] | 63.4~94.9 |
| | DMS oxidation | 22.8 | 10.0~25.6 |
| | Sinks (Tg S $yr^{-1}$) | 77.1 | |
| | Dry deposition | 28.0 | 16.0~55.0 |
| | Wet deposition | 0.0 | 0~19.9 |
| | Gas-phase oxidation | 19.3 | 6.1~22.0 |
| | Aqueous-phase oxidation | 29.8 | 24.5~57.8 |
| | Burden (Tg S) | 0.63 | 0.2~0.69 |
| | Lifetime (days) | 3.0 ↑[b] | 0.6~2.6 |
| Sulfate | Sources (Tg S $yr^{-1}$) | 50.5 | |
| | Emission | 1.4 | 0~3.5 |
| | Gas-phase oxidation | 19.3 | 6.1~22.0 |
| | Aqueous-phase oxidation | 29.8 | 24.5~57.8 |
| | Sinks (Tg S $yr^{-1}$) | 50.5 | |
| | Dry deposition | 2.9 | 0.8~18.0 |
| | Wet deposition | 47.6 | 34.7~61.1 |
| | Burden (Tg S) | 0.82 | 0.38~1.07 |
| | Lifetime (days) | 5.9 | 3.0~7.9 |





[a] including Liu et al. (2005), Lee et al. (2015), and those listed in Liu et al. (2005), the range
of sulfate is also refer to the GISS-TOMAS (Lee et al., 2010), ACCMIP (Lee et al., 2013) and the
AeroCom (Textor et al., 2006) results.
[b] outside the range of other models

Table 5. Global budgets for carbonaceous aerosol

| Species | | IAP-AACM | Other models[a] |
|---|---|---|---|
| BC | Sources (Tg yr$^{-1}$) | 7.42 | |
| | Emission | 7.42 | 7.4~19.0 |
| | Sinks (Tg yr$^{-1}$) | 7.42 | |
| | Dry deposition | 1.01 | 0.3~4.6 |
| | Wet deposition | 6.41 | 3.8~13.7 |
| | Burden (Tg) | 0.13 | 0.08~0.59 |
| | Lifetime (days) | 6.4 | 3.3~9.4 |
| OM[b] | Sources (Tg yr$^{-1}$) | 56.7 | 50~216 |
| | Emission | 48.7 | 34~144 |
| | Sinks (Tg yr$^{-1}$) | | |
| | Dry deposition | 6.79 | 2~36 |
| | Wet deposition | 49.9 | 28~209 |
| | Burden (Tg) | 1.16 | 0.7~3.8 |
| | Lifetime (days) | 7.4 | 3.5~9.2 |

[a] including Liu et al. (2005), Lee et al. (2010), Lee et al. (2013), Lee et al. (2015), Textor et al.
(2006), and those listed in Liu et al. (2005).
[b] the convert factor from OC to OM is 1.7 in IAP-AACM.

Table 6. The NMB of annual average concentration in different regions. ASO$_4$, ANO$_3$
and ANH$_4$ represents sulfate, nitrate and ammonium, respectively.

| | CO | O$_3$ | SO$_2$ | NO$_2$ | ASO$_4$ | ANO$_3$ | ANH$_4$ | BC | OC | PM$_{2.5}$ |
|---|---|---|---|---|---|---|---|---|---|---|
| Africa | -0.48 | -0.37 | | | | | | | | |
| Antarctica | -0.5 | -0.31 | | | | | | | | |
| ArcticO | -0.45 | | | | | | | | | |
| Asia | -0.28 | 0.86 | 0.25 | -0.59 | 0.36 | -0.61 | 0.85 | -0.4 | -0.67 | -0.36 |
| AtlanticO | -0.48 | 0.01 | | | | | | | | |
| Europe | -0.43 | 0.03 | 0.52 | -0.39 | 1.1 | 0.74 | 1.49 | -0.62 | -0.55 | -0.35 |
| IndianO | -0.54 | | | | | | | | | |
| NAmerica | -0.26 | -0.14 | 3.52 | -0.14 | 1.94 | 0.50 | -0.46 | 0.64 | -0.12 | 1.16 |
| Oceania | -0.34 | -0.03 | | | | | | | | |
| PacificO | -0.59 | 0.13 | | | | | | | | |
| SAmerica | -0.36 | 0.05 | | | | | | | | |



Table 7. Summary of statistics for global and nested domains. D1 and D2 represents
results of domain 1 and domain 2, respectively.

| Species | City | R | | RMSE (µg m⁻³) | | MB (µg m⁻³) | | NMB | |
|---|---|---|---|---|---|---|---|---|---|
| | | D1 | D2 | D1 | D2 | D1 | D2 | D1 | D2 |
| PM₂.₅ | Beijing | 0.69 | 0.70 | 54.28 | 55.65 | -12.33 | -16.89 | -0.14 | -0.19 |
| | Tianjin | 0.67 | 0.72 | 46.63 | 46.51 | -11.00 | -13.27 | -0.13 | -0.15 |
| | Langfang | 0.72 | 0.79 | 66.02 | 65.22 | -28.58 | -38.34 | -0.28 | -0.37 |
| | Shanghai | 0.71 | 0.71 | 29.51 | 27.99 | -18.23 | -16.00 | -0.33 | -0.29 |
| | Nantong | 0.69 | 0.75 | 31.46 | 29.70 | -18.32 | -17.84 | -0.31 | -0.30 |
| | Yancheng | 0.74 | 0.80 | 45.52 | 43.30 | -35.60 | -33.99 | -0.53 | -0.51 |
| | Guangzhou | 0.43 | 0.63 | 38.75 | 36.91 | -29.91 | -29.39 | -0.59 | -0.58 |
| | Zhongshan | 0.51 | 0.76 | 26.16 | 26.77 | -16.08 | -20.38 | -0.44 | -0.56 |
| | Urumqi | 0.31 | 0.50 | 59.32 | 48.10 | -38.40 | -25.88 | -0.70 | -0.47 |
| | Zhengzhou | 0.59 | 0.63 | 41.98 | 43.05 | 0.70 | -7.30 | 0.01 | -0.09 |
| | Wuhan | 0.57 | 0.64 | 44.49 | 42.28 | -11.32 | -12.09 | -0.13 | -0.14 |
| | Chengdu | 0.76 | 0.77 | 37.18 | 36.14 | 5.23 | -0.19 | 0.07 | 0.00 |
| SO₂ | Beijing | 0.87 | 0.89 | 26.99 | 25.00 | 21.32 | 16.58 | 0.88 | 0.68 |
| | Tianjin | 0.85 | 0.85 | 35.45 | 29.51 | -10.96 | -1.10 | -0.18 | -0.02 |
| | Langfang | 0.74 | 0.76 | 24.65 | 18.90 | 11.49 | 3.38 | 0.34 | 0.10 |
| | Shanghai | 0.50 | 0.75 | 38.48 | 18.10 | 30.43 | 12.76 | 1.71 | 0.72 |
| | Nantong | 0.69 | 0.78 | 13.55 | 12.08 | -0.23 | 2.17 | -0.01 | 0.09 |
| | Yancheng | 0.78 | 0.83 | 9.75 | 8.79 | -4.29 | -4.02 | -0.23 | -0.21 |
| | Guangzhou | 0.26 | 0.40 | 10.42 | 14.96 | -0.96 | 8.86 | -0.05 | 0.49 |
| | Zhongshan | 0.59 | 0.33 | 7.33 | 21.65 | 1.65 | 13.74 | 0.11 | 0.88 |
| | Urumqi | 0.63 | 0.60 | 23.04 | 20.01 | -11.88 | 3.68 | -0.50 | 0.16 |
| | Zhengzhou | 0.79 | 0.82 | 24.51 | 20.06 | 12.34 | 3.84 | 0.28 | 0.09 |
| | Wuhan | 0.70 | 0.48 | 18.72 | 40.28 | 12.03 | 31.47 | 0.52 | 1.36 |
| | Chengdu | 0.52 | 0.60 | 48.52 | 17.61 | 44.44 | 13.33 | 2.37 | 0.71 |
| NO₂ | Beijing | 0.48 | 0.68 | 26.00 | 26.82 | 11.98 | 15.68 | 0.21 | 0.28 |
| | Tianjin | 0.41 | 0.51 | 26.24 | 27.39 | 9.88 | 13.02 | 0.17 | 0.23 |
| | Langfang | 0.39 | 0.53 | 33.84 | 23.83 | 19.60 | 4.91 | 0.39 | 0.10 |
| | Shanghai | 0.57 | 0.56 | 29.28 | 32.17 | 8.79 | 16.79 | 0.19 | 0.35 |
| | Nantong | 0.60 | 0.59 | 21.86 | 24.11 | 3.63 | 10.69 | 0.10 | 0.28 |
| | Yancheng | 0.44 | 0.49 | 18.33 | 16.53 | -1.55 | 1.78 | -0.06 | 0.07 |
| | Guangzhou | 0.40 | 0.51 | 28.34 | 20.28 | -20.41 | -9.19 | -0.47 | -0.21 |
| | Zhongshan | 0.63 | 0.70 | 13.47 | 12.51 | -3.01 | -2.06 | -0.12 | -0.08 |
| | Urumqi | 0.24 | 0.41 | 41.73 | 30.31 | -35.18 | -21.39 | -0.69 | -0.42 |
| | Zhengzhou | 0.32 | 0.44 | 23.68 | 18.75 | 13.65 | 5.97 | 0.28 | 0.12 |
| | Wuhan | 0.25 | 0.22 | 25.36 | 28.39 | 5.77 | 6.16 | 0.10 | 0.10 |
| | Chengdu | 0.31 | 0.43 | 27.26 | 20.77 | -18.88 | -5.84 | -0.32 | -0.10 |


