# Peer review of "IAP-AACM v1.0: Global to regional evaluation of the atmospheric chemistry model in CAS-ESM"

_Atmospheric Chemistry and Physics, 2018_

## Referee Comment (RC1) · Anonymous Referee #3 · 24 Dec 2018

This paper presents the atmospheric chemistry component of CAS-ESM, IAP-AACM, and compares the offline model results (driven by WRF) with various observational data worldwide. This is an important step towards improving the Earth system simulations by CAS/IAP, a key participant of IPCC assessments. Below are a few suggestions to improve the paper.

The model evaluation focuses on comparisons with measurements of surface concentrations of pollutants, particularly aerosol pollutants. Because this model is developed primarily for climate studies, evaluation of the tropospheric chemistry (in addition to surface air quality) will be very important. Specifically, it would be very useful to include/expand the evaluation of vertical profiles and tropospheric burdens against observations. There are many satellite data for ozone, NO2, SO2 and HCHO, and many

[Figure]

vertical profile data (e.g., ATOM) for gaseous/aerosol species. Other important measures of tropospheric chemistry that can be discussed include the mean OH concentration and budgets, ozone budgets, methane lifetime, and MCF lifetime.

Measurement data often contain missing values and outliers and have different temporal resolutions from model simulations. Please specify how the measurement data are processed and how model results are sampled (temporally and spatially) according to measurements. In particular, satellite data contain large amounts of missing values. Near-surface NO2 measurements are contaminated by other nitrogen species, and what would be the implications for model evaluation (especially when discussing the model bias).

The resolution dependence discussed in Sect. 3.4 has also been studied in other recent works. It would be nice to refer to or compare against previous findings.

The spin-up time (one month) is too short for CO, ozone and other longer-lived species. This may explain part of the underestimate in CO. Please comment on the effect of spinup time.

There have been discussions in the literature on bug fixes in ISOROPIA II. Are these bugs and fixes relevant here?

Brief descriptions of WRFv3.3 would be very useful. The vertical resolution of WRF is different from that in IAP-AACM, so how is the conversion done?

Table 1 – do you extrapolate the emissions to 2014? If not, what would be implications for your model evaluation against measurements in 2014?

In the comparisons over China, only a few cities are selected, although there are CEMC measurements in other cities as well. Please explain the rationale for choosing these cities.

Specific comments:

Abstract – please specify which part of the writing is for the evaluation of global model and which is for nested model. Also, please present the bias (in addition to R) of the model.

L48-67 – the references are relatively old. Please use newer ones. Also, aerosols affect the cardiovascular diseases very significantly.

L71 – change "prediction" to "projection"

L87-88 – there have been model evaluation studies over China in recent years. Please refer to these studies.

L97 – remove "precise". Every model has its limitations.

L100 – change to "lateral (and upper) boundary conditions"

L147 – specify the resolution

L160 – do you mean "natural dust"?

L199 – do you mean the first layer center is 50 m?

Table 2 – please explain the meanings of these statistics and provide the units.

L279 – why not just use the WDCGG data in 2014?

CO model evaluation – could you comment on the effect of spinup time and coarse model resolution?

L416 – the ozone seasonality is not very well captured in many regions. Also, this paragraph is too long.

L452 – please specify the quantitative difference between GFED3 and GFED4.

L485-499 – please comment on the effect of difference in time (2006 for measurements and 2014 for model simulation).

L495 – BC depends on emissions and deposition processes.

L500 – please clarify which components are included in PM2.5

L506 – please specify the version of MODIS AOD and how data are selected/sampled.

L512 – LAC or BC?

L522-531 – please consider to present the seasonality results in a line figure.

L526 – In the model, DJF is not the season with the highest AOD over East China.

L548-558 – please be more quantitative.

L567-568 – please provide model versions.

L572 – do you mean "other regional models"?

L575-576 – what are the differences in emissions?

Fig. 14 – please specify the components in PM2.5

Table 6 – please specify which one is global model and which one is regional model. Also, please provide the mean values over these cities.

---

## Referee Comment (RC2) · Anonymous Referee #2 · 24 Dec 2018

General Comment

The authors tried to introduce their newly developed global and regional CTM (IAP-AACM) which can be used as the chemical module of their ESM. In this paper, they explained the ability of the IAP-AACM to properly simulate the spatial and temporal variation in the concentration of major atmospheric gas species and aerosols in 2014. The simulated concentrations were compared with various observations globally and in particular with those obtained in China, which showed a comparable representation level of their model to the other global and regional CTMs. I think the purpose of this paper is more suitable for other journal such as Geosci. Model Dev., but the paper can be also within the scope of ACP. I will leave the judgement to editor which journal is suitable for this paper. In any case, however, I noticed several issues in this paper

which cannot be passed over to be published in any journal. I suggested that the authors should consider the following comments.

Major Comment:

The main purpose of this paper is to clearly show the ability and/or inability of their model to simulate the observed spatial and temporal variation in the concentration of the chemical species in the atmosphere. Based on those findings the authors and also the readers of the paper can understand what kind applications are suitable for this model and what aspect of the model should be further improved in order to apply it to a particular issue. From this point of view, the self-evaluation about the ability of the model by authors were often insufficient and unclear. The good points and also the shortcomings of the model should be described more specifically in the text. I pointed out some of those points as specific comments in the following, but I strongly recommend the authors to reexamine the descriptions particularly in the model evaluation parts.

Specific Comments:

- L24: What are the aerosol effects here?

- L38: Only R-value can not ensure the accuracy of the simulation. How about MB or NMB?

-L58: Why didn't you cite the latest AR5 report here?

- L79: Typo? e.g.

- L86: EA, this should be defined at its first appearance in the text.

- L108-109: What do you mean here? Could you use more words to explain "localization of the process parameterization"?

- L135-139: Are there citable references for CoLM, CICE, and IAP-OBGCM?

- L158: What is the main difference between these two models (GNAQPMS and IAP-AACM)?

- L204: What does "synchronous time step" mean ?

- L206: What is the reason for choosing the year 2014 as the focal year?

- L224-225: Emission data used in the study are not up-to-date, the base year of each database is a bit old. Therefore, adjusting the emission data to input them to the model is suitable for the purpose of this study. However, you only mentioned about the adjustment of SO2 emission in China in the text. Did you adjust other species emission ?

- Figure2 : Why did you compare with NCEP R1, not with NCEP-FNL? What is the purpose of it?

- L244: This statement is not correct. The difference in RH2 between WRF and Re-analysis is much larger in general as shown in Fig2 over land area.

- Table2 and L246-248: If you want to mention only the correlation coefficient of annual mean values, you should remove Table 2. If you want to retain Table2, you should explain the table more precisely here. Table 2 is hard to read and insufficient caption.

- L270: Why did you take average of 2006-2015 only for WDCGG?

- Table3: It is better to include the information of region of each observation datasets.

- L298: The value (23.3) differs from that in Table4.

- Table5: Table 5 is not completely filled with the necessary information for OM (other sources and total sink are missing).

- Figure4: Fig4c and 4d should be switched to be in accordance with the order of panels in Fig3.

- L363: Typo?: Northern Hemisphere

- L368-369: Why did IAP-AACM show the lowest concentration of CO over ocean among the models considered here?.

- L378-380: The seasonal variation of surface O3 should be different in different environment even in the same region. So, I recommend the authors to compare separately for different environment (e.g. maritime area vs mountainous area). Otherwise, I can not regard the Fig6 as an evidence that the model can well simulate the seasonal variation of surface O3.

- L385: Underestimation in Antarctica is not small. Such an underestimation could be seen in the other CTMs. Can you use more words about this issue here?

- L385-387: In the NH land area, it seems that the model completely failed to represent the seasonal cycle of surface O3, but the author regarded it as just a positive bias during July-September. More words for this issue are necessary. For example, what do you think the apparent underestimation in cold season in NAmerica and Europe?

- L388-390: In Badia et al (2017), they suspected the excessive emission height of NOx which will cause low NOx at surface and consequently might lead to weak NO titration. Do the same things happen in your model?

- L390-391: The AACM apparently showed larger concentration of surface O3 in the tropical regions (central Africa, South America, and Southeast Asia) than the other models. However, the concentration of O3 precursor species (CO and NOx) in these regions are not so different among the models. Can you give discussion about the issue here?

- L391-394: These two sentences are not consistent to each other. In general, the region of high O3 concentration can be different in different season. If you look at the "annual mean" concentration, the highest O3 usually occur in the source region in summer, but that in the downwind region in winter. However, if you see the different index such as MD8H O3, you can see completely different seasonal cycle. I strongly

recommend the author to carefully revise these sentences.

- L402-403: An overall evaluation of O3 dry deposition in global CTMs can be seen in Hardecre et al. (2015). I recommend to check it out. Hardacre et al. (2015) An evaluation of ozone dry deposition in global scale chemistry climate model, Atmos. Chem. Phys., 15, 6419–6436, doi:10.5194/acp-15-6419-2015.

- L416-418: The concentration of NOx over oceanic areas are larger in AACM than in other models, which might stem from larger emission or longer life time of NOx in AACM than the other models. I recommend to discuss this issue further here.

- L436-438: This is misleading statement. The model results are not generally with in a factor of two, but they apparently tend to overestimate the observation in all the three regions. The NMB value for sulfate in Europe, 0.11, is incorrect which is 1.1 in Table 6.

- L438-439: How can you conclude like this (2ugm-3 higher)? What is the ground of this statement?

- L442-443: What aspect of the observation do you think your model can reproduce? You should be more specific.

- L446-449: About the simulation of ammonium, I can see obvious underestimation in NAmerica and overestimation in Asia and Europe.

- L455-457: The concentration of OC were obviously underestimated by the model.

- L491-492: The highest AOD in DJF in east China is not clearly seen both in satellite and model AOD.

- Figure10: It's better to show scatter plots too, at least as a supplement figure.

- Figure11: The area and the map projection of the figures for all models should be united.

- L517-519: I'm sorry I can not understand what you want to mean here.

- L545-547: What do you want to mean here? Your model overestimated the NO2 in summer in NC and YRD regions. If you don't use the NO2* observation, the model's overestimation should become worse.

- L547-550: I can not understand what aspect of seasonal difference in NO2 column observation were reproduced by your model. You should describe more specifically on it.

- L595: Typo? respects → aspects

- Conclusions should be revised according to the modifications made to respond the reviewers comments.
* * *

---

## Author Comment (AC1) · 5 Mar 2019

**The authors appreciate the reviewers very much for reviewing our manuscript and providing constructive comments. As suggested, we carefully revised the manuscript thoroughly according to the valuable advices, as well as proof-read the manuscript to minimize typographical, grammatical, and bibliographical errors. Our replies to the comments and our actions taken to revise the paper (in blue) are given below (the original comments are copied here). The figures added in the reply is represented by 'Figure', which is distinguished from 'Fig.' in the manuscript.**

**General comments:**

1. This paper presents the atmospheric chemistry component of CAS-ESM, IAP-AACM, and compares the offline model results (driven by WRF) with various observational data worldwide. This is an important step towards improving the Earth system simulations by CAS/IAP, a key participant of IPCC assessments. Below are a few suggestions to improve the paper.

   The model evaluation focuses on comparisons with measurements of surface concentrations of pollutants, particularly aerosol pollutants. Because this model is developed primarily for climate studies, evaluation of the tropospheric chemistry (in addition to surface air quality) will be very important. Specifically, It would be very useful to include/expand the evaluation of vertical profiles and tropospheric burdens against observations. There are many satellite data for ozone, NO2, SO2 and HCHO, and many vertical profile data (e.g., ATOM) for gaseous/aerosol species. Other important measures of tropospheric chemistry that can be discussed include the mean OH concentration and budgets, ozone budgets, methane lifetime, and MCF lifetime.

   Reply: It is a good suggestion to include vertical comparison to improve the model evaluation work. We evaluated the tropospheric column concentration of NO2 and O3 with satellite data (GOME2A and OMI) and discussed the profile concentration of OH with other models, in the light of reviewer's comments. The budget of ozone and CO are also elaluated in the manuscript.

   Table 1 the budget of O3 and CO compared with the other models.

| Species | Process | | IAP-AACM |
|---|---|---|---|
| CO | Emission (Tg yr$^{-1}$) Total 994 | Anthrop. | 546.4 |
| | | Bio. burning | 336.2 |
| | | Biogenic | 92.7 |
| | | Others | 18.3 |
| | Top condition inflow (Tg yr$^{-1}$) | | 28 |

| | | |
|---|---|---|
| | Chem pro (Tg yr$^{-1}$) | 1270 |
| | Chem lss (Tg yr$^{-1}$) | 2292 |
| | Dry dep (Tg yr$^{-1}$) | 0 |
| | Burden (Tg) | 327 |
| | Lifetime (days) | 52 |
| | Top condition inflow (Tg yr$^{-1}$) | 473 |
| | Chemical production (Tg yr$^{-1}$) | 3940 |
| | Chemical loss (Tg yr$^{-1}$) | 3564 |
| $O_3$ | Dry dep. (Tg yr$^{-1}$) | 849 |
| | Burden (Tg) | 370 |
| | Lifetime (days) | 30.6 |

O3: The vertical tropospheric column (VTC) of $O_3$ is compared against satellite observation derived from OMI (shown in Figure 1). In the main board, the pattern of the seasonal cycle was covered by the model. In mainland of Northern Hemisphere, the higher $O_3$ VTC appears during June-July-August (JJA), while in Northern Hemisphere, it appears during September-October-November (SON), with a range of 40-60 DU. The model still keeps a high value (40-50 DU) in tropics during DJF, possibly due to the high concentration of CO emit from biomass burning. The $O_3$ VTC is significantly underestimated over ocean in middle-high latitudes, and the reasons need to be further studied.

[Figure]

Figure 1 Seasonal mean column concentration of $O_3$ in IAP-AACM (left column) and OMI (right column). Seasons are defined as December-January-February (DJF), March-April-May (MAM), June-July-August (JJA), and September-October-November (SON). The unit is DU.

NO2: The VTC of $NO_2$ is also compared against satellite observation derived from GOME2A (shown in Figure 2). The $NO_2$ VTC has a range of 20-150 $\times 10^{14}$ molecule cm$^{-2}$ in most source areas. By and large, IAP-AACM reproduced the magnitude in different regions. In addition, the model captured seasonal variations of $NO_2$ concentration in the vertical troposphere well. In anthropogenic source areas of Northern Hemisphere (e.g., North America, Europe, East Asia), the $NO_2$ VTC is higher in SON and December-January-February (DJF) while lower in JJA, caused by seasonal human activities such as fuel heating. The column concentration in South America and South Africa is higher during JJA, while it is higher in central Africa during DJF, due to the vegetation burning in dry season. Compared with GOME2A, IAP-AACM showed a larger column concentration over ocean. The overestimation is also reflected in the comparison of surface concentration. This is probably caused by insufficient oxidation to nitrate and a higher injection height in the emission which leads to a farther transportation distance. Generally, the distribution of $NO_2$ by the model is consistent with satellite observation, except some source areas (e.g., underestimation in Australia and South America, overestimation in East Asia), which suggests a bias of emission inventory.

[Figure]

Figure 2 Seasonal mean column concentration of $NO_2$ in IAP-AACM (left column) and GOME-2A (right column). Seasons are defined as December-January-February (DJF), March-April-May (MAM), June-July-August (JJA), and September-October-November (SON). The unit is $10^{14}$ molecule cm$^{-2}$.

OH: Oxidation is the basic characteristic of atmospheric chemistry. As the most important oxidant in atmosphere, hydroxyl radical (OH) is one of the crucial species to simulate the general properties in CTMs. OH formation in troposphere is mainly due to $O_3$ photolysis with the reaction $O_3 + h\nu$ ($\lambda \leqq 320nm$) + $H2O \rightarrow 2OH+O2$. The tropospheric (200hpa to the surface) mean OH concentration of IAP-AACM is $13.0 \times 10^5$ molec cm$^{-3}$. It is a little higher than the mean OH concentration study ($11.1 \pm 1.6 \times 10^5$ molec cm$^{-3}$) from 16 ACCMIP models for 2000 by Naik et al. (2013). It potentially leads to strong atmospheric oxidation. The lower concentration of CO over oceans may be related to it. The zonal mean OH concentrations for January, April, July and October are shown in Figure 3. Like other chemistry models, OH concentration in the tropics keeps highest all the year round and decreases gradually from tropics to poles. This is due to the positive influence of solar radiation and water vapor concentration. The seasonal north-south oscillation of OH maximum area is also ascribed to the seasonal variation of these two factors. The mean OH inter-hemispheric (N/S) ratio of the model is 1.26, in accordance with the present-day multi-model mean ratio ($1.28 \pm 0.1$) for 2000 (Naik et al., 2013). Vertically, the highest concentration is in the layer of 2-4 km above the tropics. In Northern Hemisphere, the highest OH concentration appears in

summer. Peak value of OH in July is located at around 30 °N, in the sky above 2km. Generally, the range of OH concentration is similar with other models (e.g., TM5 (Huijnen et al., 2010), NMMB-MONARCH (Badia et al., 2017)), except a slightly higher peak concentration of 30-35 molec cm⁻³, compared with the other models above-mentioned (under 30 molec cm⁻³).

[Figure]

Figure 3 Zonal monthly mean concentration of OH for January, April, July and October by the IAP-AACM. The unit is $10^5$ molecule cm⁻³.

2. Measurement data often contain missing values and outliers and have different temporal resolutions from model simulations. Please specify how the measurement data are processed and how model results are sampled (temporally and spatially) according to measurements. In particular, satellite data contain large amounts of missing values. Near-surface NO2 measurements are contaminated by other nitrogen species, and what would be the implications for model evaluation (especially when discussing the model bias).

Reply: The measurement datasets (except CNEMC) collected in this paper are monthly or annual results which have been processed by the observation workgroups. The hourly CNEMC observations are processed by data quality control. The corresponding simulation data compared with aforementioned observations are sampled at the same locations and altitudes as observations, with the model grid cells containing the observational sites. The simulation of seasonal cycle in different regions or cities are first sampled at the model grid cells containing the observational sites and then averaged within sub-regions. When compared with satellite data, the missing values of satellite data are kept and shown in the figures.

As shown in Figure 12 in the manuscript, model results for $NO_2$ concentration are a bit underestimated (NMB= -0.63). As the "$NO_2$" values reported by routine monitoring sites are $NO_2^*$, which partially includes $HNO_3$ and $NO_3^-$, it is common to underestimate the observed "$NO_2$". Thus the model's overestimation should become worse. It reflects the shortcoming of multiphase processes in IAP-AACM. The overestimation of $NO_2$ and underestimation of nitrate in daytime of summer and autumn is related to the over decomposition of nitric acid at high temperature condition in the thermodynamic equilibrium module. Moreover, heterogeneous chemical reactions in the model should partly be responsible for the $NO_2$ overestimation in summer. Reactive heterogeneous uptake of gases may be crucial for the formation of secondary aerosols when the other oxidants (e.g. ozone, OH) are in low concentrations level (Jacob, 2000; Martin et al., 2003). The heterogeneous chemical module coupled in IAP-AACM has been tested in North China in winter (Li et al, 2018). The uptake of $SO_2$ by wet aerosols significantly enhanced sulfate formation under highly polluted conditions, contributing 50%-80% of total concentration of sulfate. The mechanism also reduced the overestimation of nitrate which is also appeared in other models. However, when it comes to the problem here, we checked the simulations excluded heterogeneous chemical processes and found a better performance of $NO_2$ in summer (shown in Figure 4). It implicates that a more comprehensive mechanism should be considered in model development.

[Figure]

Figure 4 Seasonal cycle of $NO_2$ ($\mu$g m$^{-3}$) simulated without heterogeneous chemical process over China. The black line and red line represent monthly mean concentration of city-averaged observation and simulation respectively. Gray shaded areas and red vertical bars show 1 standard deviation over the sites

for observations and for model results, respectively. MO and MM stand for annual mean concentration of observation and simulation respectively.

3. The resolution dependence discussed in Sect. 3.4 has also been studied in other recent works. It would be nice to refer to or compare against previous findings.

Reply: That's a good suggestion. High-resolution helps to improve CTMs performance, but it is limited by the scale applicable to the parameterization scheme of physical and chemical processes. Recently, sensitivity to horizontal grid resolution has been discussed in many regional model works. Wang et al. (2014) showed a better simulation of particles in North China with CMAQ when increasing the resolution from 36km to 12km. A study of PM2.5 heath impact assessment with CMAQ by Jiang et al. (2018) found that model results at 12 km generally performed better and had substantially lower computational burden, compared to 4 km resolution. As a global nested model, we also want to evaluate the improvements or not due to higher horizontal resolution.

4. The spin-up time (one month) is too short for CO, ozone and other longer-lived species. This may explain part of the underestimate in CO. Please comment on the effect of spin-up time.

Reply: We agree that the spin-up time of one month is not enough for longer-lived species. It may lead to an underestimation of some trace gases such as CO. But in this study we used monthly mean concentration of CO, $O_3$ and $NO_2$ from MOZART-4 as the top boundary condition. It can offset the potential underestimation of CO and $O_3$ substantially. Furthermore, to verify the effect of shorter spin-up time here, we also run a case with spin-up time of one year. The annual mean result is very similar to the case of one month spin-up time as shown in Figure 5.

The underestimation of CO potentially reflects a difference in emissions. The natural sources of CO over ocean are included in the HTAP models whereas they are not considered in IAP-AACM. Besides, it may reflect differences in chemical transformation between models. As shown in Figure 3, the OH concentration is a bit higher in IAP-AACM than the other models. Due to the sink reaction of CO (CO + OH → $CO_2$ + H), the CO loss will be faster in IAP-AACM.

[Figure]

Figure 5 The annual mean surface concentration of CO. The left one is the surface distribution with one month spin-up time, the right one is with one year spin-up time.

5. There have been discussions in the literature on bug fixes in ISOROPIA II. Are these bugs and fixes relevant here?

Reply: No, it's not relevant here. The code bug only affects the forward (in which the concentration of both gas and aerosol of each species is fixed) stable state calculation. In IAP-AACM, we use reverse mode (in which the concentration of each species in the aerosol phase is fixed) to calculate.

6. Brief descriptions of WRFv3.3 would be very useful. The vertical resolution of WRF is different from that in IAP-AACM, so how is the conversion done?

Reply: The WRF version used in this study is a global version of WRFv3.3 (GWRF). It is an extension of mesoscale WRF that was developed for global weather research and forecasting applications. GWRF has more general choice of map projection (to include both conformal and nonconformal map projections). It includes specific boundary conditions and can be run as a traditional C-grid GCM by filtering. The specification of planetary constants, physics parameterizations and timing conventions are also improved to allow the model to be run as a global model. Thus, it has multiscale and nesting capabilities, blurring the distinction between global and mesoscale models and enabling investigation of coupling between processes on all scales. The model has been applied to simulation at various scales to Mars, and at global scales to Titan and Venus (Richardson et al., 2007).

Output of WRF is interpolated to keep accordance with IAP-AACM vertically. The information has been added in the revised manuscript.

7. Table 1 – do you extrapolate the emissions to 2014? If not, what would be implications for your model evaluation against measurements in 2014?

Reply: Yes,we extrapolate the emission of $SO_2$ to 2014. As a consequence of government control policy included in the twelfth Five-Year Plan (FYP), China has achieved an obvious decrease in air pollution in the past years, especially for $SO_2$. The FYP controls suppress $SO_2$ emissions in energy and industry sectors which is the major source of $SO_2$. Considering the cutting effect on $SO_2$ (China completed the emission reduction task of 12th FYP (2010~2015) ahead of schedule in 2014 with a reduction ratio reaching by 12.9%),we adjusted the total $SO_2$ emission for 2014 by a factor of 0.9 in China. For other species, the intensity of emission reduction is not so great like $SO_2$. The study by Zheng et al. (2018) showed that the dramatic reduction of emissions is mostly happened after 2013 for China's Clean Air Action implemented during 2013-2017. Relative change rates of China's anthropogenic emissions during 2010–2017 are estimated as follows: -62% for SO2, -17% for NOx, -27% for CO, -27% for BC and -35% for OC. And the emission mostly decreased during 2013-2017, by 59% for SO2, 21% for NOx, 23% for

CO, 28% for BC and 32% for OC. Compared to 2010, emissions of trace gas in 2014 decreased not significant except $SO_2$ (shown in Figure 6). So we only extrapolate the emission of $SO_2$. It will partly be responsible for the underestimation of some species (e.g., $NO_2$ in Fig. 13) in our simulation.

[Figure]

Figure 6 Emission trends and underlying social and economic factors from 2010 to 2017 by Zheng et al. (2018).

8.  In the comparisons over China, only a few cities are selected, although there are CEMC measurements in other cities as well. Please explain the rationale for choosing these cities.

    Reply: The cities selected are divided into six regions (North China, Pearl River Delta, Yangtze River Delta, Northwest China, Central China, Southwest China). The six regions not only represent the major geographical regions over China, but also include the regions with the most severe air pollution at present which means the focus regions of research.

**Specific comments:**

1.  Abstract – please specify which part of the writing is for the evaluation of global model and which is for nested model. Also, please present the bias (in addition to R) of the model.

    Reply: Some words have been added to the abstract to specify global and nested evaluation (see below). Also, normal mean biases are supplemented, too.

    For global simulation, the 1-year simulation for 2014 shows that the IAP-AACM is within the range of other models, and well reproduces both spatial distribution and seasonal variation of trace gases and aerosols over major continents and oceans (mostly within the factor of two). The model well captures spatial variation for carbon monoxide but with a bit underestimation (normal mean bias (NMB) of -0.59~-0.23) especially over the ocean that also shown in other models, which suggests the need for more accurate emission rate of ocean source. For aerosols, the simulation of fine-mode particulate matter (PM2.5) matches observation well and it has a better simulating ability on primary aerosols (NMB are

within ±0.67) than secondary aerosols (NMB are greater than 1.0 in some regions). This calls for more investigation on aerosol chemistry. Furthermore, for nested regional simulation, IAP-AACM shows the superiority of global model, compared with regional model, on performing regional transportation for the nested simulation over East Asia. With regard to the city evaluation over China, the model reproduces variation of sulfur dioxide (SO2), nitrogen dioxide (NO2) and PM2.5 accurately in most cities, with correlation coefficients (R) above 0.5 and NMB within ±0.5.

2. L48-67 – the references are relatively old. Please use newer ones. Also, aerosols affect the cardiovascular diseases very significantly.

Reply: The citation of IPCC has been updated to the latest report. References for aerosols' health effect are also updated (see below).

Aerosols formatted from these precursor gases, together with aerosols from other sources, have a direct radiative forcing. By modifying cloud properties, the aerosols also have important indirect effects. As reported in the Fifth Assessment Report (AR5) of IPCC (Myhre et al., 2013), the radiative forcing of aerosols ranges from -1.9 ~ -0.1 W m-2, with the direct radiative forcing ranges from -0.85 ~ 0.15 W m-2. With better model performance and more robust observation network, AR5 achieved increasing confidence in the assessment compared with AR4 (Boucher et al., 2013), but the largest uncertainty to the total radiative forcing estimate is still aerosols. In addition, aerosols have adverse impacts on human health including respiratory diseases, cardiovascular risk and lung cancer, which has drawn increasing public attention (Burnett et al., 2014; Pope et al., 2011; Powell et al., 2015).

3. L71 – change "prediction" to "projection"

Reply: It has been corrected.

4. L87-88 – there have been model evaluation studies over China in recent years. Please refer to these studies.

Reply: Yes, there have been several model evaluation studies with observation in China. The description in the introduction has been updated.

5. L97 – remove "precise". Every model has its limitations.

Reply: It has been modified in the revised manuscript.

6. L100 – change to "lateral (and upper) boundary conditions"

Reply: It has been modified in the revised manuscript.

7. L147 – specify the resolution

Reply: The high resolution is 0.25 °×0.25 °, we have specified it in the revised manuscript.

8. L160 – do you mean "natural dust"?

Reply: Yes,it is.

9. L199 – do you mean the first layer center is 50 m?

Reply: Yes, we have specified the meaning.

10. Table 2 – please explain the meanings of these statistics and provide the units.

Reply: Captions and units are added in the revised manuscript.

11. L279 – why not just use the WDCGG data in 2014?

Reply: The dataset of WDCGG provides a large number of trace gases observations globally. But some sites are without invalid records in 2014. To get more data to evaluate the model over the world, we expanded the time range to ten years (2006-2015).

We have re-selected the observation data for 2014 to comparison with model results. Overall, the results have not changed much in terms of the evaluation of model's simulation capability. The simulation bias is reduced in some regions while it is increased in some other regions. The simulation of $NO_2$ performs better with the NMB of Asia and Europe closer to zero. The underestimation of CO in Antarctica disappeared due to the change of the observed value. There are some changes in the trend of the seasonal variation of $O_3$ in Northern Hemisphere. All the figures (as shown in Figure 7~ Figure 9) and tables related to these changes are updated in the manuscript, and the corresponding analysis is updated in the manuscript, too.

[Figure]

Figure 7 Annual mean concentration (ppb) of the surface layer in IAP-AACM. The circles represent site observations. The first row is CO and O₃, the bottom row is NO₂ and SO₂.

[Figure]

Figure 8 Scatter plots of annual mean concentrations (ppb) in Africa, Antarctica, Arctic Ocean (ArcticO),
Asia, Atlantic Ocean (AtlanticO), Europe, Indian Ocean (IndianO), North America (NAmerica), South

America (SAmerica), Oceania and Pacific Ocean (PacificO). The abscissa shows the observation and the ordinate shows the simulation. The color of the points represents different regions. (a) ~ (d) show CO, O$_3$, NO$_2$ and SO$_2$ respectively.

[Figure]

Figure 9 Mean seasonal variation of O$_3$ (ppb) over NAmerica, Europe, Asia, AtlanticO, PacificO, Antarctica, SAmerica, Africa and Oceania sites. Black lines and red lines represent the average of observations and simulations respectively. Gray shaded areas and red vertical bars show 1 standard deviation over the sites for observations and for model results respectively.

12. CO model evaluation – could you comment on the effect of spinup time and coarse model resolution?

Reply: We agree that the spin-up time of one month is not enough for longer-lived species. It may lead to an underestimation of some trace gases such as CO and O$_3$. But in this study we used monthly mean concentration of CO, O$_3$ and NO$_2$ from MOZART-4 as the top boundary condition. It can offset potential underestimation of CO and O$_3$ substantially. Furthermore, to verify the effect of the shorter spin-up time here, we also run a case with spin-up time of one year. The annual mean result is almost the same with the case of one month spin-up time as shown in Figure 5.

On one hand, the results of coarse-resolution models are often lower than those of high-resolution models due to the effect of gridded average on static emission sources. On the other hand, it's difficult to reproduce the atmospheric dynamics characteristics under complex underlying surface conditions for coarse resolution models. The coarse resolution of global models cannot represent local orographically driven flows or sharp gradients in mixing depths. It's unfavorable to simulate pollutant diffusion process. Furthermore, the inputs of meteorological fields with larger grids are also poorly represented.

13. L416 – the ozone seasonality is not very well captured in many regions. Also, this paragraph is too long.

Reply: Agree. We have reanalyzed the simulation of ozone in this part in the revised manuscript. The model showed poorly performance on the seasonal cycle of surface ozone in the NH land, with overestimation in Europe and EA in summer while underestimation in winter in NH land, as shown in Figure 9 (the comparison drawn with WDCGG observation only for 2014).

The surface $O_3$ are also underestimated in spring over NH land. In IAP-AACM, the stratospheric-tropospheric exchange and corresponding photochemistry are not considered. It will lead to a large negative bias in the simulating. To date it has become apparent that the measured annual cycle of ozone shows a distinct maximum during spring. The stratosphere-to-troposphere ozone transport event occurs widely across mid-latitudes in the NH (Monks et al., 2000; Akritidis et al., 2018). Since the magnitude and frequency of the transport through tropopause is still not clear. There are large uncertainties in simulating the flux. Some researches (Munzert et al, 1985; Austin and Follows, 1991) showed that the maximum in the stratosphere to troposphere flux occurs in late winter/spring. It may partly responsible for the underestimation of $O_3$ in winter, too.

The surface $O_3$ concentrations over East Asia (sites mainly located in Japan) are overestimated in summer and early autumn. The same pattern is also found in the multi-model inter-comparison of 21 HTAP models (Fiore et al., 2009). The simulations in island countries of EA are sensitive to the timing and extent of the Asian summer monsoon (Han et al., 2008). The positive model bias in this season may stem from inadequate representation of southwesterly inflow of clean marine air.

14. L452 – please specify the quantitative difference between GFED3 and GFED4.

Reply: GFED3 and GFED4 are both monthly burned area emission data gridded to 0.5 °×0.5 ° and 0.25 °×0.25 °, respectively. Due to the impact of a reduction of combustion area and decreasing in fuel consumption, there is about a 20%~30% reduction of CO emissions in GFED4 compared to GFED3 in the tropical regions (central Africa, South America, and Southeast Asia) (Werf et al., 2017). The $NO_2$ emission of GFED4 may be also decreased due to the reduction of burned area. This specific difference has been added to the revised manuscript.

15. L485-499 – please comment on the effect of difference in time (2006 for measurements and 2014 for model simulation).

Reply: As the simulation used emissions of 2010 but the measurements are for 2006, there is a mismatch on emission scenario. There is a bit increasing (less than 0.1Tg) of BC and OC emissions from 2006 to 2010 in China (Lu et al., 2011; Fu et al., 2012). Besides, the meteorological conditions also play a role.

As the analysis of the CAWNET observation over China (Zhang et al., 2015), there is no significant changes happened in the proportion of chemical component of $PM_{10}$ from 2006 to 2013. For the annual average trends of carbonaceous shown in Figure 10, both Southwest China and North China experienced a

process of declining first and then rising due to the unfavorable weather conditions. Pearl River Delta showed a significant falling (about half). Yangtze River Delta had a slight decreasing. Generally, it is reasonable to infer that the distribution of BC and OC in most areas have changed a little from 2006 to 2014, except for the Pearl River Delta region.

[Figure]

YRD      PRD

SWC      NC

Figure 10 Monthly mean concentrations of OC and EC from 2006 to 2013 by Zhang et al., 2015. YRD, PRD, SWC and NC represents Yangtze River Delta, Pearl River Delta, Southwest China and North China, respectively.

16. L495 – BC depends on emissions and deposition processes.

Reply: Yes, it has been complemented.

17. L500 – please clarify which components are included in PM2.5

Reply: The components of $PM_{2.5}$ in Fig. 10 includes primary $PM_{2.5}$, BC, OC, SNA, SOA and also natural dust, this is supplemented in the revised manuscript.

18. L506 – please specify the version of MODIS AOD and how data are selected/sampled.

Reply: The product version is MYD04_L2-MODIS/Aqua Aerosol 5-Min L2 Swath 10km. It is available at the website: http://dx.doi.org/10.5067/MODIS/MYD04_L2.006. The product version and website is supplemented in the revised manuscript.

19. L512 – LAC or BC?

Reply: it should be BC here, it has been revised.

20. L522-531 – please consider to present the seasonality results in a line figure.

Reply: That's a good suggestion. A more detailed comparison of the global gridded average AOD on the seasonality variation is displayed in Figure 11. As the seasonality cycle is different in different regions, we not only showed the global average value, but also showed the gridded average value of Africa, South America and East Asia, which are major aerosol emission areas. Generally, the model captured seasonal variation in different regions, but there is a gap in the value between observation and simulation. The discrepancy in East Asia potentially stemmed from the inaccurate simulation of dust activities in spring, which is mainly due to the simulation of meteorological field (e.g., wind, precipitation).

[Figure]

Figure 11 Gridded mean value of monthly averaged AOD for 2014, AF, EA, SA and GL represents Africa, East Asia, South America and global. Dash line and solid line represents model results and observation derived from MODIS, respectively.

21. L526 – In the model, DJF is not the season with the highest AOD over East China.

Reply: Yes, it's an incorrect expression here and we have deleted it. In fact, the highest AOD may not be in DJF, it often appears in MAM. This phenomenon is common in other model evaluation studies (e.g., GISS-TOMAS (Lee et al., 2010)). On one hand, China is frequently affected by dust in spring. On the other hand, AOD is an optical characteristic of aerosols for the whole vertical layer. It is not equivalent to aerosol mass concentration.

22. L548-558 – please be more quantitative.

Reply: To be more quantitative, we provided scatter plots of simulations in the nested domain in Figure 12. As shown in Figure 12, model results for $SO_2$, $PM_{10}$ and $PM_{2.5}$ are mostly within the factor of two with NMB within ±0.52, while $NO_2$ concentration are a bit underestimated (NMB= -0.63).

[Figure]

Figure 12 Scatter plots of annual mean concentrations (μg m$^{-3}$) in nested domain. (a)~(f) is SO$_2$, NO$_2$, PM$_{10}$ and PM$_{2.5}$ respectively. The abscissa shows the observation and the ordinate shows the simulation.

23. L567-568 – please provide model versions.

Reply: the model versions are CMAQv4.7.1, WRF-Chemv3.9 respectively. This has been added in the revised manuscript.

24. L572 – do you mean "other regional models"?

Reply: It means the regional model. Here we compared the simulation of the nested domain in IAP-AACM with regional models of MICS-Asia.

25. L575-576 – what are the differences in emissions?

Reply: The differences of emissions between IAP-AACM and MICS-Asia models are natural sources. For anthropogenic source, IAP-AACM uses MIX inventory (incorporated into HTAP for Asia) as same as MICS-Asia models. For biogenic source, IAP-AACM uses MEGAN-MACC but models of MICS-Asia uses an earlier version of MEGANv2.04. For biomass burning source, IAP-AACM uses GFEDv4 but MICS-Asia models uses GFEDv3.

26. Fig. 14 – please specify the components in PM2.5

Reply: The components of PM$_{2.5}$ in Fig. 14 includes primary PM$_{2.5}$, BC, OC, SNA and SOA, this is supplement in the caption.

27. Table 6 – please specify which one is global model and which one is regional model. Also, please provide the mean values over these cities.

Reply: Do you mean Table 7 in the ACPD document, the statistics for 12 cities in global and nested domains? If so, all the results in this table are calculated with outputs from the global model IAP-AACM. The difference between D1 and D2 is the horizontal resolution. D1 represents domain 1 (1°×1°), D2 represents domain 2 (0.33°×0.33°). The statistics over these cities are supplemented in Table 7.

**References**

Akritidis, D., Katragkou, E., Zanis, P.,et al. A deep stratosphere-to-troposphere ozone transport event over Europe simulated in CAMS global and regional forecast systems: analysis and evaluation. Atmospheric Chemistry and Physics, 18, 15515–15534, doi: 10.5194/acp-18-15515-2018, 2018.

Austin, J. F., Follows, M. J. The ozone record at Payerne: an assessment of the cross-tropopause flux. Atmospheric Envirnoment 25A, 1873-1880, 1991.

Boucher, O., D. Randall, P. Artaxo, C. Bretherton, G. Feingold, P. Forster, V.-M. Kerminen, Y. Kondo, H. Liao, U. Lohmann, P. Rasch, S.K. Satheesh, S. Sherwood, B. Stevens and X.Y. Zhang, 2013: Clouds and Aerosols. In: Climate Change 2013: The Physical Science Basis. Contribution of Working Group I to the Fifth Assessment Report of the Intergovernmental Panel on Climate Change [Stocker, T.F., D. Qin, G.-K. Plattner, M. Tignor, S.K. Allen, J. Boschung, A. Nauels, Y. Xia, V. Bex and P.M. Midgley (eds.)]. Cambridge University Press, Cambridge, United Kingdom and New York, NY, USA.

Burnett, R. T., Arden Pope, C., Ezzati, M., Olives, C., Lim, S. S., Mehta, S., Shin, H. H., Singh, G., Hubbell, B., Brauer, M., Ross, Anderson, H., Smith, K. R., Balmes, J. R., Bruce, N. G., Kan, H., Laden, F., Prüss-Ustün, A., Turner, M. C., Gapstur, S. M., Diver, W. R., and Cohen, A: An integrated risk function for estimating the global burden of disease attributable to ambient fine particulate matter exposure, Environ. Health Persp., 122, 397, https://doi.org/10.1289/ehp.1307049, 2014.

Dai, Y. J., Zeng, X. B., Dickinson, R. E. , Baker, I. ,Bonan, G. B. & Bosilovich, M. G.: The common land model. Bulletin of the American Meteorological Society, 84(8), 1013-1023, 2015. doi:10.1175/BAMS-84-8-1013, 2015.

Falk, S., & Sinnhuber, B. M. Polar boundary layer bromine explosion and ozone depletion events in the chemistry-climate model EMAC v2.52: implementation and evaluation of airsnow algorithm. Geoscientific Model Development, 11(3), 1-15, https://doi.org/10.5194/gmd-11-1115-2018, 2018.

Fiore,A. M., Dentener,F. J., Wild, O., et al. Multimodel estimates of intercontinental source-receptor relationships for ozone pollution. JOURNAL OF GEOPHYSICAL RESEARCH, 114, D04301, doi:10.1029/2008JD010816, 2009, 2009.

Fu, T.-M., Cao, J. J., Zhang, X. Y., Lee, S. C., & Henze, D. K.: Carbonaceous aerosols in china: top-down constraints on primary sources and estimation of secondary contribution. Atmospheric Chemistry and Physics, 12(5), 2725-2746, doi:10.5194/acp-12-2725-2012, 2012.

Galmarini, S., Koffi, B., Solazzo, E., Keating, T., Hogrefe, C., & Schulz, M., et al. Technical note:

coordination and harmonization of the multi-scale, multi-model activities HTAP2, AQMEII3, and MICS-Asia3: simulations, emission inventories, boundary conditions, and model output formats. Atmospheric Chemistry and Physics Discuss, 17(2), 1543-1555, 2017.

Han, Z., et al. MICS-Asia II: Model intercomparison and evaluation of ozone and relevant species, Atmos. Environ., 42, 3491 – 3509, doi:10.1016/j.atmosenv.2007.07.031, 2008.

Han, Z., Xie, Z., Wang, G., Zhang, R., & Tao, J. Modeling organic aerosols over east china using a volatility basis-set approach with aging mechanism in a regional air quality model. Atmospheric Environment,124, 186-198, 2016.

Huijnen, V., Williams, J., van Weele, M., van Noije, T., Krol, M., Dentener, F., Segers, A., Houweling, S., Peters, W., de Laat, J., Boersma, F., Bergamaschi, P., van Velthoven, P., Le Sager, P., Eskes, H., Alkemade, F., Scheele, R., Nédélec, P., and Pätz, H.-W.: The global chemistry transport model TM5: description and evaluation of the tropospheric chemistry version 3.0, Geosci. Model Dev., 3, 445–473, doi:10.5194/gmd-3-445-2010, 2010.

Jacob, D. J. . Heterogeneous chemistry and tropospheric ozone. Atmos. Environ. 34, 2131–2159, 2000.

Jiang, X. , & Yoo, E. H. The importance of spatial resolutions of community multiscale air quality (CMAQ) models on health impact assessment. Science of The Total Environment, 627, 1528-1543, doi: 10.1016/j.scitotenv.2018.01.228, 2018.

Lee, Y. H., & Adams, P. J. : Evaluation of aerosol distributions in the GISS-TOMAS global aerosol microphysics model with remote sensing observations. Atmospheric Chemistry & Physics, 10(5), 2129-2144, 2010.

Li, J., Chen, X., Wang, Z., Du, H., Yang, W., & Sun, Y., et al. : Radiative and heterogeneous chemical effects of aerosols on ozone and inorganic aerosols over East Asia, Science of the Total Environment, 622, 1327-1342, https://doi.org/10.1016/j.scitotenv.2017.12.041, 2018.

Li, Y., & Xu, Y..: Uptake and storage of anthropogenic $CO_2$ in the pacific ocean estimated using two modeling approaches. Advances in Atmospheric Sciences, 29(4), 795-809, 2012. doi: 10.1007/s00376-012-1170-4, 2012.

Jian, L., An, J., Yu, Q., Yong, C., Ying, L., & Tang, Y., et al. Local and distant source contributions to secondary organic aerosol in the Beijing urban area in summer. Atmospheric Environment, 124, 176-185, 2016.

Lu, Z, Zhang, Q, & Streets, D. G.: Sulfur dioxide and primary carbonaceous aerosol emissions in China and India, 1996–2010. Atmospheric Chemistry and Physics, 11(18), 9839-9864, 2011.

Martin, R.V., Jacob, D.J., Yantosca, R.M. . Global and regional decreases in tropospheric oxidants from photochemical effects of aerosols. J. Geophys. Res. 108 (D3):4097. https://doi.org/10.1029/2002JD002622, 2003.

Miyazaki, K. , & Bowman, K. Evaluation of ACCMIP ozone simulations and ozonesonde sampling biases using a satellite-based multi-constituent chemical reanalysis. Atmospheric Chemistry and Physics, 17(13), 8285-8312, 2017.

Monks, P. S. A review of the observations and origins of the spring ozone maximum. Atmospheric

Environment, 34(21), 3545-3561, 2000.

Munzert, K., Reiter, R., Kanter, H.-J., Potzl, K. Effect of stratospheric intrusions on the tropospheric ozone. In Proceedings of the Quad. Ozone Symposium, Halkidiki, Reidel, Dordecht, pp. 735-739, 1985.

Myhre, G., D. Shindell, F.-M. Bréon, W. Collins, J. Fuglestvedt, J. Huang, D. Koch, J.-F. Lamarque, D. Lee, B. Mendoza, T. Nakajima, A. Robock, G. Stephens, T. Takemura and H. Zhang, 2013: Anthropogenic and Natural Radiative Forcing. In: Climate Change 2013: The Physical Science Basis. Contribution of Working Group I to the Fifth Assessment Report of the Intergovernmental Panel on Climate Change [Stocker, T.F., D. Qin, G.-K. Plattner, M. Tignor, S.K. Allen, J. Boschung, A. Nauels, Y. Xia, V. Bex and P.M. Midgley (eds.)]. Cambridge University Press, Cambridge, United Kingdom and New York, NY, USA.

Richardson, M. I., Toigo, A. D. and Newman, C. E: Planet WRF: A General Purpose, Local to Global Numerical Model for Planetary Atmosphere and Climate Dynamics, Journal of Geophysical Research, 112, E09001, doi:10.1029/2006JE002825, 2007.

Naik, V., Voulgarakis, A., Fiore, A. M., Horowitz, L. W., Lamarque, J.-F., Lin, M., Prather, M. J., Young, P. J., Bergmann, D., Cameron-Smith, P. J., Cionni, I., Collins, W. J., Dalsøren, S. B., Doherty, R., Eyring, V., Faluvegi, G., Folberth, G. A., Josse, B., Lee, Y. H., MacKenzie, I. A., Nagashima, T., van Noije, T. P. C., Plummer, D. A., Righi, M., Rumbold, S. T., Skeie, R., Shindell, D. T., Stevenson, D. S., Strode, S., Sudo, K., Szopa, S., and Zeng, G.: Preindustrial to present-day changes in tropospheric hydroxyl radical and methane lifetime from the Atmospheric Chemistry and Climate Model Intercomparison Project (ACCMIP), Atmos. Chem. Phys., 13, 5277–5298, doi:10.5194/acp-13-5277-2013, 2013.

Pope III, C. A., Burnett, R. T., Turner, M. C., Cohen, A., Krewski, D., Jerrett, M. et al.: Lung cancer and cardiovascular disease mortality associated with ambient air pollution and cigarette smoke: shape of the exposure–response relationships, Environ. Health Persp., 119, 1616, https://doi.org/10.1289/ehp.1103639, 2011.

Powell, H., Krall, J. R., Wang, Y., Bell, M. L., and Peng, R. D.: Ambient coarse particulate matter and hospital admissions in the Medicare Cohort Air Pollution Study, 1999–2010, Environ. Health Persp., 123, 1152, https://doi.org/10.1289/ehp.1408720, 2015.

Simpson, W. R., von Glasow, R., Riedel, K., Anderson, P., Ariya, P., Bottenheim, J., Burrows, J., Carpenter, L. J., Frieß, U., Goodsite, M. E., Heard, D., Hutterli, M., Jacobi, H.-W., Kaleschke, L., Neff, B., Plane, J., Platt, U., Richter, A., Roscoe, H., Sander, R., Shepson, P., Sodeau, J., Steffen, A., Wagner, T., and Wolff, E.: Halogens and their role in polar boundary-layer ozone depletion, Atmos. Chem. Phys., 7, 4375–4418, doi:10.5194/acp-7- 4375-2007, 2007.

Xu, R. T., Tian, H. Q., Pan, S. F., Prior, S. A., Feng, Y. C., & Batchelor, W. D., et al.: Global ammonia emissions from synthetic nitrogen fertilizer applications in agricultural systems: Empirical and process-based estimates and uncertainty. Global Change Biology, 25, 314-325, doi: 10.1111/gcb.14499, 2019.

Wang, L. T., Jang, C., Zhang, Y., Wang, K., Zhang, Q., & Streets, D., et al.: Assessment of air quality benefits from national air pollution control policies in china. Part I: background, emission scenarios and evaluation of meteorological predictions. Atmospheric Environment, 44(28), 3449-3457, 2010.

Wang, L. T., Wei, Z., Yang, J., Zhang, Y., Zhang, F. F., & Su, J., et al. The 2013 severe haze over southern Hebei, China: model evaluation, source apportionment, and policy implications. Atmospheric Chemistry and

Physics, 14(11), 28395-28451, doi:10.5194/acp-14-3151-2014, 2014.

Werf, G. R. V. D., Randerson, J. T. , Giglio, L. , Leeuwen, T. T. V. , Chen, Y. , & Rogers, B. M. , et al. Global fire emissions estimates during 1997–2016. Earth System Science Data, 9(2), 697-720. https://doi.org/10.5194/essd-9-697-2017, 2017.

---

## Author Comment (AC2) · 5 Mar 2019

**The authors appreciate the reviewers very much for reviewing our manuscript and providing constructive comments. As suggested, we carefully revised the manuscript thoroughly according to the valuable advices, as well as proof-read the manuscript to minimize typographical, grammatical, and bibliographical errors. Our replies to the comments and our actions taken to revise the paper (in blue) are given below (the original comments are copied here). The figures added in the reply is represented by 'Figure', which is distinguished from 'Fig.' in the manuscript.**

**Major comments:**

The main purpose of this paper is to clearly show the ability and/or inability of their model to simulate the observed spatial and temporal variation in the concentration of the chemical species in the atmosphere. Based on those findings the authors and also the readers of the paper can understand what kind applications are suitable for this model and what aspect of the model should be further improved in order to apply it to a particular issue. From this point of view, the self-evaluation about the ability of the model by authors were often insufficient and unclear. The good points and also the shortcomings of the model should be described more specifically in the text. I pointed out some of those points as specific comments in the following, but I strongly recommend the authors to reexamine the descriptions particularly in the model evaluation parts.

Reply: We greatly appreciate the reviewer for insight comments on the manuscript. To respond to the reviewer's major concerns, we made thoroughly revisions and corrections according to all the insight comments of the reviewers. Besides, more crucial information and analysis will be added in the revised manuscript, as follows:

(1) A new evaluation with the WDCGG datasets only for 2014 is updated, and the evaluation is more quantitative.

(2) We provide more information to discuss the model's performance on the underestimation of CO over ocean, including a comparison of the profile concentration of OH with other models.

(3) More studies and descriptions focused on the simulation of ozone. The bias of inter-models and model-observation are discussed. In particular, the poor performance on the seasonal cycle in NH land is interpreted. We further showed the seasonal cycle of

ozone compared against sites separated by the terrain.

(4) The model's simulating ability on aerosol formation is discussed in detail. Especially, the SOA formation mechanism and the multiphase processes in the model are described.

(5) Overall, more analysis on the model's performance are shown. The discussion includes the uncertainty of emissions, the limitation of global model resolution, the superiority of the model, the shortcomings of chemical scheme, the impact of meteorology and deposition. On the basis of these comparison and analysis, some suggests are put forward in the model's further improvement.

**Specific comments:**

- L24: What are the aerosol effects here?

Reply: The aerosol effects refer to climate effect (direct, semi-direct and indirect effect) and health effect (mainly of respiratory diseases, cardiovascular risk and lung cancer).

- L38: Only R-value can not ensure the accuracy of the simulation. How about MB or NMB?

Reply: The NMB for the nested simulation are within ±0.5. It is supplemented in the abstract.

-L58: Why didn't you cite the latest AR5 report here?

Reply: Good suggestion, we have updated the citation to the latest report.

- L79: Typo? e.g.

Reply: 'For example' is used to introduce the work by Badia et al. (2017), Mann et al. (2010) and Tsigaridis et al. (2014) instead of 'e.g.' in the manuscript.

- L86: EA, this should be defined at its first appearance in the text.

Reply: Thanks, the definition has been added at its first appearance in the revised manuscript.

- L108-109: What do you mean here? Could you use more words to explain "localization of the process parameterization"?

Reply: In the dust module, the deflation mechanism and dust loading parameterization are based on a detailed analysis of the meteorological conditions, landform, and climatology from daily weather records at about 300 local stations in north China. For the heterogeneous chemistry scheme, the parameterization of uptake coefficients considered the meteorological condition of relative humidity in China. It has been added in the manuscript.

- L135-139: Are there citable references for CoLM, , and IAP-OBGCM?

Reply: the references for CoLM (Dai et al., 2003), and IAP-OBGCM (Li et al., 2012) has been supplemented in the revised manuscript.

-L158: What is the main difference between these two models (GNAQPMS and IAP-AACM)?

Reply: Generally, IAP-AACM is similar to GNAQPMS. IAP-AACM has the same model framework with GNAQPMS but has some improvements. The model was renamed when it joined the CAS-ESM.

- L204: What does "synchronous time step" mean ?

Reply: It is the time step of model's integration calculation. In order to keep the stability of calculation in the model, the integration time step will be cut into shorter sub-integration time step in different modules (e.g., advection and gas chemistry processes). So the synchronous time step means the model's integration calculation.

- L206: What is the reason for choosing the year 2014 as the focal year?

Reply: Choosing the year 2014 is a compromise between little change in emissions and more observation data published in China. The year 2014 is near now that there are more comprehensive observation data of both trace gases and particles to obtain over China. Chinese national environmental monitoring network (CNEMC) started to publish data since 2013. The site records of aerosols for 2014 are also available in China. Besides, China has implemented strict air pollution control measures since 2013. Emission sources haven't change much from 2010 to 2014, except $SO_2$. Zheng et al. (2018).

- L224-225: Emission data used in the study are not up-to-date, the base year of each database is a bit old. Therefore, adjusting the emission data to input them to the model is suitable for the purpose of this study. However, you only mentioned about the adjustment of SO2 emission in China in the text. Did you adjust other species emission?

Reply: No, we only adjust the emission of $SO_2$ in China for its dramatic variation in the past years. During 2010~2014, the fluctuation of emissions are not severe globally except China, due to a strict controlling policy known to all. The study by Zheng et al. (2018) shows that relative change rates of China's anthropogenic emissions during 2010–2017 are estimated as follows: -62% for SO2, -17% for NOx, -27% for CO, -27% for BC and -35% for OC. But the emissions decreased by 59% for SO2, 21% for NOx, 23% for CO, 28% for BC and 32% for OC during 2013-2017. The dramatic reduction of emissions is mostly happened after 2013

(shown in Figure 1) for China's Clean Air Action implemented during 2013-2017. Compared to 2010, emissions of trace gas in 2014 decreased slightly except $SO_2$. So we only adjust the emission of $SO_2$.

[Figure]

Figure 1 Emission trends and underlying social and economic factors from 2010 to 2017 by Zheng et al. Zheng et al. (2018)

- Figure2: Why did you compare with NCEP R1, not with NCEP-FNL? What is the purpose of it?

Reply: Because the meteorological field of WRF is nudged to FNL datasets. We used NCEP R1 to compare with the simulation considering data independence.

- L244: This statement is not correct. The difference in RH2 between WRF and Reanalysis is much larger in general as shown in Fig2 over land area.

Reply: The statement is correct. The figure is wrong and it has been replaced.

- Table2 and L246-248: If you want to mention only the correlation coefficient of annual mean values, you should remove Table 2. If you want to retain Table2, you should explain the table more precisely here. Table 2 is hard to read and insufficient caption.

Reply: Thank you for your good suggestion. More descriptions (see below) about Table 2 is given in the revised manuscript, and captions and units are added in Table 2.

The simulation of the meteorological factors are close to the site records in different season, with mean bias (MB) of -0.3 ~ 0 °C, -0.8 ~ -0.5 m/s and -4~ -2.3% for $T_2$, $W_{10}$ and $RH_2$ respectively. The model underestimates $T_2$ in all the seasons. Thus the summer showed the largest negative bias with Root Mean Square Error (RMSE) of 2 °C, for temperature reaches the peak in summer. As for $W_{10}$, it's also underestimated the most in summer, with MB of -0.8m/s and RMSE of 1.9 m/s. As for RH2, the underestimation is more obvious in summer

(MB= -3.2%) and autumn (MB= -3.2%), mainly stem from the insufficient precipitation. Overall, the simulation in summer is more underestimated than other seasons. The agreement in $T_2$ and $RH_2$ with observations is better than that of $W_{10}$, with annual correlation coefficients (R) of 0.98, 0.84 and 0.53, respectively. Generally, the meteorology calculated by WRF can rational reproduce the characteristics of observations.

- L270: Why did you take average of 2006-2015 only for WDCGG?

Reply: The WDCGG datasets provides a large number of trace gases observations globally. The datasets can help to evaluate model performance of CO and ozone in different regions, but some sites are without invalid records in 2014. To get more data to evaluate the model over the world, we expanded the time range to ten years (2006-2015) and take the average of 2006-2015 as the statement of the air in the initial evaluation.

We have re-selected the observation data for 2014 to comparison with model results. Overall, the results have not changed much in terms of the evaluation of model's simulation capability. The simulation bias is reduced in some regions while it is increased in some other regions. The simulation of $NO_2$ performed better with the NMB of Asia and Europe closer to zero. The underestimation of CO in Antarctica disappeared due to the change of the observed value. There are some changes in the trend of the seasonal variation of $O_3$ in Northern Hemisphere. All the figures (as shown in Figure 2~ Figure 4) and tables related to these changes are updated in the manuscript, and the corresponding analysis is updated in the manuscript, too.

[Figure]

Figure 2 Annual mean concentration (ppb) of the surface layer in IAP-AACM. The circles represent site observations. The first row is CO and O$_3$, the bottom row is NO$_2$ and SO$_2$.

[Figure]

Figure 3 Scatter plots of annual mean concentrations (ppb) in Africa, Antarctica, Arctic Ocean (ArcticO), Asia, Atlantic Ocean (AtlanticO), Europe, Indian Ocean (IndianO), North America (NAmerica), South America (SAmerica), Oceania and Pacific Ocean (PacificO). The abscissa shows the observation and the ordinate shows the simulation. The color of the points represents different regions. (a) ~ (d) show CO, O$_3$, NO$_2$ and SO$_2$ respectively.

[Figure]

Figure 4 Mean seasonal variation of $O_3$ (ppb) over NAmerica, Europe, Asia, AtlanticO, PacificO, Antarctica, SAmerica, Africa and Oceania sites. Black lines and red lines represent the average of observations and simulations respectively. Gray shaded areas and red vertical bars show 1 standard deviation over the sites for observations and for model results respectively.

- Table3: It is better to include the information of region of each observation datasets.

Reply: That's a good suggestion and we have added the region of each observation datasets.

- L298: The value (23.3) differs from that in Table4.

Reply: Yes, it's a writing mistake. We have corrected it to 22.8 in the sentence.

- Table5: Table 5 is not completely filled with the necessary information for OM (other sources and total sink are missing).

Reply: Other sources and total sink are added to Table 5.

- Figure4: Fig4c and 4d should be switched to be in accordance with the order of panels in Fig3.

Reply: The subplot in Fig. 4 has been switched to the same order as Fig. 3.

- L363: Typo?: Northern Hemisphere

Reply: Yes, we have revised it.

L368-369: Why did IAP-AACM show the lowest concentration of CO over ocean among the models considered here?

Reply: It potentially reflects a difference in emissions. The natural sources of CO over ocean are included in the HTAP models whereas they are not considered in IAP-AACM. Besides, it may reflect differences in chemical transformation between models. The tropospheric (200hpa to the surface) mean OH concentration of IAP-AACM is $13.0 \times 10^5$ molec $cm^{-3}$. It is a little higher than the mean OH concentration study ($11.1 \pm 1.6 \times 10^5$ molec $cm^{-3}$) from 16 ACCMIP models for 2000 by Naik et al. (2013). It potentially leads to strong atmospheric oxidation. As shown in Figure 5, there is a slightly higher peak concentration of 30-35 molec $cm^{-3}$ in IAP-AACM, compared with the other models (under 30 molec $cm^{-3}$) (Huijnen et al., 2010; Badia et al., 2017). Due to the sink reaction of CO (CO + OH $\rightarrow$ $CO_2$ + H), the CO loss will be faster in IAP-AACM.

[Figure]

Figure 5 Zonal monthly mean concentration of OH for January, April, July and October by the IAP-AACM. The unit is $10^5$ molecule cm$^{-3}$.

- L378-380: The seasonal variation of surface O3 should be different in different environment even in the same region. So, I recommend the authors to compare separately for different environment (e.g. maritime area vs mountainous area). Otherwise, I can not regard the Fig6 as an evidence that the model can well simulate the seasonal variation of surface O3.

Reply: That's a helpful suggestion. The seasonal cycle of ozone shows different characteristics in different topographic conditions due to different control factors. We separate the observational sites as maritime area, inland and mountain due to the altitudes (shown in Figure 6). At the mountain sites the model tends to underestimate $O_3$ concentrations more. As high-altitude sites more frequently sample free tropospheric air (Fiore et al., 2009), it is more likely to be influenced by foreign emissions. The weaker intercontinental pollutant transport within troposphere in the model may be responsible for the underestimation. Moreover, steep topographic gradients at local may not be represented using coarse resolution models. Steep topographic gradients are averaged out within one model grid cell. It is difficult to capture the spatio-temporal variation of ozone caused by topographically driven flows or sharp gradients in mixing depths. For inland, the model tends to overestimate $O_3$ concentrations in summer time. Uncertainties in volatile organic compounds (VOCs)-NOx-$O_3$ chemistry may contribute. The natural source of isoprene from

vegetation is important in the $O_3$ formation due to its high proportion of VOCs emission in summer (as estimated to be 40.9 Tg/yr in China by Fu et al., 2012)

[Figure]

Figure 6 Mean seasonal variation of $O_3$ (ppb) over inland, mountain and maritime area in Northern Hemisphere compared with site records. Black lines and red lines represent the average of observations and simulations respectively. Gray shaded areas and red vertical bars show 1 standard deviation over the sites for observations and for model results respectively.

- L385: Underestimation in Antarctica is not small. Such an underestimation could be seen in the other CTMs. Can you use more words about this issue here?

Reply: In IAP-AACM, ozone concentration is about 10~15 ppb lower than site observations in Antarctica. An Evaluation of ACCMIP ozone simulations shows most models underestimate ozone concentration at high latitudes in the southern hemisphere at 800 hPa (Miyazaki & Bowman, 2017). As displayed in their seasonal comparison results, the underestimation of ozone is more significant (10~20 ppb) during April-August, which is in accordance with our study. It may be caused by a lack of halogen chemistry in our model. Remarkable ozone depletion events which is driven by halogen chemistry (mostly notably as bromine) is observed in the polar boundary layer (Simpson et al., 2007). Furthermore, Falk & Sinnhuber (2018) used EMAC v2.52 to interpret the significant underestimation of situ observed ozone in Antarctica, indicating that there are missing sources of bromine release from ice and snow in the model. The over prediction of dry deposition velocity to sea ice also plays a role in the underestimation of ozone. The dry deposition velocity to ice is under 0.02 cm s$^{-1}$ across 15 HTAP models (Ganzeveld et al., 2009). In IAP-AACM, it's obviously higher (0.035~0.048 cm s$^{-1}$) than those models, as shown in Figure 7.

[Figure]

Figure 7 Annual mean dry deposition velocity of ozone in IAP-AACM. The unit is cm s⁻¹.

- L385-387: In the NH land area, it seems that the model completely failed to represent the seasonal cycle of surface O3, but the author regarded it as just a positive bias during July-September. More words for this issue are necessary. For example, what do you think the apparent underestimation in cold season in NAmerica and Europe?

Reply: Yes, the model showed poor performance on the seasonal cycle of surface ozone in the NH land area, with overestimation in Europe and EA in summer while underestimation in winter in NH land, as shown in Figure 4 (the comparison drawn with WDCGG observation only for 2014).

According to Figure 6, the underestimation of ozone in cold seasons mainly occurs at sites of mountain and marine area, where is relatively clear and tend to be impacted by foreign pollutions. The underestimation in winter may relate to the weaker intercontinental pollutant transport within troposphere in the model. Winds are generally stronger in winter than in summer, causing intercontinental transport to be more rapid during winter months (HTAP, 2010). The observation at those sites will be influenced by source emissions more frequently.

The surface $O_3$ are also underestimated in spring over NH land. In IAP-AACM, the stratospheric-tropospheric exchange is not considered. It will lead to a large negative bias in simulation. To date it has become apparent that the measured annual cycle of ozone shows a distinct maximum during spring. The stratosphere-to-troposphere ozone transport event occurs widely across mid-latitudes in the NH (Monks et al., 2000; Akritidis et al., 2018). Since the magnitude and frequency of the transport through tropopause is still not clear. There are large uncertainties in simulating the flux. Some researches (Munzert et al, 1985; Austin and Follows, 1991) showed that the maximum in the stratosphere to troposphere flux occurs in late winter/spring. It may partly be responsible for the underestimation of $O_3$ in winter, too.

The surface $O_3$ concentrations over East Asia (sites mainly located in Japan) are overestimated in summer and early autumn. The same pattern is also found in the multi-model

inter-comparison of 21 HTAP models (Fiore et al., 2009). The simulations in island countries of EA are sensitive to the timing and extent of the Asian summer monsoon (Han et al., 2008). The positive model bias in this season may stem from inadequate representation of southwesterly inflow of clean marine air.

- L388-390: In Badia et al (2017), they suspected the excessive emission height of NOx which will cause low NOx at surface and consequently might lead to weak NO titration. Do the same things happen in your model?

Reply: Yes, there is the same situation in our model. In Badia et al. (2017), all the land-based anthropogenic emissions are emitted in the first 500m of the model. In IAP-AACM, the energy emissions and industry emissions are emitted in the first five layers considering the stack height, which the top injection height is over 500m.

- L390-391: The AACM apparently showed larger concentration of surface O3 in the tropical regions (central Africa, South America, and Southeast Asia) than the other models. However, the concentration of O3 precursor species (CO and NOx) in these regions are not so different among the models. Can you give discussion about the issue here?

Reply: Yes, the concentrations of CO and NOx in the tropical regions are not so different among the models. There are several uncertainties in the model performance. The same module schemes applied in different models may display different result (Tsigaridis et al. 2014; Hardecre et al., 2015). Furthermore, the meteorological conditions also play a important role in the simulation. The chemical reactions and dynamical processes (transportation and diffusion) of the matters are sensitive to meteorological field (e.g., wind, precipitation, temperature). In addition, the biomass burning emissions used in IAP-AACM is different from the other models. For multi-model activities of HTAP, groups use GFED3 data as the biomass burning emissions (Galmarini et al., 2017). In IAP-AACM, we use GFED4. A comparison of different versions of GFED emissions (Werf et al., 2017) shows the impact of a minor reduction in burned area and decreasing fuel consumption.

- L391-394: These two sentences are not consistent to each other. In general, the region of high O3 concentration can be different in different season. If you look at the "annual mean" concentration, the highest O3 usually occur in the source region in summer, but that in the downwind region in winter. However, if you see the different index such as MD8H O3, you can see completely different seasonal cycle. I strongly recommend the author to carefully revise these sentences.

Reply: We totally agree with your comment. For ozone, the seasonal cycle of high value area

between annual mean value and MD8H $O_3$ always show different patterns. As for the surface distribution of annual mean concentration shown in Fig. 5, the four models shows the common features of the the NOx titration effect that high concentrations mainly occur downwind of highly polluted areas. But compared with the other models, IAP-AACM exhibits a higher concentration in the source regions and a lower concentration downwind. As the $NO_2$ emission is emitted at a higher altitude in the model, it is transported to a further distance over the ocean. Thus the concentration of $O_3$ is lower in the source area (e.g., East China) due to a weaker NOx titration effect, and it's higher in the downwind area. We have revised those sentences.

- L402-403: An overall evaluation of O3 dry deposition in global CTMs can be seen in Hardecre et al. (2015). I recommend to check it out. Hardacre et al. (2015) An evaluation of ozone dry deposition in global scale chemistry climate model, Atmos. Chem. Phys., 15, 6419–6436, doi:10.5194/acp-15-6419-2015.

Reply: Thanks for your suggestion. According to Hardecre et al. (2015), the dry deposition velocity to sea varies little (around 0.05 cm $s^{-1}$) in different CTMs models using the deposition scheme by Wesely (1989). Besides, the study of Ganzeveld et al. (2009) shows that surface ozone differed by up to 60% if the deposition velocity of ozone varies from 0.01 to 0.05 cm $s^{-1}$. In IAP-AACM, the deposition velocity over the oceans varies from 0.042 to 0.05 cm $s^{-1}$, as shown in Figure 7. The variation in absolute terms between IAP-AACM and the other models is smaller than 0.008 cm $s^{-1}$. Hence the difference of surface ozone caused by dry deposition should be less than 12%. We revised sentences of line 402-403 in the manuscript.

- L416-418: The concentration of NOx over oceanic areas are larger in AACM than in other models, which might stem from larger emission or longer life time of NOx in AACM than the other models. I recommend to discuss this issue further here.

Reply: We totally agree with reviewer's suggestion. Compared with the other models shown in Fig. 5, the surface $NO_2$ over ocean is larger in IAP-AACM. This may reflect larger emission or less sinks of $NO_2$ in IAP-AACM. From our research, the most likely cause is the discrepancy in chemical conversion. As displayed in Fig. 7, nitrate is underestimated on the sea of East Asia, which implicates the oxidation of NOx to nitrate is insufficient in the model. Consequently, the higher $NO_2$ over ocean also leads to higher concentration of surface ozone over equatorial oceans, too. In addition, the higher injection height of emission sources leads to farther transportation distance. This is added in the revised manuscript.

- L436-438: This is misleading statement. The model results are not generally with in a factor of two, but they apparently tend to overestimate the observation in all the three regions. The NMB value for sulfate in Europe, 0.11, is incorrect which is 1.1 in Table 6.

Reply: The '0.11' is a typo, we have corrected it and renewed a new description about the simulation of sulfate as follows. In general, the surface distribution of SNA in IAP-AACM is close to the site records as shown in Fig. 7. When it comes to the site bias, Sulfate is overestimated more or less as shown in Fig. 8. Specifically, in Asia, the simulations at most sites here are within a factor of two of observations, with NMB of 0.36. However, In NAmerica and Europe, it's significantly overestimated, with NMB of 1.94 and 1.1 respectively.

- L438-439: How can you conclude like this (2ugm-3 higher)? What is the ground of this statement?

Reply: We calculated the sites average value in the model and compared it against the observation but we didn't mention it, now we have added this description in the revised paper.

- L442-443: What aspect of the observation do you think your model can reproduce? You should be more specific.

Reply: As shown in Fig. 7, IAP-AACM reproduces the nitrate distribution in western America well but overrates it in eastern America. Hence the west-east gradient over America is overestimated. The model doesn't fully capture the north-south gradient over Europe due to an overestimation at most of the sites. As for Asia, there is an underestimation in Southeast Asia and Japan. We have provided a detailed description of the distribution in the revised manuscript.

- L446-449: About the simulation of ammonium, I can see obvious underestimation in NAmerica and overestimation in Asia and Europe.

Reply: The simulation of ammonium is more diverse since there are more uncertainties in the emission of NH3 (precursor of ammonium) from croplands (Xu et al., 2019). There is slight negative bias in America and positive bias in Asia, with NMB less than ±1 (-0.46 and 0.85 respectively). In Europe, there is significant positive bias with NMB of 1.49. This has been added to the revised manuscript.

- L455-457: The concentration of OC were obviously underestimated by the model.

Reply: Yes, the meteorological conditions and emission inventories in the model are inconsistent with the observation year (2006) of carbonaceous in China. This may be partially

responsible for the bias of OC. According to recent study, there is a slightly increasing (less than 0.1Tg) of both BC and OC emissions from 2006 to 2010 in China (Lu et al., 2011; Fu et al., 2012). As the analysis of the CAWNET observation over China (Zhang et al., 2015), there is no significant changes happened in the proportion of chemical component of $PM_{10}$ from 2006 to 2013, which means the source of carbonaceous are also changed slightly. However, as shown in Fig. 8, the simulation of BC at most sites are close to observations while the simulation of OC is significantly underestimated. The study by Fu et al. (2012) showed a significant underestimation of OC emissions over China. Furthermore, Zhao et al. (2016) found that the pathway of intermediate volatile organic compounds (IVOC) to SOA is very important for the formation of SOA. Their experiments in three dimension model with new SOA scheme called 2B-VBS suggest that OA aging and intermediate-volatility emissions increased OA concentrations in Eastern China by 42%. IVOCs constitute over 40% of OA concentrations, and over half of SOA concentrations. Yang et al. (2018) also showed the significant increase of SOA concentration in an observation-based box model which included the IVOCs reactions. The reaction for IVOC is not included in our SOA module. The SOA module in IAP-AACM is Two-Product scheme. The simulations using Two-Product model substantially underestimated SOA and OA compared with observations. According to recent modeling studies with Two-Product scheme in China, it is estimated to an underestimation of OA by 40-78% (Lin et al., 2016; Han et al., 2016). This can explain the closely simulating of BC but greatly underestimating of OC and throw light on an improvement of SOA formation in the model.

- L491-492: The highest AOD in DJF in east China is not clearly seen both in satellite and model AOD.

Reply: Yes, it's an incorrect expression here and we have deleted it. In fact, the highest AOD may not be in DJF, it often appears in MAM. On one hand, China is frequently affected by dust in spring. On the other hand, AOD is an optical characteristic of aerosols for the whole vertical layer. It is not equivalent to the surface aerosol mass concentration.

- Figure10: It's better to show scatter plots too, at least as a supplement figure.

Reply: To be more quantitative, we provided scatter plots of the species in Figure 8. As shown in Figure 8, model results for $SO_2$, $PM_{10}$ and $PM_{2.5}$ are mostly within the factor of two with NMB within ±0.52, while $NO_2$ concentration are a bit underestimated (NMB= -0.63).

[Figure]

Figure 8 Scatter plots of annual mean concentrations (μg m⁻³) in nested domain. (a)~(f) is SO₂, NO₂, PM₁₀ and PM₂.₅ respectively. The abscissa shows the observation and the ordinate shows the simulation.

- Figure11: The area and the map projection of the figures for all models should be united.

Reply: The model results have been adapted to the same area and projection (shown in Figure 9). Also, a more detailed analysis has been displayed in the manuscript.

[Figure]

Figure 9 Annual surface distributions from nested IAP-AACM compared with regional models from MICS-Asia. Each row from top to bottom represents IAP-AACM, WRF-Chem, CMAQ and NAQPMS respectively. The left column is SO₂, the middle column is NO₂ and the right column is PM₂.₅. The unit for gases is ppb and for particles is μg m⁻³.

- L517-519: I'm sorry I can not understand what you want to mean here.

Reply: As shown in Fig. 11, the nested simulation of SO₂ in IAP-AACM has consistent spatial distribution of pollutants in the far northwest of the domain, which cannot be reproduced in the regional models. The PM₂.₅ from NQAPMS is higher than IAP-AACM in Northwest of China because it includes dust aerosol in NQAPMS.

L545-547: What do you want to mean here? Your model overestimated the NO2 in summer in NC and YRD regions. If you don't use the NO2* observation, the model's overestimation should become worse.

Reply: Yes, the overestimation of NO₂ reflects the shortcoming of multiphase processes in IAP-AACM. The overestimation of NO₂ and underestimation of nitrate in daytime of summer and autumn is related to the over decomposition of nitric acid at high temperature condition in the thermodynamic equilibrium module. Moreover, heterogeneous chemical reactions in the

model should partly be responsible for the $NO_2$ overestimation in summer. Reactive heterogeneous uptake of gases may be crucial for the formation of secondary aerosols when the other oxidants (e.g. ozone, OH) are in low concentrations level (Jacob, 2000; Martin et al., 2003). The heterogeneous chemical module coupled in IAP-AACM has been tested in North China in winter (Li et al, 2018). The uptake of $SO_2$ by wet aerosols significantly enhanced sulfate formation under highly polluted conditions, contributing 50%-80% of total concentration of sulfate. The mechanism also reduced the overestimation of nitrate which is also appeared in other models. However, when it comes to the problem here, we checked the simulations excluded heterogeneous chemical processes and found a better performance of $NO_2$ in summer (shown in Figure 10). It implicates that a more comprehensive mechanism should be considered in model development.

[Figure]

Figure 10 Seasonal cycle of $NO_2$ (μg m$^{-3}$) simulated without heterogeneous chemical process over China. The black line and red line represent monthly mean concentration of city-averaged observation and simulation respectively. Gray shaded areas and red vertical bars show 1 standard deviation over the sites for observations and for model results, respectively. MO and MM stand for annual mean concentration of observation and simulation respectively.

- L547-550: I can not understand what aspect of seasonal difference in NO2 column observation were reproduced by your model. You should describe more specifically on it.

Reply: As shown in Fig. S3, the model captured seasonal variations of NO$_2$ column concentrations in the vertical troposphere well. In China, the NO$_2$ VTC is higher during September-October-November and December-January-February while lower in June-July-August, likely caused by seasonal human activities such as fuel heating.

- L595: Typo? respects ! aspects

Reply: Yes, we have revised it.

- Conclusions should be revised according to the modifications made to respond the reviewers comments.

Reply: Yes, new conclusions will be updated in the manuscript.

**References**

Akritidis, D., Katragkou, E., Zanis, P.,et al. A deep stratosphere-to-troposphere ozone transport event over Europe simulated in CAMS global and regional forecast systems: analysis and evaluation. Atmospheric Chemistry and Physics, 18, 15515–15534, doi: 10.5194/acp-18-15515-2018, 2018.

Austin, J. F., Follows, M. J. The ozone record at Payerne: an assessment of the cross-tropopause flux. Atmospheric Envirnoment 25A, 1873-1880, 1991.

Dai, Y. J., Zeng, X. B., Dickinson, R. E. , Baker, I. ,Bonan, G. B. & Bosilovich, M. G.: The common land model. Bulletin of the American Meteorological Society, 84(8), 1013-1023, 2015. doi:10.1175/BAMS-84-8-1013, 2015.

Falk, S., & Sinnhuber, B. M. Polar boundary layer bromine explosion and ozone depletion events in the chemistry-climate model EMAC v2.52: implementation and evaluation of airsnow algorithm. Geoscientific Model Development, 11(3), 1-15, https://doi.org/10.5194/gmd-11-1115-2018, 2018.

Fiore,A. M., Dentener,F. J., Wild, O., et al. Multimodel estimates of intercontinental source-receptor relationships for ozone pollution. JOURNAL OF GEOPHYSICAL RESEARCH, 114, D04301, doi:10.1029/2008JD010816, 2009, 2009.

Fu,T.-M., Cao, J. J., Zhang, X. Y., Lee, S. C., & Henze, D. K.: Carbonaceous aerosols in china: top-down constraints on primary sources and estimation of secondary contribution. Atmospheric Chemistry and Physics, 12(5), 2725-2746, doi:10.5194/acp-12-2725-2012, 2012.

Galmarini, S., Koffi, B., Solazzo, E., Keating, T., Hogrefe, C., & Schulz, M., et al. Technical note: coordination and harmonization of the multi-scale, multi-model activities HTAP2,

AQMEII3, and MICS-Asia3: simulations, emission inventories, boundary conditions, and model output formats. Atmospheric Chemistry and Physics Discuss, 17(2), 1543-1555, 2017.

Han, Z., et al. MICS-Asia II: Model intercomparison and evaluation of ozone and relevant species, Atmos. Environ., 42, 3491 – 3509, doi:10.1016/j.atmosenv.2007.07.031, 2008.

Han, Z., Xie, Z., Wang, G., Zhang, R., & Tao, J. Modeling organic aerosols over east china using a volatility basis-set approach with aging mechanism in a regional air quality model. Atmospheric Environment,124, 186-198, 2016.

Hardacre, C., Wild, O., & Emberson, L.. An evaluation of ozone dry deposition in global scale chemistry climate models. Atmospheric Chemistry and Physics, 15(11), 6419-6436, doi:10.5194/acp-15-6419-2015, 2015.

Huijnen, V., Williams, J., van Weele, M., van Noije, T., Krol, M., Dentener, F., Segers, A., Houweling, S., Peters, W., de Laat, J., Boersma, F., Bergamaschi, P., van Velthoven, P., Le Sager, P., Eskes, H., Alkemade, F., Scheele, R., Nédélec, P., and Pätz, H.-W.: The global chemistry transport model TM5: description and evaluation of the tropospheric chemistry version 3.0, Geosci. Model Dev., 3, 445–473, doi:10.5194/gmd-3-445-2010, 2010.

Jacob, D. J. . Heterogeneous chemistry and tropospheric ozone. Atmos. Environ. 34, 2131–2159, 2000.

Jiang, X. , & Yoo, E. H. The importance of spatial resolutions of community multiscale air quality (CMAQ) models on health impact assessment. Science of The Total Environment, 627, 1528-1543, doi: 10.1016/j.scitotenv.2018.01.228, 2018.

Lee, Y. H., & Adams, P. J. : Evaluation of aerosol distributions in the GISS-TOMAS global aerosol microphysics model with remote sensing observations. Atmospheric Chemistry & Physics, 10(5), 2129-2144, 2010.

Li, J., Chen, X., Wang, Z., Du, H., Yang, W., & Sun, Y., et al. : Radiative and heterogeneous chemical effects of aerosols on ozone and inorganic aerosols over East Asia, Science of the Total Environment, 622, 1327-1342, https://doi.org/10.1016/j.scitotenv.2017.12.041, 2018.

Li, Y., & Xu, Y..: Uptake and storage of anthropogenic $CO_2$ in the pacific ocean estimated using two modeling approaches. Advances in Atmospheric Sciences, 29(4), 795-809, 2012. doi: 10.1007/s00376-012-1170-4, 2012.

Jian, L., An, J., Yu, Q., Yong, C., Ying, L., & Tang, Y., et al. Local and distant source contributions to secondary organic aerosol in the Beijing urban area in summer. Atmospheric Environment, 124, 176-185, 2016.

Lu, Z, Zhang, Q, & Streets, D. G.: Sulfur dioxide and primary carbonaceous aerosol emissions in China and India, 1996–2010. Atmospheric Chemistry and Physics, 11(18), 9839-9864, 2011.

Martin, R.V., Jacob, D.J., Yantosca, R.M. . Global and regional decreases in tropospheric oxidants from photochemical effects of aerosols. J. Geophys. Res. 108 (D3):4097. https://doi.org/10.1029/2002JD002622, 2003.

Miyazaki, K. , & Bowman, K. Evaluation of ACCMIP ozone simulations and ozonesonde sampling biases using a satellite-based multi-constituent chemical reanalysis. Atmospheric Chemistry and Physics, 17(13), 8285-8312, 2017.

Monks, P. S. A review of the observations and origins of the spring ozone maximum. Atmospheric Environment, 34(21), 3545-3561, 2000.

Munzert, K., Reiter, R., Kanter, H.-J., Potzl, K. Effect of stratospheric intrusions on the tropospheric ozone. In Proceedings of the Quad. Ozone Symposium, Halkidiki, Reidel, Dordecht, pp. 735-739, 1985.

Naik, V., Voulgarakis, A., Fiore, A. M., Horowitz, L. W., Lamarque, J.-F., Lin, M., Prather, M. J., Young, P. J., Bergmann, D., Cameron-Smith, P. J., Cionni, I., Collins, W. J., Dalsøren, S. B., Doherty, R., Eyring, V., Faluvegi, G., Folberth, G. A., Josse, B., Lee, Y. H., MacKenzie, I. A., Nagashima, T., van Noije, T. P. C., Plummer, D. A., Righi, M., Rumbold, S. T., Skeie, R., Shindell, D. T., Stevenson, D. S., Strode, S., Sudo, K., Szopa, S., and Zeng, G.: Preindustrial to present-day changes in tropospheric hydroxyl radical and methane lifetime from the Atmospheric Chemistry and Climate Model Intercomparison Project (ACCMIP), Atmos. Chem. Phys., 13, 5277–5298, doi:10.5194/acp-13-5277-2013, 2013.

Simpson, W. R., von Glasow, R., Riedel, K., Anderson, P., Ariya, P., Bottenheim, J., Burrows, J., Carpenter, L. J., Frieß, U., Goodsite, M. E., Heard, D., Hutterli, M., Jacobi, H.-W., Kaleschke, L., Neff, B., Plane, J., Platt, U., Richter, A., Roscoe, H., Sander, R., Shepson, P., Sodeau, J., Steffen, A., Wagner, T., and Wolff, E.: Halogens and their role in polar boundary-layer ozone depletion, Atmos. Chem. Phys., 7, 4375–4418, doi:10.5194/acp-7-4375-2007, 2007.

Task Force on Hemispheric Transport of Air Pollution (HTAP), Hemispheric Transport of Air Pollution, Part A: Ozone and Particulate Matter, edited by F. Dentener, T. Keating, and H. Akimoto, United Nations Publ. ECE/EB.AIR/100, U. N. Econ. Comm. for Eur. Inf. Serv., Geneva, Switzerland. 2010.

Tsigaridis, K., Daskalakis, N., Kanakidou, M., Adams, P. J., & Zhang, X.. The aerocom evaluation and intercomparison of organic aerosol in global models. Atmos. Chem. Phys., 14(19), 6027-6161, doi:10.5194/acp-14-10845-2014, 2014.

Xu, R. T., Tian, H. Q., Pan, S. F., Prior, S. A., Feng, Y. C., & Batchelor, W. D., et al.: Global ammonia emissions from synthetic nitrogen fertilizer applications in agricultural systems: Empirical and process-based estimates and uncertainty. Global Change Biology, 25, 314-325, doi: 10.1111/gcb.14499, 2019.

Wang, L. T., Jang, C., Zhang, Y., Wang, K., Zhang, Q., & Streets, D., et al.: Assessment of air quality benefits from national air pollution control policies in china. Part I: background, emission scenarios and evaluation of meteorological predictions. Atmospheric Environment, 44(28), 3449-3457, 2010.

Wang, L. T., Wei, Z., Yang, J., Zhang, Y., Zhang, F. F., & Su, J., et al. The 2013 severe haze over southern Hebei, China: model evaluation, source apportionment, and policy implications. Atmospheric Chemistry and Physics, 14(11), 28395-28451, doi:10.5194/acp-14-3151-2014,

2014.

Werf, G. R. V. D., Randerson, J. T. , Giglio, L. , Leeuwen, T. T. V. , Chen, Y. , & Rogers, B. M. , et al. Global fire emissions estimates during 1997–2016. Earth System Science Data, 9(2), 697-720. https://doi.org/10.5194/essd-9-697-2017, 2017.

Yang, W. Y., Li, J., Wang, M., Sun Y. L., Wang, Z. F.. A Case Study of Investigating Secondary Organic Aerosol Formation Pathways in Beijing using an Observation-based SOA Box Model, Aerosol and Air Quality Research, 18: 1606–1616, doi: 10.4209/aaqr.2017.10.0415, 2018.

Zhao, B., Wang, S., Donahue, N. M., Jathar, S. H., & Robinson, A. L. .. Quantifying the effect of organic aerosol aging and intermediate-volatility emissions on regional-scale aerosol pollution in china. Scientific Reports, 6, 2016.

---

## Author Response (AR1)

**The authors appreciate the reviewers very much for reviewing our manuscript and providing constructive comments. As suggested, we carefully revised the manuscript thoroughly according to the valuable advices, as well as proof-read the manuscript to minimize typographical, grammatical, and bibliographical errors. Our replies to the comments and our actions taken to revise the paper (in blue) are given below (the original comments are copied here). The revised parts in the manuscript are marked up in red.**

**Note: The figures added in the comments reply is represented by 'Figure', which is distinguished from 'Fig.' in the manuscript.**

**Anonymous Referee #3**

**General comments:**

This paper presents the atmospheric chemistry component of CAS-ESM, IAP-AACM, and compares the offline model results (driven by WRF) with various observational data worldwide. This is an important step towards improving the Earth system simulations by CAS/IAP, a key participant of IPCC assessments. Below are a few suggestions to improve the paper.

The model evaluation focuses on comparisons with measurements of surface concentrations of pollutants, particularly aerosol pollutants. Because this model is developed primarily for climate studies, evaluation of the tropospheric chemistry (in addition to surface air quality) will be very important. Specifically, It would be very useful to include/expand the evaluation of vertical profiles and tropospheric burdens against observations. There are many satellite data for ozone, NO2, SO2 and HCHO, and many vertical profile data (e.g., ATOM) for gaseous/aerosol species. Other important measures of tropospheric chemistry that can be discussed include the mean OH concentration and budgets, ozone budgets, methane lifetime, and MCF lifetime.

**Reply:** It is a good suggestion to include vertical comparison to improve the model evaluation work. We evaluated the tropospheric column concentration of NO2 and O3 with satellite data (GOME2A and OMI) and discussed the profile concentration of OH with other models. The budget of ozone and CO are also evaluated in Table 1. In addition, the ozonesonde measurements and simulated vertical profiles have been compared in the model evaluation of GNAQPMS. The evaluation of vertical profiles of O3 refers to Chen (2013) (shown as Figure 1).

**The budget for $O_3$ and CO:** The budgets for CO and $O_3$ are also displayed as a supplement in Table S2. As for CO, the total emissions are 994 Tg yr-1 in IAP-AACM. It's smaller than the other models (e.g., TM5:1159 (Huijnen et al., 2010), MOZART-4: 1210.7 (Emmons et al., 2010)). Direct emissions and oxidation contribute 43.4% and 55.4% to the total CO, respectively. The global burden is 327 Tg, smaller than the results of other

models (353~399 Tg) (Horowitz et al., 2003; Huijnen et al., 2010; Badia et al., 2017;). As for ozone, dry deposition contributes 21.3% to the total loss (4924 Tg yr-1), and photochemical reaction is responsible for the rest loss. The dry deposition (1049 Tg yr-1) is larger than the mean value of model collection of ACCENT and ACCMIP (Young et al., 2018).

Table 1 the budget of $O_3$ and CO compared with the other models.

| Species | Process | | IAP-AACM |
|---|---|---|---|
| CO | Emission (Tg yr$^{-1}$) Total 994 | Anthrop. | 546.4 |
| | | Bio. burning | 336.2 |
| | | Biogenic | 92.7 |
| | | Others | 18.3 |
| | Top condition inflow (Tg yr$^{-1}$) | | 28 |
| | Chem pro (Tg yr$^{-1}$) | | 1270 |
| | Chem lss (Tg yr$^{-1}$) | | 2292 |
| | Dry dep (Tg yr$^{-1}$) | | 0 |
| | Burden (Tg) | | 327 |
| | Lifetime (days) | | 52 |
| $O_3$ | Top condition inflow (Tg yr$^{-1}$) | | 398 |
| | Chemical production (Tg yr$^{-1}$) | | 4526 |
| | Chemical loss (Tg yr$^{-1}$) | | 3875 |
| | Dry dep. (Tg yr$^{-1}$) | | 1049 |
| | Burden (Tg) | | 370 |
| | Lifetime (days) | | 27.4 |

**Changes in the manuscript**: Please refer to Table S2 and Line 351-360.

[Figure]

Figure 1 Comparison of vertical profiles between ozonesonde measurements and simulation in GNAQPMS, it followed the methodology of Tilmes et al. (2012) for the selection and treatment of the measurements.

**The comparison with O₃ satellite observation:** The vertical tropospheric column (VTC) of $O_3$ is compared against satellite observation derived from OMI (shown in Figure 2). In the main board, the pattern of the seasonal cycle was covered by the model. In mainland of Northern Hemisphere the higher $O_3$ VTC appears during June-July-August (JJA), while in Southern Hemisphere it appears during September-October-November (SON),with a range of 40-60 DU. The model keeps a high value (40-50 DU) in tropics during DJF, possibly due to the high emission of CO in biomass burning. The underestimation of cloud cover in the Intertropical Convergence Zone may contribute, too. The $O_3$ VTC is underestimated over ocean in middle-high latitudes. As the stratospheric chemistry is not considered in IAP-AACM. The lack of stratospheric-tropospheric exchanges should partly be responsible for the underestimation of column burden.

**Changes in the manuscript**: Please refer to Fig. 8 and Line 558-568.

[Figure]

Figure 2 Seasonal mean column concentration of $O_3$ in IAP-AACM (left column) and OMI (right column). Seasons are defined as December-January-February (DJF), March-April-May (MAM), June-July-August (JJA), and September-October-November (SON). The unit is DU.

**The comparison with NO₂ satellite observation:** The VTC of $NO_2$ is compared against satellite observation derived from GOME-2A (shown in Figure 3). The $NO_2$ VTC has a range of 20-150 $\times 10^{14}$ molecule $cm^{-2}$ in most source areas. By and large, IAP-AACM reproduced the magnitude in different regions. In addition, the model captured seasonal variations of $NO_2$ concentration in the vertical troposphere well. In anthropogenic source areas of Northern Hemisphere (e.g., North America, Europe, East Asia), the $NO_2$ VTC is higher in SON and December-January-February (DJF) while lower in JJA, caused by unfavorable diffusion conditions and weak photochemistry. The column concentration is higher during JJA in South America and South Africa, while it is higher during DJF in central Africa, due to the vegetation burning in dry season. Compared with GOME-2A, IAP-AACM showed a larger column concentration over ocean. The overestimation is also reflected in the comparison of surface concentration. This is probably caused by insufficient oxidation to nitrate and a higher injection height of emission which leads to a farther transportation distance as suggested in Badia et al. (2017). Generally, the distribution of $NO_2$ by the model is consistent with satellite observation.

**Changes in the manuscript**: Please refer to Fig. 9 and Line 569-582.

[Figure]

Figure 3 Seasonal mean column concentration of $NO_2$ in IAP-AACM (left column) and GOME-2A (right column). Seasons are defined as December-January-February (DJF), March-April-May (MAM), June-July-August (JJA), and September-October-November (SON). The unit is $10^{14}$ molecule $cm^{-2}$.

**The zonal distribution of OH:** Oxidation is the basic characteristic of atmospheric chemistry. As the most important oxidant in atmosphere, OH is the crucial species in CTMs. OH in troposphere is mainly produced by the reaction $O_3 + h\nu$ ($\lambda \leqq 320nm$) + $H_2O \rightarrow$ 2OH+$O_2$. The tropospheric mean concentration of OH in IAP-AACM is $13.0 \times 105$ molec cm-3. It is a little higher than the mean OH concentration ($11.1 \pm 1.6 \times 105$ molec cm-3) given by 16 ACCMIP models in Naik et al. (2013). The high concentration indicates a stronger atmospheric oxidation. This could explain the lower concentration of CO over ocean. The zonal mean OH concentrations for January, April, July and October are shown in Fig. 3. Like other chemistry models, OH concentration in the tropics keeps highest all the year round and decreases gradually from tropics to poles. This is due to the positive influence of solar radiation and water vapor concentration. The seasonal north-south shift of OH maximum area is also ascribed to the seasonal variation of these two factors. The mean inter-hemispheric (N/S) ratio of OH in the model is 1.26, in accordance with the multi-model mean ratio of $1.28 \pm 0.1$ (Naik et al., 2013). Vertically, the highest concentration is in the layer of 2-4 km over the tropics. In Northern Hemisphere, the highest OH concentration appears in summer. Peak value is located at around 30 °N, in the atmosphere above 2km. Generally, the distribution of OH concentration is similar with other models

(Huijnen et al., 2010; Badia et al., 2017).

**Changes in the manuscript**: Please refer to Fig. 3 and Sect. 3.2.1.

[Figure]

Figure 4 Zonal monthly mean concentration of OH for January, April, July and October by the IAP-AACM. The unit is $10^5$ molecule cm$^{-3}$.

Measurement data often contain missing values and outliers and have different temporal resolutions from model simulations. Please specify how the measurement data are processed and how model results are sampled (temporally and spatially) according to measurements. In particular, satellite data contain large amounts of missing values. Near-surface NO2 measurements are contaminated by other nitrogen species, and what would be the implications for model evaluation (especially when discussing the model bias).

**Reply:** The measurement datasets (except CNEMC) collected in this paper are monthly or annual mean results which have been processed by the observation workgroups. The hourly CNEMC observations are processed by data quality control. The corresponding simulation data compared with aforementioned observations are sampled in the model grid cells containing observational sites. The simulation of seasonal cycle in different regions or cities are first sampled at the model grid cells containing the observational sites and then averaged within sub-regions. When compared with satellite data, the missing values of satellite data are kept and shown in the figures.

As shown in the scatter plot in Figure 12, model results for $NO_2$ concentrations are in good agreement with observation with NMB of -0.02. As the "$NO_2$" values reported by routine monitoring sites are $NO_2^*$, which

partially includes $HNO_3$ and $NO_3^-$. It implicates that the model may overestimate "$NO_2$".

**Changes in the manuscript**: The description of simulations sampled according to site observations is revised in Line 295-298. The discussion on model bias between NO2 and $NO_2^*$ is revised in Line 620-622 and Fig. 11(b) in the manuscript.

The resolution dependence discussed in Sect. 3.4 has also been studied in other recent works. It would be nice to refer to or compare against previous findings.

**Reply:** That's a good suggestion. High-resolution helps to improve CTMs performance, but it is limited by the scale applicable to the parameterization scheme of physical and chemical processes. Recently, sensitivity to horizontal grid resolution has been discussed in many regional model works. Wang et al. (2014) showed a better simulation of particles in North China with CMAQ when increasing the resolution from 36km to 12km. A study of $PM_{2.5}$ heath impact assessment with CMAQ by Jiang et al. (2018) found that model results at 12 km generally performed better and had substantially lower computational burden, compared to 4 km resolution.

**Changes in the manuscript**: Previous model resolution studies are discussed in Line 722-729 in Sect. 3.4.

The spin-up time (one month) is too short for CO, ozone and other longer-lived species. This may explain part of the underestimate in CO. Please comment on the effect of spin-up time.

**Reply:** We agree that the spin-up time of one month is not enough for longer-lived species. It may lead to an underestimation of some trace gases such as CO. But in this study we used monthly mean concentration of CO, $O_3$ and $NO_2$ from MOZART-4 as initial conditions and top boundary conditions. It can offset the potential underestimation of CO and $O_3$ substantially. Furthermore, to verify the effect of shorter spin-up time here, we also run a case with spin-up time of one year. The annual mean result is similar to the case of one month spin-up time as shown in Figure 5.

The underestimation of CO potentially reflects a difference in emissions. The natural sources of CO over ocean are included in the HTAP models whereas they are not considered in IAP-AACM. Besides, it may partly owe to differences in chemical transformation between models. As shown in Figure 4, the OH concentration is a bit higher in IAP-AACM than the other models. Due to the sink reaction of CO ($CO + OH \rightarrow CO_2 + H$), the CO loss will be faster in IAP-AACM.

**Changes in the manuscript**: The discussion on the underestimation of CO refers to Line 396-401.

[Figure]

Figure 5 The annual mean surface concentration of CO. The left one is the surface distribution with one month spin-up time, the right one is with one year spin-up time.

There have been discussions in the literature on bug fixes in ISOROPIA II. Are these bugs and fixes relevant here?

**Reply:** No, it's not relevant here. The code bug only affects the forward (in which the concentration of both gas and aerosol of each species is fixed) stable state calculation. In IAP-AACM, we use reverse mode (in which the concentration of each species in the aerosol phase is fixed) to calculate.

Brief descriptions of WRFv3.3 would be very useful. The vertical resolution of WRF is different from that in IAP-AACM, so how is the conversion done?

**Reply:** The WRF version used in this study is a global version of WRFv3.3. It is an extension of mesoscale WRF that was developed for global weather research and forecasting applications. It has more general choice of map projection (to include both conformal and non-conformal map projections). The specification of planetary constants, physics parameterizations and timing conventions are also improved to allow the model to be run as a global model. Thus, it has multiscale and nesting capabilities, blurring the distinction between global and mesoscale models and enabling investigation of coupling between processes on all scales (Richardson et al., 2007).

Output of WRF is interpolated to the vertical layers defined in IAP-AACM.

**Changes in the manuscript**: The information of WRFv3.3 has been added in Line 231-239 of the revised manuscript. The interpretation of interpolation in the vertical is added in Line 242.

Table 1 – do you extrapolate the emissions to 2014? If not, what would be implications for your model evaluation against measurements in 2014?

Reply: Yes,we extrapolate the emission of $SO_2$ to 2014. As a consequence of government control policy included in the twelfth Five-Year Plan (FYP), China has achieved an obvious decrease in air pollution in the past years, especially for $SO_2$. The FYP controls suppress $SO_2$ emissions in energy and industry sectors which is the major source of $SO_2$. Considering the cutting effect on $SO_2$ (China completed the emission reduction

task of 12th FYP (2010~2015) ahead of schedule in 2014 with a reduction ratio reaching by 12.9%), we adjusted the total $SO_2$ emission for 2014 by a factor of 0.9 in China. For other species, the intensity of emission reduction is not so great like $SO_2$. The study by Zheng et al. (2018) showed that the dramatic reduction of emissions is mostly happened after 2013 for China's Clean Air Action implemented during 2013-2017. Relative change rates of China's anthropogenic emissions during 2010–2017 are estimated as follows: -62% for SO2, -17% for NOx, -27% for CO, -27% for BC and -35% for OC. And the emission mostly decreased during 2013-2017, by 59% for SO2, 21% for NOx, 23% for CO, 28% for BC and 32% for OC.   Compared to 2010, emissions of trace gas in 2014 decreased not significant except $SO_2$ (shown in Figure 6). So we only extrapolate the emission of $SO_2$. It will partly be responsible for the overestimation of some species (e.g., $NO_2$ in Fig. 16) in our simulation.

[Figure]

Figure 6 Emission trends and underlying social and economic factors from 2010 to 2017 by Zheng et al. (2018).

**Changes in the manuscript**: The interpretation and reference are added in Line 227-229.

In the comparisons over China, only a few cities are selected, although there are CEMC measurements in other cities as well. Please explain the rationale for choosing these cities.

**Reply:** The cities are selected in six regions (North China, Pearl River Delta, Yangtze River Delta, Northwest China, Central China, Southwest China). The six regions not only represent the major geographical regions over China, but also include regions with the most severe air pollution at present which are studied most.

**Changes in the manuscript**: The reason for choosing these cities are shown in Line 304-306.

**Specific comments:**

Abstract – please specify which part of the writing is for the evaluation of global model and which is for nested model. Also, please present the bias (in addition to R) of the model.

**Reply:** The global and nested description has been specified and NMB has been added in the manuscript.

**Changes in the manuscript**: It has been revised in the Abstract, in Line 25, Line 35-36 and Line 40.

L48-67 – the references are relatively old. Please use newer ones. Also, aerosols affect the cardiovascular diseases very significantly.

**Reply:** The citation of IPCC has been updated to the latest report. References for aerosols' health effect are also updated (see below).

Aerosols formatted from these precursor gases, together with aerosols from other sources, have a direct radiative forcing. By modifying cloud properties, the aerosols also have important indirect effects. As reported in the Fifth Assessment Report (AR5) of IPCC (Myhre et al., 2013), the radiative forcing of aerosols ranges from -1.9 ~ -0.1 W m$^{-2}$, with the direct radiative forcing ranges from -0.85 ~ 0.15 W m$^{-2}$. With better model performance and more robust observation network, AR5 achieved increasing confidence in the assessment compared with AR4 (Boucher et al., 2013), but the largest uncertainty to the total radiative forcing estimate is still aerosols. In addition, aerosols have adverse impacts on human health including respiratory diseases, cardiovascular risk and lung cancer, which has drawn increasing public attention (Burnett et al., 2014; Pope et al., 2011; Powell et al., 2015).

**Changes in the manuscript**: References for aerosols' health effect and IPCC are updated in Line 55-65.

L71 – change "prediction" to "projection"

**Reply:** It has been corrected in Line 72.

L87-88 – there have been model evaluation studies over China in recent years. Please refer to these studies.

**Reply:** Yes, there have been several model evaluation studies with observation in China. The discussion has been added in Line 725-726 in the manuscript.

L97 – remove "precise". Every model has its limitations.

**Reply:** It has been deleted in the revised manuscript.

L100 – change to "lateral (and upper) boundary conditions"

**Reply:** It has been modified in the revised manuscript in Line 102.

L147 – specify the resolution

**Reply:** The high resolution is 0.25 °×0.25 °, we have specified it in Line 154.

L160 – do you mean "natural dust"?

**Reply:** Yes,it is.

L199 – do you mean the first layer center is 50 m?

**Reply:** Yes, we have specified the meaning in Line 204.

Table 2 – please explain the meanings of these statistics and provide the units.

**Reply:** Captions and units are added to Table 2.

L279 – why not just use the WDCGG data in 2014?

**Reply:** The dataset of WDCGG provides a large number of trace gases observations globally. But there is no observation data for some sites in 2014. To get more data to evaluate the model over the world, we expanded the time range to ten years (2006-2015).

We have re-selected the observation data for 2014 to comparison with model results (shown in Figure 7~ Figure 9). Overall, the results have not changed much in terms of the evaluation of model's simulation capability. The simulation of $NO_2$ performs better with the NMB closer to zero in Asia and Europe. The underestimation of CO in Antarctica disappeared due to the change of the observed value. There are some changes in the trend of the seasonal variation of $O_3$ in Northern Hemisphere.

**Changes in the manuscript**: All the figures and tables related to these changes are updated in the manuscript (please refer to Fig. 4 and Fig.6), and the corresponding analysis is updated in the manuscript, too (please refer to Sect. 3.2.2).

[Figure]

Figure 7 Annual mean concentration (ppb) of the surface layer in IAP-AACM. The circles represent site

observations. The first row is CO and $O_3$, the bottom row is $NO_2$ and $SO_2$.

[Figure]

Figure 8 Scatter plots of annual mean concentrations (ppb) in Africa, Antarctica, Arctic Ocean (ArcticO), Asia, Atlantic Ocean (AtlanticO), Europe, Indian Ocean (IndianO), North America (NAmerica), South America (SAmerica), Oceania and Pacific Ocean (PacificO). The abscissa shows the observation and the ordinate shows the simulation. The color of the points represents different regions. (a) ~ (d) show CO, $O_3$, $NO_2$ and $SO_2$ respectively.

[Figure]

Figure 9 Mean seasonal variation of $O_3$ (ppb) over NAmerica, Europe, Asia, AtlanticO, PacificO, Antarctica, SAmerica, Africa and Oceania sites. Black lines and red lines represent the average of observations and

simulations respectively. Gray shaded areas and red vertical bars show 1 standard deviation over the sites for observations and for model results respectively.

CO model evaluation – could you comment on the effect of spinup time and coarse model resolution?

**Reply:** We agree that the spin-up time of one month is not enough for longer-lived species. It may lead to an underestimation of some trace gases such as CO and $O_3$. But in this study we used monthly mean concentration of CO, $O_3$ and $NO_2$ from MOZART-4 as initial conditions and top boundary conditions. It can offset potential underestimation of CO and $O_3$ substantially. Furthermore, to verify the effect of the shorter spin-up time here, we also run a case with spin-up time of one year. The annual mean result is almost the same with the case of one month spin-up time as shown in Figure 5.

On one hand, the results of coarse-resolution models are often lower than those of high-resolution models due to the effect of gridded average on static emission sources. On the other hand, it's difficult to reproduce the atmospheric dynamics characteristics under complex underlying surface conditions for coarse resolution models. The coarse resolution of global models can hardly represent local orographically driven flows or sharp gradients in mixing depths. It's unfavorable to simulate pollutant diffusion process.

L416 – the ozone seasonality is not very well captured in many regions. Also, this paragraph is too long.

**Reply:** Agree. We have reanalyzed the simulation of ozone in this part in the revised manuscript. The model showed poor performance on the seasonal cycle of surface ozone in the NH land, with overestimation in Europe and EA in summer while underestimation in winter in NH land, as shown in Figure 9 (the plot with WDCGG observations only for 2014).

The surface $O_3$ are also underestimated in spring over NH land. In IAP-AACM, the stratospheric chemistry is not considered. Thus the stratospheric-tropospheric exchange is weak. It leads to a large negative bias in the simulating. To date it has become apparent that the measured annual cycle of ozone shows a distinct maximum during spring. The stratosphere-to-troposphere ozone transport event occurs widely across mid-latitudes in the NH (Monks et al., 2000; Akritidis et al., 2018). Since the magnitude and frequency of the transport through tropopause is still not clear. There are large uncertainties in simulating the flux. Some researches (Munzert et al, 1985; Austin and Follows, 1991) showed that the maximum in the stratosphere to troposphere flux occurs in late winter/spring. It may partly responsible for the underestimation of $O_3$ in winter, too.

The surface $O_3$ concentrations over East Asia (sites mainly located in Japan) are overestimated in summer and early autumn. The same pattern is also found in the multi-model inter-comparison of 21 HTAP models (Fiore et al., 2009). The simulations in island countries of EA are sensitive to the timing and extent of the Asian summer monsoon (Han et al., 2008). The positive model bias in this season may stem from inadequate representation of southwesterly inflow of clean marine air.

**Changes in the manuscript**: The discussion on ozone evaluation is updated, please refer to Line 432-468.

L452 – please specify the quantitative difference between GFED3 and GFED4.

**Reply:** GFED3 and GFED4 are both monthly burned area emission data gridded to 0.5 °×0.5 °and 0.25 °×0.25 °, respectively. Due to the impact of a reduction of combustion area and decreasing in fuel consumption, there is about a reduction of 20%~30% for CO emissions in GFED4 compared with GFED3 in the burned areas (Werf et al., 2017).

**Changes in the manuscript**: The specific difference has been added to the manuscript, please refer to Line 412-417.

L485-499 – please comment on the effect of difference in time (2006 for measurements and 2014 for model simulation).

**Reply:** As the simulation used emissions of 2010 but the measurements are for 2006, there is a mismatch on emission scenario. Besides, the meteorological conditions also play a role.

As the analysis of the CAWNET observation over China (Zhang et al., 2015), there is no significant changes happened in the proportion of chemical component of $PM_{10}$ from 2006 to 2013. For the annual average trends of carbonaceous shown in Figure 10, both Southwest China and North China experienced a process of declining first and then rising due to the unfavorable weather conditions. Pearl River Delta showed a significant falling (about half). Yangtze River Delta had a slight decreasing. Generally, it is reasonable to infer that the distribution of BC and OC in most areas have changed a little from 2006 to 2014, except for the Pearl River Delta region.

[Figure]

YRD          PRD

[Figure]

SWC                    NC

Figure 10 Monthly mean concentrations of OC and EC from 2006 to 2013 by Zhang et al., 2015. YRD, PRD, SWC and NC represents Yangtze River Delta, Pearl River Delta, Southwest China and North China, respectively.

L495 – BC depends on emissions and deposition processes.

**Reply:** Yes, it has been corrected, please refer to Line 533.

L500 – please clarify which components are included in PM2.5

**Reply:** To be uniform in the context, the figures of $PM_{2.5}$ showed in the revised manuscript are all calculated with components of primary $PM_{2.5}$, BC, POA, SOA and SNA.

**Changes in the manuscript**: The clarification is added to the manuscript in Line 626.

L506 – please specify the version of MODIS AOD and how data are selected/sampled.

**Reply:** The product version is MYD04_L2-MODIS/Aqua Aerosol 5-Min L2 Swath 10km. It is available at the website: http://dx.doi.org/10.5067/MODIS/MYD04_L2.006.

**Changes in the manuscript**: The product version and website is supplemented in the revised manuscript. Please refer to Line 286.

L512 – LAC or BC?

**Reply:** It should be BC, it has been revised in Line 589.

L522-531 – please consider to present the seasonality results in a line figure.

**Reply:** That's a good suggestion. A more detailed comparison of the global gridded average AOD on the seasonality variation is displayed in Figure 11. As the seasonality cycle is different in different regions, we not only showed the global average value, but also showed the gridded average value of Africa, South America and East Asia, which are major aerosol emission areas. Generally, the model captured seasonal variation in different regions. The discrepancy in East Asia potentially stemmed from the bias of dust simulation in spring.

**Changes in the manuscript**: The figure is supplemented as Fig. S4, and the analysis is shown in Line 307-308 in the revised manuscript.

[Figure]

Figure 11 Gridded mean value of monthly averaged AOD for 2014, AF, EA, SA and GL represents Africa, East Asia, South America and global. Dash line and solid line represents model results and observation derived from MODIS, respectively.

L526 – In the model, DJF is not the season with the highest AOD over East China.

**Reply:** Yes, it's an incorrect expression here and we have deleted it. In fact, the highest AOD may not be in DJF, it often appears in MAM. Since East Asia is frequently affected by dust in spring, this phenomenon is common in other model evaluation studies (e.g., GISS-TOMAS (Lee et al., 2010)). The seasonal variation of relative humidity also impacts the simulating of AOD.

L548-558 – please be more quantitative.

**Reply:** To be more quantitative, we provided scatter plots of simulations in the nested domain in Figure 12. As shown in Figure 12, model results for $NO_2$, $SO_2$ and $PM_{2.5}$ are mostly within the factor of two with NMB within ±0.3. $PM_{10}$ concentrations are underestimated at all sites with NMB of -0.51 due to the simulation without dust.

**Changes in the manuscript**: The figure is added in the revised manuscript, as Fig. 11(b). Descriptions refer to Line 621-625.

[Figure]

Figure 12 Scatter plots of annual mean concentrations (μg m$^{-3}$) in nested domain. (a)~(f) is SO$_2$, NO$_2$, PM$_{10}$ and PM$_{2.5}$ respectively. The abscissa shows the observation and the ordinate shows the simulation.

L567-568 – please provide model versions.

**Reply:** the model versions are CMAQv4.7.1, WRF-Chemv3.9 respectively.

**Changes in the manuscript**: This has been added in the revised manuscript. Please refer to Line 628-629.

L572 – do you mean "other regional models"?

**Reply:** It means the regional model. Here we compared the simulation of the nested domain in IAP-AACM with regional models of MICS-Asia.

L575-576 – what are the differences in emissions?

**Reply:** The differences of emissions between IAP-AACM and MICS-Asia models are natural sources. For anthropogenic source, IAP-AACM uses MIX inventory (incorporated into HTAP for Asia) as same as MICS-Asia models. For biogenic source, IAP-AACM uses MEGAN-MACC but models of MICS-Asia uses an earlier version of MEGANv2.04. For biomass burning source, IAP-AACM uses GFEDv4 but MICS-Asia models uses GFEDv3.

Fig. 14 – please specify the components in PM2.5

**Reply:** To be uniform in the context, the figures of PM$_{2.5}$ showed in the revised manuscript are all calculated

with components of primary PM$_{2.5}$, BC, POA, SOA and SNA.

**Changes in the manuscript**: It is added in the caption of Fig. 15.

Table 6 – please specify which one is global model and which one is regional model. Also, please provide the mean values over these cities.

**Reply:** Do you mean Table 7 in the ACPD document, the statistics for 12 cities in global and nested domains? If so, all the results in this table are calculated with outputs from the global model IAP-AACM. The difference between D1 and D2 is the horizontal resolution. D1 represents domain 1 (1 °×1 °), D2 represents domain 2 (0.33 °×0.33 °).

**Changes in the manuscript**: The mean values over these cities are added in Table 6.

**Anonymous Referee #2**

**Major comments:**

The main purpose of this paper is to clearly show the ability and/or inability of their model to simulate the observed spatial and temporal variation in the concentration of the chemical species in the atmosphere. Based on those findings the authors and also the readers of the paper can understand what kind applications are suitable for this model and what aspect of the model should be further improved in order to apply it to a particular issue. From this point of view, the self-evaluation about the ability of the model by authors were often insufficient and unclear. The good points and also the shortcomings of the model should be described more specifically in the text. I pointed out some of those points as specific comments in the following, but I strongly recommend the authors to reexamine the descriptions particularly in the model evaluation parts.

**Reply:** We greatly appreciate the reviewer for insight comments on the manuscript. To respond to the reviewer's major concerns, we made thoroughly revisions and corrections according to all the insight comments of the reviewers. Besides, more crucial information and analysis will be added in the revised manuscript, as follows:

(1) A new evaluation with the WDCGG datasets only for 2014 is updated, and the evaluation is more quantitative.

(2) We provide more information to discuss the model's performance on the underestimation of CO over ocean, including a comparison of the profile concentration of OH with other models.

(3) More discussions and descriptions are added in the evaluation of ozone. The bias of inter-models and model-observation are discussed. In particular, the poor performance on the seasonal cycle in NH land is interpreted. We further showed the seasonal cycle of ozone compared against sites separated by the terrain.

(4) The model's ability in aerosol simulating is discussed in detail. Especially, the SOA formation mechanism and the multiphase processes in the model are described.

(5) Overall, more analysis on the model's performance are shown, and some improvements are put forward in the model's further work.

**Specific comments:**

- L24: What are the aerosol effects here?

**Reply:** The aerosol effects refer to climate effect (direct, semi-direct and indirect effect) and health effect

(mainly of respiratory diseases, cardiovascular risk and lung cancer).

**Changes in the manuscript**: It has been added in Line 25 in the revised manuscript.

- L38: Only R-value can not ensure the accuracy of the simulation. How about MB or NMB?

**Reply:** For most of the cities, NMB for the nested simulations are within ±0.5, and MB for the nested simulations are within ±25.

**Changes in the manuscript**: The description has been added in the Abstract. Please refer to Line 40.

-L58: Why didn't you cite the latest AR5 report here?

**Reply:** Thanks for your suggestion, the citation are updated to the latest report.

**Changes in the manuscript**: The update refer to Line 55-62.

- L79: Typo? e.g.

**Reply:** 'For example' is used to introduce the work by Badia et al. (2017), Mann et al. (2010) and Tsigaridis et al. (2014) instead of 'e.g.' in the manuscript.

- L86: EA, this should be defined at its first appearance in the text.

**Reply:** Thanks, the definition has been added at its first appearance (Line 91) in the revised manuscript.

- L108-109: What do you mean here? Could you use more words to explain "localization of the process parameterization"?

**Reply:** In the dust module, the deflation mechanism and dust loading parameterization are based on a detailed analysis of the meteorological conditions, landform, and climatology from daily weather records at about 300 local stations in North China. For the heterogeneous chemistry scheme, the parameterization of uptake coefficients improved the simulating of sulfate and nitrate in severe haze period in China.

**Changes in the manuscript**: It has been added in the manuscript. Please refer to Line 110-115.

- L135-139: Are there citable references for CoLM , and IAP-OBGCM?

**Reply:** the references for CoLM (Dai et al., 2015), and IAP-OBGCM (Li et al., 2012) has been supplemented in the revised manuscript.

**Changes in the manuscript**: Please refer to Line 143 and Line 146.

-L158: What is the main difference between these two models (GNAQPMS and IAP-AACM)?

**Reply:** Generally, IAP-AACM is similar to GNAQPMS. IAP-AACM has the same model framework with GNAQPMS but has some improvements. It extended the gas phase chemistry from CBMZ to an alternative

simplified scheme specifically for CAS-ESM. The model was renamed when it joined the CAS-ESM.

- L204: What does "synchronous time step" mean ?

**Reply:** It is the time step of model's integration calculation. In order to keep the stability of calculation in the model, the integration time step will be cut into shorter sub-integration time step in different modules (e.g., advection and gas chemistry processes).

- L206: What is the reason for choosing the year 2014 as the focal year?

**Reply:** The year 2014 is closest year to 2010 having global emission inventory. In addition, observation data sets were only available in 2014 since Chinese National Environmental Monitoring Network (CNEMC) started to publish data in 2013.

- L224-225: Emission data used in the study are not up-to-date, the base year of each database is a bit old. Therefore, adjusting the emission data to input them to the model is suitable for the purpose of this study. However, you only mentioned about the adjustment of SO2 emission in China in the text. Did you adjust other species emission?

**Reply:** No, we only adjust the emission of $SO_2$ for its dramatic variation in the past years in China. During 2010~2014, the change of $SO_2$ emissions are significant in China, due to a strict controlling policy by the government. The study by Zheng et al. (2018) shows that relative change rates of China's anthropogenic emissions during 2010–2017 are estimated as follows: -62% for SO2, -17% for NOx, -27% for CO, -27% for BC and -35% for OC. But the emissions decreased by 59% for SO2, 21% for NOx, 23% for CO, 28% for BC and 32% for OC during 2013-2017. The dramatic reduction of emissions is mostly happened after 2013 (shown in Figure 6) for China's Clean Air Action implemented during 2013-2017. Compared to 2010, emissions of trace gas in 2014 decreased slightly except $SO_2$. So we only adjust the emission of $SO_2$.

**Changes in the manuscript**: More words to explain are added in Line 227-229 in the revised manuscript.

- Figure2: Why did you compare with NCEP R1, not with NCEP-FNL? What is the purpose of it?

**Reply:** Because the meteorological field of WRF is nudged to FNL datasets. We used NCEP R1 to compare with the simulation considering data independence.

- L244: This statement is not correct. The difference in RH2 between WRF and Reanalysis is much larger in general as shown in Fig2 over land area.

**Reply:** The statement is correct. The figure is wrong and it has been replaced.

**Changes in the manuscript**: Please refer to Fig. 2.

- Table2 and L246-248: If you want to mention only the correlation coefficient of annual mean values, you

should remove Table 2. If you want to retain Table2, you should explain the table more precisely here. Table 2 is hard to read and insufficient caption.

**Reply:** Thank you for your good suggestion. The simulation of the meteorological factors are close to the site records in different season, with mean bias (MB) of -0.3 ~ 0 ℃, -0.8 ~ -0.5 m/s and -4~ -2.3% for $T_2$, $W_{10}$ and $RH_2$ respectively. The model underestimates $T_2$ in all the seasons. The summer showed a negative bias with Root Mean Square Error (RMSE) of 2 ℃. As for $W_{10}$, it's also underestimated the most in summer, with MB of -0.8m/s and RMSE of 1.9 m/s. As for RH2, the underestimation is more obvious in summer (MB= -3.2%) and autumn (MB= -3.2%), mainly stem from the insufficient precipitation. Overall, the agreement in $T_2$ and $RH_2$ with observations is better than that of $W_{10}$, with annual correlation coefficients (R) of 0.98, 0.84 and 0.53, respectively. Generally, the meteorology calculated by WRF can rationally reproduce the characteristics of observations.

**Changes in the manuscript**: Captions and units are added in Table 2. The description about Table 2 refers to Line 257-267.

- L270: Why did you take average of 2006-2015 only for WDCGG?

**Reply:** The WDCGG datasets provides a large number of trace gases observations globally. The datasets can help to evaluate model performance of CO and ozone in different regions, but there are no observations for some sites in 2014. To get more data to evaluate the model over the world, we expanded the time range to ten years (2006-2015) and take the average of 2006-2015 as the statement of the air in the initial evaluation.

We have re-selected the observation data for 2014 to comparison with model results. Overall, the results have not changed much in terms of the evaluation of model's simulation capability. The simulation of $NO_2$ performed better with the NMB closer to zero in Asia and Europe. The underestimation of CO in Antarctica disappeared due to the change of the observed value. There are some changes in the trend of the seasonal variation of $O_3$ in Northern Hemisphere.

**Changes in the manuscript**: All the figures and tables related to these changes are updated in the manuscript (please refer to Fig. 4 and Fig.6), and the corresponding analysis is updated in the manuscript, too (please refer to Sect. 3.2.2).

- Table3: It is better to include the information of region of each observation datasets.

**Reply:** That's a good suggestion, we have added the region of each observation datasets in Table3.

- L298: The value (23.3) differs from that in Table4.

**Reply:** Thanks, it's a typo. We have corrected it to 22.8 in the sentence in Line 324.

- Table5: Table 5 is not completely filled with the necessary information for OM (other sources and total sink

are missing).

**Reply:** Other sources and total sink are added to Table 5.

- Figure4: Fig4c and 4d should be switched to be in accordance with the order of panels in Fig3.

**Reply:** The subplot in Fig. 4(b) has been switched to the same order as Fig. 4(a) in the revised manuscript.

- L363: Typo?: Northern Hemisphere

**Reply:** Yes, we have revised it. Please refer to Line 407.

L368-369: Why did IAP-AACM show the lowest concentration of CO over ocean among the models considered here?

**Reply:** It potentially reflects a difference in emissions. The natural sources of CO over ocean are included in the HTAP models whereas they are not considered in IAP-AACM. Besides, it may reflect differences in chemical transformation between models. The tropospheric (200hpa to the surface) mean OH concentration of IAP-AACM is $13.0 \times 10^5$ molec cm$^{-3}$. It is a little higher than the mean OH concentration study ($11.1 \pm 1.6 \times 10^5$ molec cm$^{-3}$) from 16 ACCMIP models for 2000 by Naik et al. (2013). It potentially leads to strong atmospheric oxidation. As shown in Figure 4, there is a slightly higher peak concentration of 30-35 molec cm$^{-3}$ in IAP-AACM, compared with the other models (under 30 molec cm$^{-3}$) (Huijnen et al., 2010; Badia et al., 2017). Due to the sink reaction of CO (CO + OH → CO$_2$ + H), the CO loss is larger in IAP-AACM.

**Changes in the manuscript**: More discussion on the underestimation of CO refers to Line 396-401.

- L378-380: The seasonal variation of surface O3 should be different in different environment even in the same region. So, I recommend the authors to compare separately for different environment (e.g. maritime area vs mountainous area). Otherwise, I can not regard the Fig6 as an evidence that the model can well simulate the seasonal variation of surface O3.

**Reply:** That's a helpful suggestion. The seasonal cycle of ozone shows different characteristics in different topographic conditions due to different control factors. We separate the observational sites as maritime area, inland and mountain due to the altitudes (shown in Figure 13). For inland, the model tends to overestimate O$_3$ concentrations in summer time. The simulation of cloud may contribute to the positive bias. Furthermore, uncertainties in volatile organic compounds (VOCs)-NOx-O$_3$ chemistry may contribute. The natural source of isoprene from vegetation is important in the O$_3$ formation due to its high proportion of VOCs emission in summer (as estimated to be 40.9 Tg/yr in China by Fu et al., 2012).

**Changes in the manuscript**: Please refer to 456-464.

[Figure]

Figure 13 Mean seasonal variation of $O_3$ (ppb) over inland, mountain and maritime area in Northern Hemisphere compared with site records. Black lines and red lines represent average observations and simulations respectively. Gray shaded areas and red vertical bars show 1 standard deviation over the sites for observations and for model results respectively.

- L385: Underestimation in Antarctica is not small. Such an underestimation could be seen in the other CTMs. Can you use more words about this issue here?

**Reply:** In IAP-AACM, ozone concentration is about 5~15 ppb lower than site observations in Antarctica. It may be caused by the lack of halogen chemistry in the model. Remarkable ozone depletion events which is driven by halogen chemistry (mostly notably as bromine) is observed in the polar boundary layer (Simpson et al., 2007). The model study by Falk & Sinnhuber (2018) indicated that there are missing sources of bromine release from ice and snow in EMAC v2.52. The over prediction of dry deposition velocity to sea ice also plays a role here. The dry deposition velocity to ice is under 0.02 cm s$^{-1}$ across 15 HTAP models (Hardecre et al., 2015). In IAP-AACM, it's higher (0.035~0.048 cm s$^{-1}$) than those models, as shown in Figure 14.

**Changes in the manuscript**: The discussion on the underestimation of ozone in Antarctica refers to Line 465-471

[Figure]

Figure 14 Annual mean dry deposition velocity of ozone in IAP-AACM. The unit is cm s[-1].

- L385-387: In the NH land area, it seems that the model completely failed to represent the seasonal cycle of surface O3, but the author regarded it as just a positive bias during July-September. More words for this issue are necessary. For example, what do you think the apparent underestimation in cold season in NAmerica and Europe?

**Reply:** Yes, the model showed poor performance on the seasonal cycle of surface ozone in the NH land, with overestimation in Europe and EA in summer while underestimation in NH land in winter, as shown in Figure 9 (the comparison drawn with WDCGG observation only for 2014).

The surface $O_3$ are underestimated in cold seasons over NH land. In IAP-AACM, the stratospheric chemistry is not considered. Thus the stratospheric-tropospheric exchange is weak. It leads to a large negative bias in simulation. To date it has become apparent that the measured annual cycle of ozone shows a distinct maximum during spring. The stratosphere-to-troposphere ozone transport event occurs widely across mid-latitudes in the NH (Monks et al., 2000; Akritidis et al., 2018). Since the magnitude and frequency of the transport through tropopause is still not clear. There are large uncertainties in simulating the flux. Some researches (Munzert et al, 1985; Austin and Follows, 1991) showed that the maximum in the stratosphere to troposphere flux occurs in late winter/spring. It may partly explain the underestimation of $O_3$ in winter.

The surface $O_3$ concentrations over East Asia (sites mainly located in Japan) are overestimated in summer and early autumn. The same pattern is also found in the multi-model inter-comparison of 21 HTAP models (Fiore et al., 2009). The simulations in island countries of EA are sensitive to the timing and extent of the Asian summer monsoon (Han et al., 2008). The positive model bias in this season may stem from inadequate representation of southwesterly inflow of clean marine air. Furthermore, the underestimation of cloud cover in summer may also responsible for the overestimation of $O_3$ due to stronger photochemistry. Additionally, it's difficult for global model with coarse resolution to resolve local orographically driven flows or sharp gradients in mixing depths under complex underlying surface conditions in lands.

**Changes in the manuscript**: The discussion on the seasonal cycle of ozone in NH land refers to Line 435-456.

- L388-390: In Badia et al (2017), they suspected the excessive emission height of NOx which will cause low NOx at surface and consequently might lead to weak NO titration. Do the same things happen in your model?

**Reply:** Yes, there is the same situation in our model. In Badia et al. (2017), all the land-based anthropogenic emissions are emitted in the first 500m of the model. In IAP-AACM, the energy emissions and industry emissions are emitted in the first five layers considering the stack height.

- L390-391: The AACM apparently showed larger concentration of surface O3 in the tropical regions (central Africa, South America, and Southeast Asia) than the other models. However, the concentration of O3 precursor species (CO and NOx) in these regions are not so different among the models. Can you give discussion about the issue here?

**Reply:** Yes, the concentrations of CO and NOx in the tropical regions are not so different among the models. There are several uncertainties in the model performance. Even the same module schemes applied in different models may display different result (Tsigaridis et al. 2014; Hardecre et al., 2015). Furthermore, the meteorological conditions also play an important role in simulation. The chemical reactions and dynamical processes (transportation and diffusion) of the matters are sensitive to meteorological field (e.g., wind, precipitation, cloud cover, temperature). In addition, the biomass burning emissions used in IAP-AACM is different from the other models. For multi-model activities of HTAP, groups use GFED3 data as the biomass burning emissions (Galmarini et al., 2017). In IAP-AACM, we use GFED4. A comparison of different versions of GFED emissions (Werf et al., 2017) shows the impact of a minor reduction in burned area and decreasing fuel consumption.

- L391-394: These two sentences are not consistent to each other. In general, the region of high O3 concentration can be different in different season. If you look at the "annual mean" concentration, the highest O3 usually occur in the source region in summer, but that in the downwind region in winter. However, if you see the different index such as MD8H O3, you can see completely different seasonal cycle. I strongly recommend the author to carefully revise these sentences.

**Reply:** We totally agree with your comment. We have revised those sentences. Please refer to Line 425-430.

- L402-403: An overall evaluation of O3 dry deposition in global CTMs can be seen in Hardecre et al. (2015). I recommend to check it out. Hardacre et al. (2015) An evaluation of ozone dry deposition in global scale chemistry climate model, Atmos. Chem. Phys., 15, 6419–6436, doi:10.5194/acp-15-6419-2015.

**Reply:** Thanks for your suggestion. According to Hardecre et al. (2015), the dry deposition velocity to sea varies little (around 0.05 cm s$^{-1}$) in different CTMs models using the deposition scheme by Wesely (1989). Besides, the study of Ganzeveld et al. (2009) shows that surface ozone differed by up to 60% if the deposition velocity of ozone varies from 0.01 to 0.05 cm s$^{-1}$. In IAP-AACM, the deposition velocity over the oceans varies from 0.042 to 0.05 cm s$^{-1}$, as shown in Figure 14. The variation in absolute terms between IAP-AACM

and the other models is smaller than 0.008 cm s$^{-1}$. Hence the difference of surface ozone caused by dry deposition should be less than 12%.

**Changes in the manuscript**: We revised sentences in the manuscript. Please refer to Line 428-434.

- L416-418: The concentration of NOx over oceanic areas are larger in AACM than in other models, which might stem from larger emission or longer life time of NOx in AACM than the other models. I recommend to discuss this issue further here.

**Reply:** We totally agree with reviewer's suggestion. Compared with the other models shown in Fig. 5, the surface $NO_2$ over ocean is larger in IAP-AACM. This may reflect larger emission or less sinks of $NO_2$ in IAP-AACM. The higher injection height of emission sources leads to further transportation distance and low $NO_x$ at surface of source areas. Consequently, it leads to higher concentration of surface ozone in NH source areas due to weak NOx titration.

**Changes in the manuscript**: This is added in Line 585-590 in the revised manuscript.

- L436-438: This is misleading statement. The model results are not generally with in a factor of two, but they apparently tend to overestimate the observation in all the three regions. The NMB value for sulfate in Europe, 0.11, is incorrect which is 1.1 in Table 6.

**Reply:** The '0.11' is a typo, we have corrected it and renewed a new description about the simulation of sulfate as follows. As shown in Fig. 8, Sulfate is overestimated more or less. Specifically, in Asia, the simulations at most sites here are within a factor of two of observations, with NMB of 0.36. However, In NAmerica and Europe, it's overestimated with NMB of 1.94 and 1.1 respectively.

**Changes in the manuscript**: Please refer to Line 505-508.

- L438-439: How can you conclude like this (2ugm-3 higher)? What is the ground of this statement?

**Reply:** We calculated the sites average value in the model and compared it against the observation but we didn't mention it, we have added this description in the revised paper.

**Changes in the manuscript**: Please refer to Line 508-510.

- L442-443: What aspect of the observation do you think your model can reproduce? You should be more specific.

**Reply:** As shown in Fig. 8 in the revised manuscript, the concentration of nitrate is higher in eastern America and lower in western America. IAP-AACM reproduces the distribution of nitrate in western America well but overestimates it in eastern America. The model doesn't fully capture the spatial variation over Europe, with an overestimation at most of the sites. As for Asia, there is an underestimation in Southeast Asia and Japan.

**Changes in the manuscript**: We have provided a detailed description of the distribution in the revised

manuscript. Please refer to Line 514-518.

- L446-449: About the simulation of ammonium, I can see obvious underestimation in NAmerica and overestimation in Asia and Europe.

**Reply:** The performance of ammonium varies in different regions since there are more uncertainties in the emission of NH3 (precursor of ammonium) from croplands (Xu et al., 2019). There is slight negative bias in America and positive bias in Asia, with NMB less than ±1 (-0.46 and 0.85 respectively). In Europe, there is significant positive bias with NMB of 1.49.

**Changes in the manuscript**: This has been added to the revised manuscript. Please refer to Line 520-525.

- L455-457: The concentration of OC were obviously underestimated by the model.

**Reply:** Yes, the meteorological conditions and emission inventories in the model are inconsistent with the observation year (2006) of carbonaceous in China. This may be partially responsible for the bias of OC. According to recent study, there is a slightly increasing (less than 0.1Tg) of both BC and OC emissions from 2006 to 2010 in China (Lu et al., 2011; Fu et al., 2012). As shown in Fig. 7, the simulation of BC at most sites are close to observations while the simulation of OC is significantly underestimated. The study by Fu et al. (2012) showed a significant underestimation of OC emissions over China. Furthermore, Zhao et al. (2016) found that the pathway of intermediate volatile organic compounds (IVOC) to SOA is very important for the formation of SOA. Their model experiments suggest that IVOCs constitute over 40% of OM concentrations in Eastern China. Yang et al. (2018) also showed the significant increase of SOA concentration in an observation-based box model which included the IVOCs reactions. IVOC reactions are not included in our SOA module. The SOA module in IAP-AACM is Two-Product scheme. Model studies with Two-Product scheme estimated an underestimation of OM by 40-78% in China (Lin et al., 2016; Han et al., 2016). Thus the closely simulating of BC but greatly underestimating of OC requires an improvement in SOA formation mechanism in IAP-AACM.

**Changes in the manuscript**: The discussion refers to Line 536-553.

- L491-492: The highest AOD in DJF in east China is not clearly seen both in satellite and model AOD.

**Reply:** Yes, it's an incorrect expression here and we have deleted it. In fact, the highest AOD may not be in DJF, it often appears in MAM. Since East Asia is frequently affected by dust in spring, this phenomenon is common in other model evaluation studies (e.g., GISS-TOMAS (Lee et al., 2010)). The seasonal variation of relative humidity also impacts the simulating of AOD.

- Figure10: It's better to show scatter plots too, at least as a supplement figure.

**Reply:** The figure is added in the revised manuscript, as Fig. 11(b). Descriptions refer to Line 621-625.

- Figure11: The area and the map projection of the figures for all models should be united.

**Reply:** The model results have been adapted to the same area and projection (shown in Figure 15).

**Changes in the manuscript**: The new figure is updated to Fig. 12 in the manuscript.

[Figure]

Figure 15 Annual surface distributions from nested IAP-AACM compared with regional models from MICS-Asia. Each row from top to bottom represents IAP-AACM, WRF-Chem, CMAQ and NAQPMS respectively. The left column is $SO_2$ (ppb), the middle column is $NO_2$ (ppb) and the right column is $PM_{2.5}$ (μg m$^{-3}$).

- L517-519: I'm sorry I can not understand what you want to mean here.

**Reply:** Since the boundary conditions are provided by the parent grids in IAP-AACM, it's updated real-time. The spatial distribution of pollutant near boundary areas varies consecutively if we check the hourly simulation. But as for the annual averaged results, the discrepancy is masked.

L545-547: What do you want to mean here? Your model overestimated the NO2 in summer in NC and YRD regions. If you don't use the NO2* observation, the model's overestimation should become worse.

**Reply:** As shown in Figure 12, the annual mean simulations of $NO_2$ are in good agreement with observations. Due to the $NO_2$* observation, it implicates that the model may overestimate "$NO_2$". As shown in Fig. 14 in the

manuscript, the $NO_2$ is overestimated in some cities in summer. It is associated with deposition removal process and multiphase chemistry in IAP-AACM. The overestimation of $NO_2$ and underestimation of nitrate in daytime of summer and autumn relates to over decomposition of nitric acid at high temperature condition in the thermodynamic equilibrium module. Moreover, heterogeneous chemical reactions in the model should partly be responsible for the overestimation in summer. The heterogeneous chemical module coupled in IAP-AACM has been applied in North China in winter (Li et al, 2018). The mechanism significantly improved sulfate simulation under highly polluted conditions (contributing 50%-80% of total concentration of sulfate) and reduced the overestimation of nitrate. However, the simulations excluded heterogeneous chemical processes showed better performance of $NO_2$ (shown in Fig. S5). It indicates that a more reasonable mechanism should be considered in model development.

**Changes in the manuscript**: The discussion on the overestimation of $NO_2$ is revised in Line 663-674. The seasonal cycle of $NO_2$ ($\mu g\ m^{-3}$) simulated without heterogeneous chemical process is supplemented as Fig. S5.

[Figure]

Figure 16 Seasonal cycle of $NO_2$ ($\mu g\ m^{-3}$) simulated without heterogeneous chemical process over China. The black line and red line represent monthly mean concentration of city-averaged observation and simulation respectively. Gray shaded areas and red vertical bars show 1 standard deviation over the sites for observations and for model results, respectively. MO and MM stand for annual mean concentration of observation and simulation respectively.

- L547-550: I can not understand what aspect of seasonal difference in NO2 column observation were reproduced by your model. You should describe more specifically on it.

**Reply:** As shown in Fig. S3, the model captures seasonal variations of $NO_2$ column concentrations in the vertical troposphere well. In China, the $NO_2$ VTC is higher during September-October-November and December-January-February while lower in June-July-August, due to unfavorable diffusion conditions and weaker photochemical reactions.

**Changes in the manuscript**: The updated description refers to Line 676-679.

- L595: Typo? respects ! aspects

**Reply:** Yes, we have revised it.

- Conclusions should be revised according to the modifications made to respond the reviewers comments.

**Reply:** Yes, Conclusions and Abstract has been updated in the manuscript.

**Changes in the manuscript**: Please refer to Abstract and Conclusions.

[revised manuscript text omitted]

EANET

[revised manuscript text omitted]

**Figures**

[Figure]

Fig. S1. Annual mean precipitation of WRF compared with GPCP data. The left column is WRF simulation (unit: mm), the middle column is GPCP reanalysis data (unit: mm), the right column is the difference between simulation and reanalysis (WRF-GPCP) (unit: mm day$^{-1}$).

1280

[Figure]

Fig. S2. Annual mean dry deposition velocity of ozone in IAP-AACM. The unit is cm s$^{-1}$.

[Figure]

1285

Fig. S3. Mean seasonal variation of $O_3$ (ppb) over inland, mountain and maritime area

in Northern Hemisphere compared with site records. Black lines and red lines represent the average of observations and simulations respectively. Gray shaded areas and red vertical bars show 1 standard deviation over the sites for observations and for 1290    model results respectively.

[Figure]

Fig. S4. Gridded mean value of monthly averaged AOD for 2014, AF, EA, SA and GL represents Africa, East Asia, South America and global. Dash line and solid line represents model results and observation derived from MODIS, respectively.

[Figure]

1295

Fig. S5. Seasonal cycle of $NO_2$ (µg m$^{-3}$) simulated without heterogeneous chemical process over China. The black line and red line represent monthly mean concentration of city-averaged observation and simulation respectively. Gray shaded areas and red

vertical bars show 1 standard deviation over the sites for observations and for model

1300 results, respectively. MO and MM stand for annual mean concentration of observation and simulation respectively.

[Figure]

Fig. S6. Seasonal mean column concentration ($10^{14}$ molecule $cm^{-2}$) of $NO_2$ in IAP-AACM and GOME-2A over China.